# SpinBench: Perspective and Rotation as a Lens on Spatial Reasoning in VLMs

**Yuyou Zhang[1,2], Radu Corcodel[2], Chiori Hori[2], Anoop Cherian[2], Ding Zhao[1]**
[1]Carnegie Mellon University, [2]Mitsubishi Electric Research Labs
{yuyouz,dingzhao}@andrew.cmu.edu,
{corcodel,chori,cherian}@merl.com

## Abstract

We present SpinBench, a cognitively grounded diagnostic benchmark for evaluating spatial reasoning in vision language models (VLMs). SpinBench is designed around the core challenge of spatial reasoning: perspective taking, the ability to reason about how scenes and object relations change under viewpoint transformation. Since perspective taking requires multiple cognitive capabilities, such as recognizing objects across views, relative positions grounding, and mentally simulating transformations, SpinBench introduces a set of fine-grained diagnostic categories. Our categories target translation, rotation, object relative pose, and viewpoint change, and are progressively structured so that single-object simpler tasks scaffold toward the most demanding multi-object perspective-taking setting. We evaluate 43 state-of-the-art VLMs, both proprietary and open source. Results reveal systematic weaknesses: strong egocentric bias, poor rotational understanding, and inconsistencies under symmetrical and syntactic reformulations. Scaling analysis shows both smooth improvements and emergent capabilities. While human subjects achieve high accuracy (91.2%), task difficulty as measured by human response time shows strong correlation with VLM accuracy, indicating that SpinBench captures spatial reasoning challenges shared across humans and VLMs. Together, our findings highlight the need for structured, cognitively inspired diagnostic tools to advance spatial reasoning in multimodal foundation models. Our website can be found at https://spinbench25.github.io/.

## 1 Introduction

Spatial reasoning is a fundamental component of human cognition and a key capability for embodied agents operating in the physical world (Xia et al., 2018). From recognizing object configurations to simulating motion and perspective changes, spatial understanding enables agents to interpret their environment and plan actions accordingly.

Multimodal foundation models, particularly vision-language models (VLMs), have recently achieved impressive progress in visual understanding (Li et al., 2025b; Wang et al., 2025; Qwen et al., 2025; Team et al., 2025; Li et al., 2024b), however their spatial reasoning capabilities remain poorly understood and underdiagnosed. The demonstrated utility in downstream tasks, such as navigation (Elnoor et al., 2025; Song et al., 2024), manipulation (Yang et al., 2025c; Sermanet et al., 2024), autonomous driving (Xie et al., 2025; Pan et al., 2024), and physical commonsense reasoning (Chow et al., 2025) primarily reflects end-to-end performance at the application level, where spatial reasoning is entangled with high-level language and planning objectives. They do not directly test whether models understand geometric primitives, such as rotation, translation, object-relative pose, and viewpoint changes and thus can not expose failures underlies spatial intelligence.

As a result, it remains unclear whether VLMs are genuinely capable of spatial reasoning, or whether they rely on dataset biases and shallow pattern matching. Recent benchmarks like MindCube (Yin et al., 2025) and Space (Ramakrishnan et al., 2024) reveal striking failures in mental modeling and spatial generalization, often exposing large performance gaps between models and humans. While efforts such as SpaceOm and Spacethinker (Chen et al., 2025a) explore linguistic training enhance-

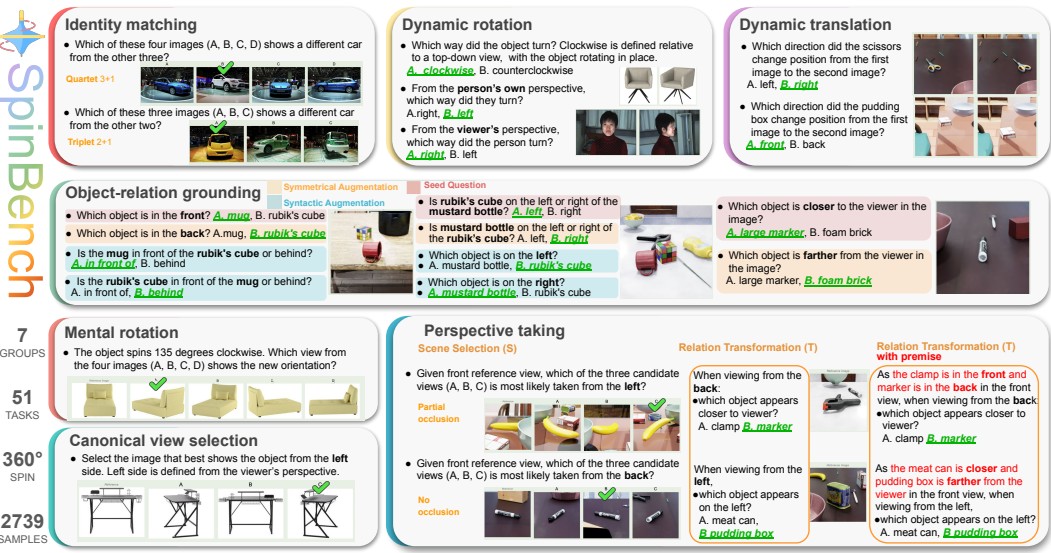

Figure 1: Overview of SPINBENCH task design across seven task groups. Representative subtasks are illustrated for each group with simplified question wording for clarity. In the released benchmark, all queries include explicit frame-of-reference definitions to avoid ambiguity. Human face data are sourced from the Stereo Face Database Fransens et al. (2005) and are licensed for research use only.

ments via reinforcement learning, they still exhibit limited transfer of these gains to spatial reasoning tasks (Yin et al., 2025). This calls for a structured diagnosis of: (1) what specifically breaks down in VLMs' spatial reasoning, and (2) how such reasoning can be systematically evaluated.

Our approach is inspired by foundational insights in cognitive science. Early behaviorist theories treated thinking as verbal behavior (Skinner, 1957), but classic mental rotation experiments (Shepard & Metzler, 1971) demonstrated that spatial cognition often depends on analog, imagery-based processes-continuous, imagistic simulations that go beyond linguistic representations. These insights motivate the central question: *Can VLMs engage in such imagery-based spatial reasoning, or are they limited to symbolic and linguistic associations?*

To address this, we introduce **SPINBENCH**, a cognitively grounded, diagnostically structured benchmark as shown in Fig. 1. Our design is informed by both psychological paradigms and system-level considerations. SPINBENCH emphasizes **progressive structure, cognitive fidelity, and controlled variation** for diagnostic value. Our progression of tasks reflects increasing spatial complexity and scale (Hegarty et al., 2006): At the low level, we assess single-object perception tasks such as *object identity matching*, *canonical view selection*, *mental rotation*, and *dynamic translation/rotation*; At the higher level, we evaluate *object-elation grounding* and *perspective taking* in cluttered, multi-object scenes. Our most challenging task, multi-object cluttered scene *perspective taking*, requires models to integrate subskills from all prior tasks, making it a holistic probe of spatial cognition. We include both real-world and photo-realistic synthetic data across diverse domains (e.g., household objects, vehicles, human faces), ensuring validity while maintaining evaluation rigor. Each task type is carefully designed to evaluate specific spatial skills and is embedded within a controlled variation regime: we manipulate frame-of-reference (FoR) (Zhang et al., 2025c), introduce premise-based question structures, apply syntactic and symmetrical augmentations, and vary the number of visual inputs (e.g., single, triplet, quartet). These tasks serve as interpretable bridges from raw perceptual features to fundamental spatial concepts and then to challenging spatial reasoning.

Together, SPINBENCH provides an interpretable and rigorous framework for diagnosing the spatial reasoning capabilities of modern VLMs and for understanding the role of rotation as a window into 3D spatial understanding. Our empirical analysis reveals key failure modes in VLM spatial reasoning: persistent egocentric bias, difficulty with rotation and viewpoint changes, inconsistencies in handling symmetry, and failures in linguistic-only spatial inference. We also observe diverse scaling behaviors across tasks and limited correlation with existing benchmarks, suggesting that SPINBENCH offers novel and complementary diagnostic insights into VLM spatial competence.

## 2 RELATED WORK

**Spatial reasoning benchmarks**  A wide range of benchmarks have been proposed to evaluate the spatial reasoning abilities. Early diagnostic datasets like CLEVR (Johnson et al., 2017) introduced synthetic, rendered scenes with simple 3D shapes. Recent spatial reasoning benchmarks for vision-language models have explored diverse aspects of spatial cognition. Some, such as Mind-Cube and VSI-Bench (Yin et al., 2025; Yang et al., 2025b), emphasize cognitive mapping, how models represent and track spatial information across scenes. SpaCE-10, SPHERE, and 3DSR-Bench (Gong et al., 2025; Zhang et al., 2024; Ma et al., 2025a) define a range of atomic spatial skills (e.g., counting, height, orientation), yet often lack controlled variation in perspective, reference frame, or multi-frame reasoning. BLINK (Fu et al., 2024) highlights perception-level gaps in multimodal models, and ViewSpatial-Bench (Li et al., 2025a) focuses on viewpoint-dependent localization. MulSeT (Zhang et al., 2025b) covers distance, occlusion, and viewpoint-dependent localization with synthetic data. Meanwhile, OmniSpatial, 3D-PC and SPACE (Jia et al., 2025a; Linsley et al., 2024; Ramakrishnan et al., 2024) draw from cognitive psychology to design spatial tasks, but sometimes entangle spatial reasoning with functionality and physical commonsense or are limited to abstract 2D plane geometry. Our tasks are carefully designed to isolate spatial reasoning by controlling for distractors, motion dynamics, reference frame shifts, and multi-image input formats. We incorporate both real-world and photo-realistic synthetic data to ensure domain diversity and real-world relevance. Instead of emphasizing task comprehensiveness, SPINBENCH offers diagnostic value by introducing fine-grained control over key spatial factors such as premise structure, symmetry, and syntactic variation. As summarized in Tab. 1, our benchmark uniquely combines progressive task structure, cognitive grounding, and controlled variation. For quantitative evidence that SPINBENCH targets spatial skills that differ from prior spatial VLM benchmarks, we refer readers to Appendix B.3, where we provide full correlation analyses showing weak correlation and task-level orthogonality with prior benchmarks.

| Benchmark | Reference Var. | Premise Var. | Symmetric Var. | Syntactic Var. | Domain | Multi-Image | Tasks | Size |
|---|---|---|---|---|---|---|---|---|
| CLEVR Johnson et al. (2017) | ✗ | ✗ | ✗ | ✗ | cubes | ✗ | 90 | 853k |
| BLINK Fu et al. (2024) | ✗ | ✗ | ✗ | ✗ | mixed | ✓ | 14 | 3.8k |
| SpaCE-10 Gong et al. (2025) | ✗ | ✗ | ✗ | ✗ | indoor | ✗ | 8 | 6k |
| 3DSRBench Ma et al. (2025a) | ✗ | ✗ | ✓ | ✗ | mixed | ✗ | 12 | 2.8k |
| SPHERE Zhang et al. (2024) | ✓ | ✗ | ✗ | ✗ | MsCOCO | ✗ | 9 | 2.3k |
| ViewSpatial Li et al. (2025a) | ✓ | ✗ | ✗ | ✗ | ScanNET, MsCOCO | ✗ | 5 | 5.7k |
| MindCube Yin et al. (2025) | ✓ | ✗ | ✗ | ✗ | indoor/outdoor | ✓ | 4 | 21k |
| OmniSpatial Jia et al. (2025a) | ✓ | ✗ | ✗ | ✗ | web, driving, tests | ✓ | 50 | 1.5k |
| **SpinBench (Ours)** | ✓ | ✓ | ✓ | ✓ | Household, car, face, infinigen (Raistrick et al., 2024) | ✓ | 51 | 2.7k |

Table 1: Benchmark comparison highlighting the controlled structure and diagnostic focus of SPIN-BENCH. Our benchmark supports reference frame variations, premise-based variations, symmetric and syntactic variations, and multi-image spatial reasoning across both real and synthetic domains.

**Spatial reasoning models**  To improve spatial reasoning in VLMs, recent work has explored 3D abstractions and finetuning. Methods like SpatialReasoner (Ma et al., 2025b), SSR (Liu et al., 2025) and APC (Lee et al., 2025) use explicit 3D representations for perspective-aware reasoning, . Others, such as MetaSpatial (Pan & Liu, 2025), Embodied-R (Zhao et al., 2025), SpatialVLM (Chen et al., 2024), and SVQA-R1 (Wang & Ling, 2025), adopt reinforcement learning or large-scale pretraining to enhance spatial understanding across 2D and video data. Despite progress, purely linguistic approaches remain limited, humans rely on structured, often non-verbal representations to reason about space, motivating models that move beyond language-based reasoning alone.

## 3 DATASET AND BENCHMARK RECIPES

### 3.1 DIAGNOSTIC APPROACH TO SPATIAL REASONING

SPINBENCH is designed around the core challenge of perspective taking: reasoning about how scenes and object relations change under viewpoint transformation. Perspective taking is a highly integrative ability as it requires recognizing objects across views, grounding their relative positions, and mentally simulating their transformations. To better diagnose model strengths and weaknesses, SPINBENCH decomposes this advanced reasoning evaluation into a set of targeted diagnostic categories. Each category represents a fundamental spatial reasoning ability that supports perspective taking, such as object identity recognition, relation grounding, translation, and rotation. Together,

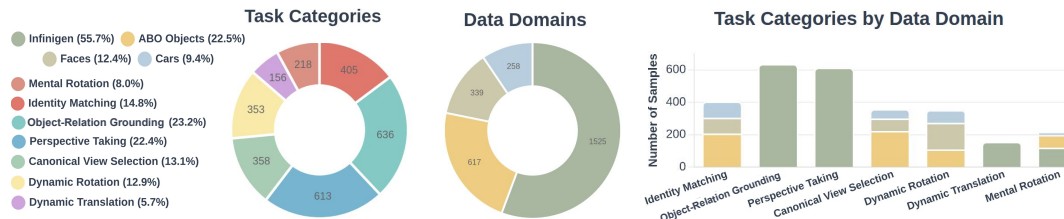

Figure 2: Distribution of SPINBENCH tasks across seven spatial reasoning categories and four visual domains. Right: Task breakdown by domain.

these tasks allow us to disentangle where current vision language models succeed, where they fail, and how these skills compose in the perspective taking setting. To minimize confounds, all tasks are defined in a horizontal 2D plane. Vertical relations (e.g., above/below) and height differences are excluded, and viewpoint changes are restricted to horizontal orbits around the scene.

### 3.2 TASK CATEGORIES AND DESIGN RATIONALE

The seven categories below are organized so that simpler diagnostic abilities scaffold toward the most demanding task: perspective taking. Representative examples for each category are summarized in Figure 1. Task category details are in the Appendix A.3.

1. **Identity Matching** Evaluates whether models can consistently recognize the same object across different viewpoints. This ability is a prerequisite for cross-view reasoning, ensuring models can track object identity before more complex spatial inference.

2. **Object-Relation Grounding** Tests understanding of object-relative configurations within a single static image, including directional relations (left/right, front/behind) or distance relations (near/far) between two objects. This isolates spatial grounding from temporal or multi-view demands, providing a controlled measure of static scene interpretation.

3. **Dynamic Translation** Assesses reasoning about linear object displacement over time. Given two temporally ordered frames of the same object, models must identify whether it moved left, right, front, or back relative to the viewer. By excluding rotation, this category isolates translational understanding from other motion cues.

4. **Dynamic Rotation** Focuses specifically on rotational transformations. Models are given two images of an object before and after in-place rotation and must determine the rotation direction (e.g., clockwise vs. counterclockwise, defined from a top-down view). Restricting the task to a single rotated object avoids background or displacement confounds, allowing fine-grained analysis of rotational reasoning.

5. **Canonical View Selection** Examines whether models can map objects across canonical viewpoints. Given a reference view (typically the front), models must select the correct candidate from alternative perspectives (left, right, back). This setting avoid the complexity of multi-object scenes.

6. **Mental Rotation** Tests whether models can mentally simulate object transformations. Given an object and specified degree and direction of rotation, models must select the correct resulting configuration. This requires internal spatial visualization and supports analysis of whether models can simulate transformations beyond what is directly observed.

7. **Perspective Taking** The centerpiece of SPINBENCH, perspective-taking tasks require reasoning about entire scenes under viewpoint changes. Two subtypes are included: (S) selecting the correct scene image from a new perspective, and (T) predicting how object relations transform under perspective shifts. This category integrates all diagnostic abilities and probes compositional spatial reasoning in its most demanding form.

### 3.3 DATASET COMPOSITION AND DOMAIN COVERAGE

SPINBENCH combines one simulation-generated synthetic dataset with three real-world datasets, chosen to test spatial reasoning generalization across diverse visual domains and object categories. A

detailed breakdown of dataset composition, sampling strategies, and annotation pipelines is provided in the Appendix A.

- **Infinigen Scenes** We generate indoor environment table-top multi-object synthetic scenes using Infinigen Raistrick et al. (2024) in the Isaac Sim environment NVIDIA, with objects drawn from the YCB dataset Calli et al. (2015; 2017). Randomized object selection, placement, and lighting yield diverse yet controlled settings. Data are generated for three task categories: *object-relation grounding*, *dynamic translation*, and cluttered scene *perspective taking*. For the *perspective taking*, we provide occlusion and no-occlusion variants to probe reasoning under visual ambiguity.

- **ABO Objects** We sample household items from the Amazon Berkeley Objects (ABO) dataset Collins et al. (2022), which provides high-quality 3D models of real commercial products. Objects include 360° views (72 images at 5° intervals) with diverse geometries and textures. We select geometrically structured objects and exclude highly symmetrical cases to avoid ambiguous rotation or relation judgments.

- **Cars** Vehicle rotation sequences are drawn from the Multi-View Car Dataset Ozuysal et al. (2009), which contains 20 cars imaged every 3–4 degrees during a full 360° rotation. Cars are ideal for viewpoint-dependent reasoning due to their strong canonical orientations (front, back, side views). Since degree annotations are not provided, we sample and label images at 45° intervals to ensure consistent angular coverage.

- **Faces** Human faces are sourced from the Stereo Face Database Fransens et al. (2005), containing 100 individuals captured in 8 distinct poses. Faces pose biologically relevant challenges and require distinguishing viewer- versus object-centered reference frames. Their natural asymmetry (left vs. right profiles) enables unambiguous evaluation of perspective-taking.

## 3.4 CONTROLLED VARIATIONS

SPINBENCH is designed with fine-grained, controlled variations to evaluate how models handle allocentric and egocentric reference, integrate visual and linguistic information, and model reasoning consistency with symmetric and syntactic variations, providing a diagnostic lens for identifying systematic biases, inconsistency, or modality-specific weaknesses. Detailed variations and examples are provided in the Appendix A.2 and A.4.

**Allocentric and Egocentric Reference** Reference frame ambiguity is a common source of error in pretrained models, arising because natural language often leaves the frame of reference implicit. Humans flexibly switch between defaults (e.g., egocentric vs. allocentric) depending on context, but models may struggle without explicit cues. Our face rotation tasks directly test this by presenting identical transformations under two interpretations: the viewer's perspective (e.g., "turn left" as seen by the observer) versus the object's own perspective (e.g., "turn left" as for the person). This contrast reveals whether models exhibit systematic biases toward particular frames or can adapt to contextual cues. In domains where objects lack intrinsic orientation, all relations are defined from the viewer's (camera) perspective to ensure consistency.

**Consistency via Data Augmentation** To probe reasoning stability, we systematically generate equivalent variants of spatial relation tasks using two augmentation strategies: *(i) Symmetrical augmentation*: Logically equivalent variants are created by flipping relations and answers (e.g., from "Which object is on the left?" to "Which object is on the right?"). This ensures models maintain consistent reasoning under symmetrical transformations. *(ii) Syntactic augmentation*: Questions are reformulated while preserving meaning (e.g., "Which object is on the left?" → "Is A on the left or right of B?"). This tests whether models rely on surface phrasing or demonstrate robust spatial understanding. Augmentations are applied across static (left/right, near/far, front/behind), with combined variants yielding comprehensive test sets for consistency evaluation.

**Visual vs. Linguistic Failures** To disentangle sources of error, we introduce premise-based task variants. In the *with-premise* condition, the spatial relation (e.g., "A is to the right of B in the front view") is explicitly provided in the prompt, while in the *without-premise* condition, models must

infer relations solely from the image. Comparing performance across conditions reveals whether failures stem from visual grounding difficulties or from applying geometric reasoning when the premise is known.

# 4 EVALUATIONS

## 4.1 EVALUATION SETUP

**Evaluated models** We evaluated 43 vision-language models spanning both proprietary and open-source models to assess spatial reasoning capabilities across diverse model scales and designs. We included 7 proprietary VLMs: GPT-5, o4-mini, GPT-4o, GPT-4.1 OpenAI et al. (2024), Gemini 2.5 Pro Comanici (2025), Claude 4 Sonnet, and Claude 3.5 Haiku. For open-source models, our evaluation covered major model families, model sizes ranging from 1B to 38B, resulting in 33 models: InternVL2.5 (1B–8B) Chen et al. (2025b), InternVL3 (1B–38B) Zhu et al. (2025), InternVL3.5 (1B–38B) Wang et al. (2025), Qwen2-VL (2B–7B) Yang et al. (2024a), Qwen2.5-VL (3B–32B) Qwen et al. (2025), Qwen3-VL (4B-30B) Yang et al. (2025a), Gemma-3 models (4B–27B) Team et al. (2025), LLaVA-interleave Li et al. (2024b), LLaVA-OneVision (7B) Li et al. (2024a), Molmo-7B Deitke et al. (2024), MiniCPM-V-2.6 Yao et al. (2024), Phi-3.5-vision Abdin et al. (2024). We also include physical or spatial domain-specific models, including SpaceQwen2.5-VL Jia et al. (2025b), and three spatial reasoning models: SpaceOm Jia et al. (2025b), Space-Thinker Chen et al. (2024), and Cosmos-Reason1 NVIDIA et al. (2025). We included CoT variants for 3 specialized spatial reasoning models (Cosmos-Reason1 NVIDIA et al. (2025), SpaceOm Jia et al. (2025b), SpaceThinker Chen et al. (2024)) to assess the impact of explicit linguistic reasoning on spatial task performance. Proprietary models were evaluated via official APIs. Open-source models implementation details are in Appendix D.

**Evaluation metrics** We employ three complementary metrics to assess model performance. **Raw accuracy** measures the proportion of correctly answered questions in all evaluated questions. **Cohen's kappa** ($\kappa$) (Cohen, 1960; Coenen, 2014) provides a chance-corrected accuracy measure that accounts for varying option cardinality, enabling fair comparisons across different tasks. To evaluate reasoning stability, we introduce **Pairwise consistency**, which calculates the average of symmetric consistency rates across pairs of questions and their augmentations, measuring whether models produce identical outcomes (both correct or both incorrect) for logically equivalent questions.

## 4.2 RESULTS

**Overall performance** Figure 3 presents the overall performance of 43 VLMs across 23 grouped task variants, organized under 7 spatial reasoning categories, and reveals a clear performance gradient across spatial reasoning categories. Object relation grounding emerges as the easiest category, with many models achieving $\kappa > 0.6$, indicating reliable extraction of basic spatial relations (e.g., left/right, front/behind) from single images. Identity matching displays a bimodal pattern: smaller models perform near chance, while larger models reach near-perfect accuracy, suggesting an emergent scaling ability. Dynamic spatial reasoning, especially tasks involving rotation, shows substantial difficulty. Mental rotation and perspective taking generally yield the near chance overall scores, with most models performing at or below chance, underscoring the absence of robust internal representations for rotational transformations. Rankings of model overall accuracy averaged across tasks and model pair-wise consistency are shown on the left side of Figure 4. The top proprietary model is gpt-5, which ranks first in both overall accuracy and consistency, while the top open source model is InternVL3-38B, which ranks third in overall accuracy and second in consistency. Notably, the leading model in overall accuracy, gpt-5, and the second strongest model, gemini 2.5 pro, also rank first and second on *mental rotation* and achieve the second and third highest performance on *perspective taking*. This links overall success to competence on the most challenging tasks and highlights that models excelling in complex, compositional viewpoint reasoning also perform strongly on simpler diagnostic tasks. More detailed results, including raw accuracy and ungrouped performance, are provided in Appendix B.1, Fig. 32, 33, 34.

**Consistency evaluations** As shown in Figure 4, models exhibit severe inconsistencies in logically equivalent spatial queries, revealing fundamental gaps in spatial reasoning. While top performers

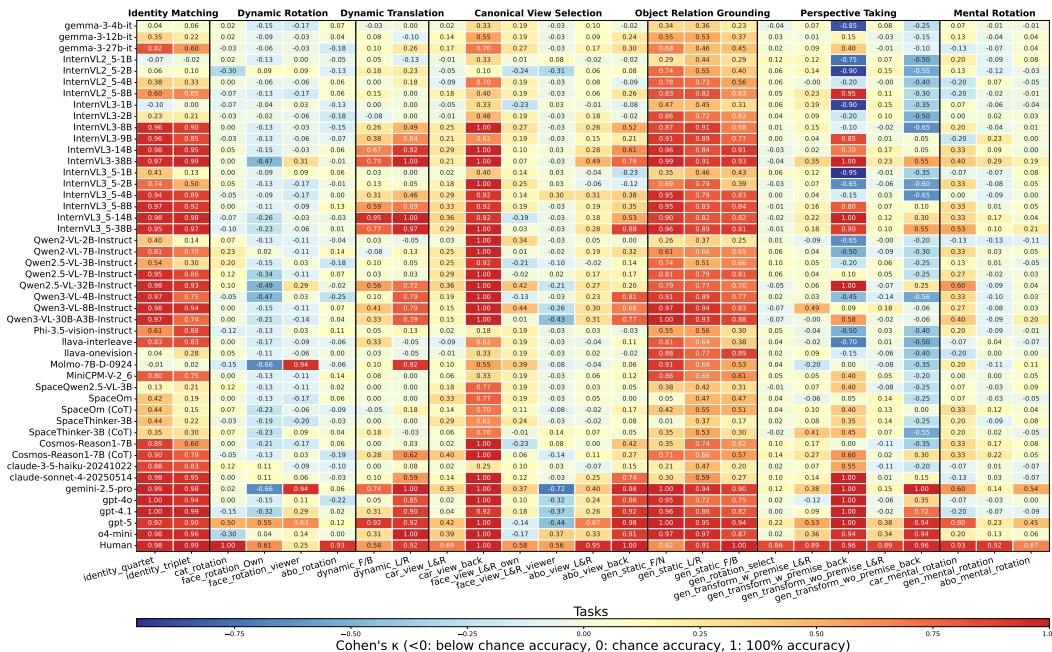

Figure 3: Performance heatmap of 43 VLMs across 26 grouped task variants, organized under 7 spatial reasoning categories. Cohen's kappa values ($\kappa$) measure chance-adjusted performance, where $\kappa = 0$ indicates chance-level and $\kappa = 1$ perfect accuracy. 3 chain-of-thought (CoT) variants of space reasoning models are included for comparison.

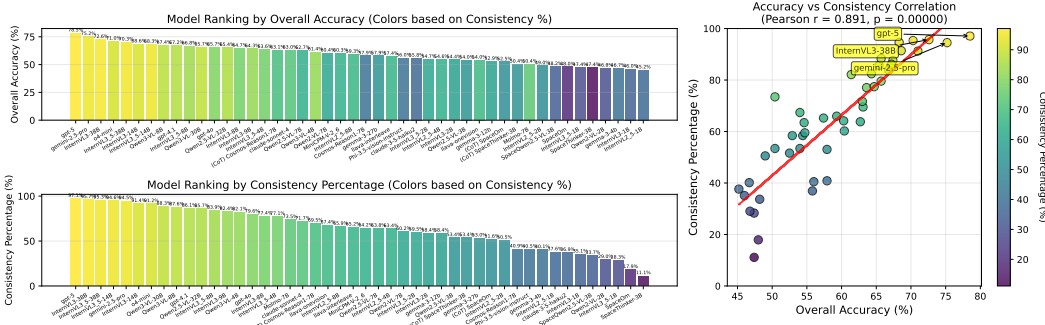

Figure 4: Strong correlation between spatial reasoning accuracy and consistency across vision-language models. Left: Model rankings by overall accuracy (top) and pair-wise consistency percentage (bottom), with colors indicating consistency levels. Right: Scatter plot revealing robust positive correlation (Pearson $r = 0.874, p < 0.05$) between the two metrics.

like gpt-5 achieves 97.1% consistency, most models fail dramatically, with bottom performers below 30% consistency. The strong correlation ($r = 0.891, p < 0.05$) between overall accuracy and consistency suggests these failures stem from incompetent spatial reasoning. Models that cannot maintain "A left of B" equals "B right of A" equivalency lack genuine spatial understanding. Although overall accuracy and consistency strongly correlate, the differences among top models show that consistency alone is not sufficient. gemini 2.5 pro achieves the second highest overall accuracy but only the fifth highest consistency, whereas InternVL3-38B attains the second highest consistency yet ranks third in accuracy, trailing gemini 2.5 pro by $2.6\%$ points. This non-linearity in the upper right corner of Figure 4 demonstrate that high consistency is only the first requirement: once models reach very high levels of consistency, accuracy can still diverge substantially. Strong spatial reasoning therefore requires not only maintaining logical coherence across equivalent queries but also consistently selecting the correct answer. At lower performance levels, better consistency can

Table 2: Performance improvement from CoT reasoning across models and tasks. Delta reflects the change in Cohen's $\kappa$ score. **Bolded** values indicate the task with the greatest improvement per model, and gray-highlighted cells indicate negative performance improvement.

| Task | SpaceOm(3B) | | | SpaceThinker(3B) | | | Cosmos-Reason1-7B | | |
|------|----------|-----|----------|----------|-----|----------|----------|-----|----------|
| | Baseline | CoT | $\Delta$ | Baseline | CoT | $\Delta$ | Baseline | CoT | $\Delta$ |
| Object-relation grounding | 0.332 | 0.493 | +0.162 | 0.185 | 0.393 | +0.208 | 0.569 | 0.649 | +0.080 |
| Identity matching | 0.103 | 0.088 | -0.015 | 0.143 | 0.217 | +0.074 | 0.612 | 0.753 | +0.141 |
| Dynamic | 0.000 | 0.064 | +0.064 | 0.000 | 0.077 | +0.077 | 0.013 | 0.449 | +0.436 |
| Car canonical view selection (back) | 0.775 | 0.700 | -0.075 | 0.625 | 0.700 | +0.075 | 1.000 | 1.000 | +0.000 |
| ABO canonical view selection (back) | 0.000 | 0.167 | +0.167 | 0.076 | 0.045 | -0.030 | 0.424 | 0.273 | -0.152 |
| Perspective-taking (T) w/ premise (back) | 0.050 | 0.400 | **+0.350** | 0.350 | 0.450 | +0.100 | 0.000 | 0.600 | +0.600 |
| Perspective-taking (T) w/o premise (back) | -0.250 | 0.000 | +0.250 | -0.250 | -0.550 | -0.300 | -0.350 | 0.300 | **+0.650** |
| Perspective-taking (T) w/ premise (L&R) | -0.063 | 0.102 | +0.165 | 0.075 | 0.407 | **+0.331** | 0.165 | 0.270 | +0.105 |
| Perspective-taking (T) w/o premise (L&R) | 0.138 | 0.133 | -0.005 | 0.137 | 0.066 | -0.071 | -0.115 | 0.016 | +0.130 |

indicate higher accuracy, but among top-performing systems, consistency alone does not guarantee reliable spatial reasoning. Our findings underscore the difference between stochastic inconsistency and consistent but systematically incorrect behavior. Detailed breakdowns of augmentation strategy analysis, consistency pattern distribution, and comprehensive performance metrics can be found in Appendix B.2.

Table 3: Cohen's kappa ($\kappa$) values for dynamic rotation tasks in the face domain reveal a strong view-centric bias. Models that perform best on the egocentric task (`face_rotation_viewer`) perform worst on the allocentric variant (`face_rotation_own`)

| Model | Allocentric (`face_rotation_own`) | Egocentric (`face_rotation_viewer`) |
|-------|-----------------------------------|-------------------------------------|
| Gemini 2.5 pro | -0.66 (worst) | 0.94 (best) |
| Molmo-7B-D-0924 | -0.66 (worst) | 0.94 (best) |
| InternVL3-38B | -0.47 | 0.31 |
| Qwen2.5-VL-32B-Instruct | -0.49 | 0.29 |

**Biased perspective** Models exhibit a strong bias toward the viewer's perspective in dynamic rotation tasks, even when the question explicitly requires an alternate viewpoint. As shown in Table 3, the top-performing models on the egocentric task are the worst on the allocentric version. This asymmetry suggests an inductive bias toward egocentric interpretation, likely influenced by training data dominated by first-person visual descriptions. Such bias limits the models' ability to generalize across frames of reference and poses challenges for applications like robotics and navigation that require flexible spatial reasoning. Within the GPT family, we observe a trend of improvement. gpt-4o, gpt-4.1, and especially gpt-5 exhibit steadily stronger performance on the egocentric task, with gpt-5 additionally showing marked gains in both egocentirc and allocentric reasoning. Notably, gpt-5 ranks best on the allocentric task ($\kappa = 0.55$, 0.78 accuracy) and second-best on the egocentric task ($\kappa = 0.63$, 0.814 accuracy). While current vision-language models generally default to egocentric interpretations, gpt-5 shows a meaningful step toward reliable reasoning across frames of reference.

**Visual failures or linguistic failures** Perspective-taking (T) tasks test whether models can reason about how object relations transform under viewpoint shifts. In the premise-based variant, all relevant spatial relations are explicitly stated in the prompt, so no visual grounding is required. Yet many models still fail, revealing that errors persist even when the task reduces to purely linguistic reasoning over spatial abstractions. As shown in Figure 5 (a), four models (InternVL3_5-1B, InternVL2_5-2B, InternVL3-1B, gemma-3-4b-it) consistently select the wrong answer with accuracy below 0.1, indicating systematic misinterpretation of reference frames. At the same time, seven models, including gpt-4o, claude-sonnet-4, and several large InternVL variants, achieve near-perfect accuracy (>95%), showing that this reasoning is learnable. Overall, 16 of 39 models (41%) perform below chance, underscoring that even abstracted at the linguistic level, spatial concepts are not robustly encoded or manipulated by most VLMs.

**Does chain-of-thought reasoning help spatial reasoning?** We evaluate the effect of CoT prompting on three models: SpaceOm, SpaceThinker, and Cosmos-Reason1-7B (Table 2). Results show substantial but heterogeneous gains. Cosmos-Reason1-7B benefits most, with an average improvement of +0.221 across tasks and gains in 7 of 9 categories. Its largest boosts occur on perspective-

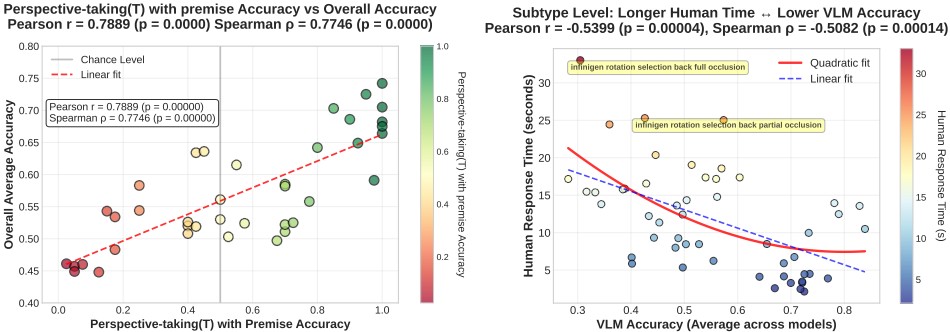

Figure 5: (a) Scatter plot comparing Perspective-taking(T) with premise accuracy against overall accuracy for each model, demonstrating that linguistic spatial reasoning failures are correlated with general model competence. Models are color-coded by Perspective-taking(T) with premise accuracy. (b) Scatter plot showing the relationship between VLM accuracy (x-axis) and human response time (y-axis) across 51 task subtypes.

taking tasks, $+0.650$ and $+0.650$ on perspective-taking with and without premise (back), indicating that CoT is especially effective for spatial transformations requiring explicit reasoning steps. SpaceOm improves moderately ($+0.118$ average), particularly on object-relation grounding ($+0.162$). SpaceThinker shows the weakest effect ($+0.052$ average), including a sharp drop ($-0.300$) on perspective-taking without premise (back). Across all models, object-relation grounding consistently benefits from CoT, while canonical view selection tasks show mixed results. Overall, CoT prompting provides a more significant advantage for complex, multi-step spatial transformations, with larger models demonstrating more improvement. These patterns suggest that CoT alone may not reliably address errors rooted in perceptual or spatial misinterpretation, which aligns with emerging directions Yang et al. (2025d) that perform reasoning directly within the visual embedding space rather than relying solely on textual CoT.

**The Effect of Different Inputs**  To investigate the impact of explicit spatial cues and input image resolution on model performance, we conducted two variations in addition to our standard setting: the *Depth Input* setting and the *High Resolution Input* setting. The *Depth Input* setting provided a depth map, generated by DepthAnything Yang et al. (2024b), which was horizontally concatenated with the original image. This was intended to explicitly supply depth cues to the VLMs without altering the input image number. However, as shown in Figure 6, this modification consistently led to a decrease in overall accuracy across nearly all evaluated models. A possible explanation is that current VLMs are heavily tuned to natural RGB images. Also, simply appending a depth map changes both the distribution and geometry of the input; without any architectural adaptation or fine-tuning, the models may fail to treat depth as a structured cue. In the *High Resolution Input* setting, we use the original images instead of the downsampled version used in the main experiments (max size 256). Despite nearly $10\times$ more pixels, we observe no systematic accuracy gains. This is consistent with the fact that most VLM backbones downsample early in the visual encoder and were pre-trained at relatively modest resolutions. In our evaluations, extra pixels provide limited benefit without corresponding changes in architecture or training.

**Human response time and VLM accuracy correlation**  We further validate that SPINBENCH reflects genuine spatial reasoning difficulty by comparing human and model performance. As shown in Figure 5 (b), task subtypes that required longer human response times also elicited lower VLM accuracy, with a significant negative correlation ($r = -0.54$, $p < 0.05$). This alignment indicates that tasks harder for humans are also systematically harder for models, supporting that SPINBENCH serves as a diagnostic benchmark whose progressively structured tasks reveal core spatial reasoning challenges. More details on the human evaluations setup and results are provided in Appendix C.

**Scaling laws and emergent capability**  Overall performance improves with model scale, but scaling patterns differ sharply across task types (Figure 7). Object relation grounding tasks (e.g., left/right, front/behind) improve smoothly and monotonically across model families. In contrast,

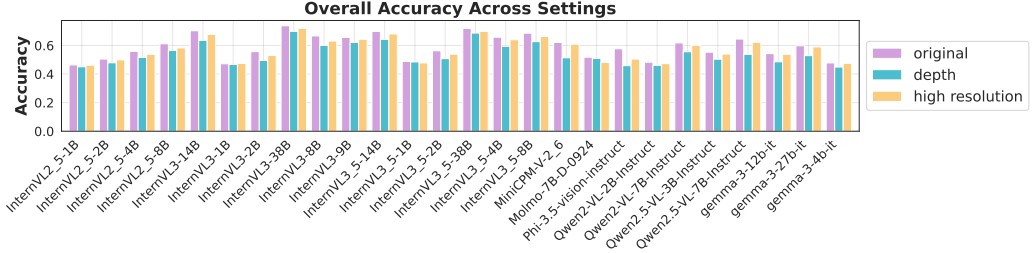

Figure 6: Overall Accuracy Across Settings. We conducted evaluations in 2 additional settings: the *depth* setting, and the *high resolution* setting.

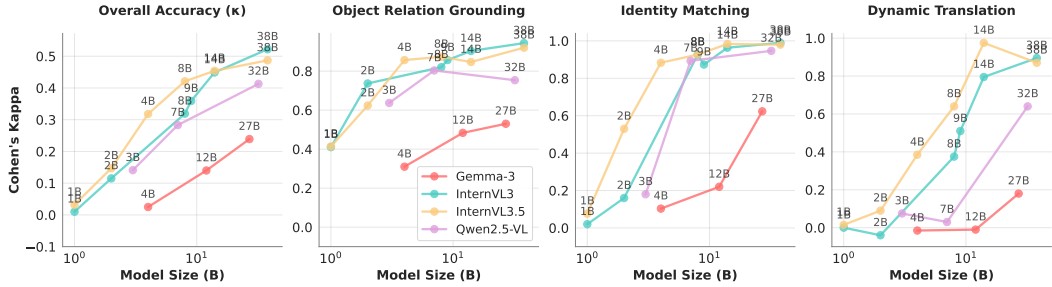

Figure 7: Scaling laws across spatial reasoning tasks. Each line shows Cohen's $\kappa$ (chance-adjusted accuracy) with respect to model size for four model families. While overall performance increases gradually with scale, different task types show distinct scaling patterns.

identity matching exhibits clear *emergence*: smaller models remain at chance, while larger models (7B–8B+) achieve near-perfect accuracy. This non-linear jump suggests that cross-image 3D abstraction only becomes possible once models reach sufficient capacity, consistent with emergent abilities reported in language models Wei et al. (2022). A similar but more gradual emergent trend appears in dynamic translation (e.g., object moving left/right). These distinct scaling behaviors highlight the diagnostic value of our fine-grained benchmark: exposing clear gaps between small and large models and enabling diagnosis of scaling laws in spatial reasoning.

## 5 CONCLUSION AND LIMITATIONS

We present SPINBENCH, a cognitively grounded diagnostic benchmark for evaluating spatial reasoning in vision language models through fine-grained, controlled tasks targeting geometric transformations and viewpoint changes. By decomposing complex perspective taking into interpretable subskills, SPINBENCH facilitates precise diagnosis of model limitations. Our evaluation of 37 VLMs reveals systematic weaknesses, including consistent reference-frame bias, failures in rotation understanding, and linguistic spatial inference, alongside diverse scaling behaviors and emergent capabilities. These findings suggest that different aspect of spatial reasoning are not uniformly learned and often remains underdeveloped even in advanced models. Human evaluation further validates the benchmark, showing a strong correlation between human response times and VLM accuracy, suggesting that SPINBENCH captures genuine cognitive difficulty shared across humans and models. SPINBENCH goes beyond scorekeeping by providing a diagnostic lens on spatial competencies, offering conceptual clarity about what aspects of spatial reasoning VLMs do and do not master, and guiding the development of multimodal foundation models. These diagnostic insights are directly actionable for embodied AI, where failures in reference-frame reasoning or rotation understanding can lead to breakdowns in navigation, manipulation, and other safety-critical tasks. A key limitation is that we do not yet cover other important spatial concepts such as containment, support, or vertical relations (e.g., "in," "on," "above").

**Ethics Statement**   This work includes human evaluations conducted to measure benchmark difficulty. All participants were adults who gave informed consent, and their data were collected and analyzed anonymously. The study followed institutional ethics guidelines and posed no foreseeable risks to participants. Beyond this, our research uses only public or synthetic datasets under appropriate licenses. While failures in spatial reasoning can have implications for safety-critical systems, SpinBench is intended solely as a diagnostic tool to improve transparency and safety in model development. We affirm full adherence to the ICLR Code of Ethics throughout this work.

**Reproducibility Statement**   We have taken steps to ensure that our work can be reproduced. The design of SpinBench, including task categories, dataset composition, and controlled variations, is described in detail in Section 3 and Appendix A. Experimental settings, model lists, and evaluation metrics are provided in Section 4 and Appendix D, along with additional results in Appendix B and C. All datasets used are either publicly available or generated using documented pipelines, and details of sampling and annotation are included in the appendix. To further support reproducibility, we provide an anonymous project website with benchmark resources and plan to release code and data generation scripts in the near future.

ACKNOWLEDGMENTS

This work was fully supported by Mitsubishi Electric Research Labs (MERL).

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

TABLE OF APPENDIX CONTENTS

## A  SPINBENCH

### A.1  DETAILED DATASET COLLECTION PROCESS

**A.1.1  Simulation**  We adopt a synthetic dataset generation pipeline that integrates Infinigen-generated indoor environments Raistrick et al. (2024) with the Isaac Sim simulator NVIDIA. The pipeline is fully automated through a custom script built on top of the Infinigen SDG (synthetic data generation) framework. The process can be summarized as follows:

1. **Environment loading.** A set of nine indoor dining-room scenes are retrieved from the Infinigen asset library. Each scene is instantiated as a USD stage, with ceiling meshes optionally hidden for improved lighting and camera coverage. Colliders are added to all major surfaces (walls, floors, dining table) to enable realistic object–surface interactions.

2. **Object assets.** Everyday objects are imported from the Yale-CMU-Berkeley (YCB) dataset Calli et al. (2015; 2017). We include 21 distinct items (e.g., banana, soup can, mug, Rubik's cube), each automatically labeled by parsing their USD asset names. Gravity and rigid-body dynamics are attached using PhysX APIs to support physically plausible placement and falling behavior. Additional assets can be manually labeled with explicit semantic tags.

3. **Scene composition.** For each scene, objects are sampled and placed in the working area above the dining table. Object poses are randomized within bounded 3D ranges (position, orientation, scale). Distractor meshes and primitive shapes are also injected.

4. **Lighting.** Three movable sphere lights are added per scene and randomized in location, intensity (500–2500 lumens), and color balance. Dome lights with HDR textures are randomized per capture to simulate natural variations in sky illumination (clear, cloudy, evening, night).

5. **Cameras.** Multiple cameras (default: five per scene) are defined, with randomized intrinsics and extrinsics. We support both (i) random camera placements on a viewing sphere around a target object, and (ii) structured camera orbits with fixed angular increments to capture viewpoint changes.

6. **Physics simulation.** The scene is stepped forward for several frames to resolve collisions and allow objects to settle into stable configurations. Captures are taken both after this settling, producing "dropped" views with objects resting on the table.

7. **Data capture.** Render products are generated at $480 \times 480$ resolution using the RTX Path Tracing renderer. For each environment and camera, both RGB images and corresponding semantic pose annotations are written to disk through Isaac Replicator writers. On average, we capture 100 frames per environment (500 frames total per scene when multiplied across cameras).

In addition to randomized placement, we explicitly manipulate object positions to generate controlled spatial displacements. Using custom utility functions, each object is sequentially shifted relative to the initial position:

- **Left/Right.** Objects are translated along the $x$-axis by fixed increments (e.g., `move_left(distance=0.1)` and `move_right(distance=0.2)`). This simulates lateral displacements in the viewer's frame of reference.

- **Near/Far.** Objects are shifted along the $y$-axis (`move_near(distance=0.1)`, `move_far(distance=0.2)`), simulating depth changes toward or away from the camera viewpoint.

This procedure yields a diverse and physically consistent dataset covering static spatial relations, translational dynamics, and multi-view perspective taking (with and without occlusion). The modular design of the script enables controlled variation in object placement, illumination, and camera trajectories, while preserving reproducibility through fixed random seeds.

**A.1.2  Real-world dataset curation.**  To unify diverse real-world sources under a common spatial reasoning framework, we implemented a multi-dataset curator that standardizes input formats,

view sampling, and question generation. Each dataset is wrapped in a dedicated handler class that exposes object discovery, available views, and sample generation routines. The curation pipeline proceeds as follows:

- **Object discovery.** For each dataset, we enumerate object folders (ABO product IDs, car object IDs, and face subject IDs). Only objects with complete view coverage are retained (e.g., 72 views in ABO, consistent rotation sequences in Cars, and multiple head poses in Faces). This ensures all curated objects can support viewpoint-based reasoning tasks.

- **View normalization.** Views are mapped to standardized angular indices. For ABO, we map 72 canonical views to $0°$–$355°$ in $5°$ steps. For Cars and Faces, we parse angles and normalize them to $0°$–$359°$. This allows cross-dataset comparison of viewpoint-sensitive tasks.

- **Task generation.** Each dataset supports three primary families of tasks:

  1. *Object identity.* Odd-one-out tasks (triplets or quartets) where two or three views depict the same object/person and one depicts a distractor.

  2. *Rotation classification.* Pairwise comparisons where an object rotates by a known offset (e.g., $45°$, $90°$), and the model must classify the rotation direction (clockwise/counterclockwise). For Faces, we explicitly test both *viewer-centric* and *object-centric* frames of reference.

  3. *Canonical view selection.* Given a front view, models must identify left, right, or back profiles from among candidate images. This directly probes viewpoint reasoning and perspective-taking.

- **Mental rotation (ABO only).** Leveraging ABO's dense 72-view coverage, we generate multiple-choice mental rotation tasks where the model must predict the outcome of rotating an object by $45°$–$180°$ in either direction. Distractors are sampled to ensure a minimum angular separation, preventing trivial cues.

- **Splitting and statistics.** After sample generation, the curator splits data into train/validation/test sets with dataset-specific ratios (e.g., ABO: 80/10/10; Faces: 70/10/20; Cars: test-only). Statistics such as the number of objects, samples per task type, and split sizes are logged for reproducibility.

- **Query variation.** To avoid linguistic bias and encourage genuine spatial reasoning, each task type is associated with multiple natural language templates. For example, an odd-one-out task may be phrased as "Which of these three images shows a different object?" or alternatively as "Two of these images show the same object at different views, which one is different?" During dataset generation, a random template is selected from the available pool for each sample, ensuring linguistic diversity across training and evaluation.

- **Answer option randomization.** In addition to varying the textual query, we randomize the ordering of candidate options (A/B/C or A/B/C/D). For odd-one-out tasks, the distractor image can appear in any position; for rotation classification, the labels "clockwise" and "counterclockwise" are shuffled; and for canonical view selection, left/right/back views are permuted across options. This randomization ensures that models cannot exploit positional biases (e.g., always guessing option C) and must instead rely on actual spatial reasoning to succeed.

This unified curation procedure ensures that disparate real-world datasets contribute consistently formatted, balanced tasks, enabling controlled evaluation of spatial reasoning across product-scale objects (ABO), structured geometric entities (Cars), and biologically stimuli (Faces).

## A.2  DATA ANNOTATION PROTOCOL

**A.2.1  General Guidelines**  All annotations are designed to probe spatial reasoning while minimizing confounds. We adopt the following principles: (i) all questions must be unambiguous under a specified frame of reference, (ii) tasks must balance object categories and viewpoints, and (iii) phrasing diversity is required to prevent overfitting to a single query template.

**A.2.2  Data Format and Structure**  Each annotated instance is serialized as JSON with four fields: `problem` (natural language question), `answer` (ground truth label, always a single capital letter), `images` (paths to associated views), and `metadata` (structured fields such as object IDs, view indices, occlusion condition, task type). This format ensures compatibility with VQA pipelines while retaining rich metadata for controlled analysis. All datasets are organized by dataset type (ABO, Cars, Faces, Infinigen), and further by task subtype.

**A.2.3  Quality Control and Validation**  We employ both automated and manual checks: for Infinigen, annotation scripts display candidate images to the curator, who confirms correctness with keystrokes (e.g., pressing "y" to validate a generated left/right relation). For real-world datasets, handlers enforce strict view coverage (72 views for ABO, complete rotation for Cars, multi-pose coverage for Faces). Random seeds are fixed during sampling for reproducibility.

**A.2.4  Handling Ambiguities**  To ensure tasks probe genuine spatial reasoning rather than noise, we implement explicit constraints to minimize annotation ambiguities:

- **Angular separation.** In ABO mental rotation tasks, distractor views are required to differ by at least $30°$ from the target orientation. This prevents trivial confounds where two options appear nearly identical. Car and Face rotation classification restricts rotations to canonical offsets ($45°$, $90°$, $180°$) for clearer discriminability.
- **Visibility filtering.** In Infinigen, only objects with projected visibility above $0.8$ are considered valid. Scenes where occlusion prevents reliable labeling are discarded. For occlusion tasks, annotators explicitly tag each scene as no, partial, or full occlusion.
- **Positional thresholds.** Static left/right judgments are computed from object cuboid centers projected in image space. Objects are required to have distinct $x$-coordinates to avoid ambiguous ties. Near/far relations are based on $y$-coordinates, requiring a minimal vertical separation. In dynamic relation tasks, movement distances are set to non-trivial shifts ($0.2$ scene units) to guarantee perceptibility.
- **Symmetry control.** Centrally symmetric objects (e.g., square stool) are excluded from ABO to avoid cases where left/right or rotation cannot be distinguished visually.
- **Frame-of-reference disambiguation.** For face rotation, tasks are duplicated under both object-centric ("the person turned their own head left") and viewer-centric ("the person turned to the viewer's right") frames.

These constraints, enforced both in code and manual filtering, ensure that all retained samples are unambiguous and diagnostic of the intended spatial relation.

## A.3  TASK CATEGORIES AND SUBTYPES

We provide a comprehensive breakdown of the dataset constitution across major task groups, their subtypes, and the configuration details for each subtype. Table 4 summarizes the complete distribution across all 51 distinct task subtypes.

Table 4: Full task subtype breakdown with configuration details.

| Group | Subtype | #Queries | #Images | #Options |
|-------|---------|----------|---------|----------|
| | car_identity | 80 | 0+3 | 3 |
| | car_identity_quartet_imagefirst | 9 | 0+4 | 4 |
| | car_identity_quartet_interleaved | 5 | 0+4 | 4 |
| | car_identity_quartet_textfirst | 6 | 0+4 | 4 |
| | face_identity | 79 | 0+3 | 3 |
| | face_identity_quartet_imagefirst | 7 | 0+4 | 4 |
| identity | face_identity_quartet_interleaved | 8 | 0+4 | 4 |
| matching | face_identity_quartet_textfirst | 4 | 0+4 | 4 |
| | object_identity_imagefirst | 33 | 0+3 | 3 |
| | object_identity_interleaved | 35 | 0+3 | 3 |

| Group | Subtype | #Queries | #Images | #Options |
|---|---|---|---|---|
| | object_identity_quartet_imagefirst | 42 | 0+4 | 4 |
| | object_identity_quartet_interleaved | 38 | 0+4 | 4 |
| | object_identity_quartet_textfirst | 22 | 0+4 | 4 |
| | object_identity_textfirst | 37 | 0+3 | 3 |
| object-relation grounding | infinigen_spatial_relation_grounding_far_near | 152 | 1+0 | 2 |
| | infinigen_spatial_relation_grounding_left_right | 286 | 1+0 | 2 |
| | infinigen_spatial_relationship_front_behind | 198 | 1+0 | 2 |
| dynamic rotation | car_rotation_classification | 80 | 2+0 | 2 |
| | face_rotation_classification_own_perspective | 94 | 2+0 | 2 |
| | face_rotation_classification_viewer_perspective | 70 | 2+0 | 2 |
| | object_rotation_classification_imagefirst | 35 | 2+0 | 2 |
| | object_rotation_classification_interleaved | 47 | 2+0 | 2 |
| | object_rotation_classification_textfirst | 27 | 2+0 | 2 |
| dynamic translation | infinigen_spatial_relationship_dynamic_front_back | 78 | 2+0 | 2 |
| | infinigen_spatial_relationship_dynamic_left_right | 78 | 2+0 | 2 |
| canonical view selection | car_canonical_view_selection_back | 19 | 1+3 | 3 |
| | car_canonical_view_selection_left | 19 | 1+3 | 3 |
| | car_canonical_view_selection_right | 20 | 1+3 | 3 |
| | face_canonical_view_selection_own_perspective_left | 23 | 1+2 | 2 |
| | face_canonical_view_selection_own_perspective_right | 19 | 1+2 | 2 |
| | face_canonical_view_selection_viewer_perspective_left | 17 | 1+2 | 2 |
| | face_canonical_view_selection_viewer_perspective_right | 18 | 1+2 | 2 |
| | object_canonical_view_selection_back | 80 | 1+3 | 3 |
| | object_canonical_view_selection_left | 86 | 1+3 | 3 |
| | object_canonical_view_selection_right | 57 | 1+3 | 3 |
| perspective taking | infinigen_rotation_selection_back_full_occlusion | 9 | 1+3 | 3 |
| | infinigen_rotation_selection_back_no_occlusion | 49 | 1+3 | 3 |
| | infinigen_rotation_selection_back_partial_occlusion | 47 | 1+3 | 3 |
| | infinigen_rotation_selection_left_full_occlusion | 5 | 1+3 | 3 |
| | infinigen_rotation_selection_left_no_occlusion | 62 | 1+3 | 3 |
| | infinigen_rotation_selection_left_partial_occlusion | 43 | 1+3 | 3 |
| | infinigen_rotation_selection_right_full_occlusion | 7 | 1+3 | 3 |
| | infinigen_rotation_selection_right_no_occlusion | 61 | 1+3 | 3 |
| | infinigen_rotation_selection_right_partial_occlusion | 40 | 1+3 | 3 |
| | infinigen_spatial_relation_transformation_w_premise_back | 33 | 1+0 | 2 |
| | infinigen_spatial_relation_transformation_w_premise_left | 58 | 1+0 | 2 |
| | infinigen_spatial_relation_transformation_w_premise_right | 53 | 1+0 | 2 |
| | infinigen_spatial_relation_transformation_wo_premise_back | 36 | 1+0 | 2 |
| | infinigen_spatial_relation_transformation_wo_premise_left | 58 | 1+0 | 2 |
| | infinigen_spatial_relation_transformation_wo_premise_right | 52 | 1+0 | 2 |
| mental rotation | object_mental_rotation | 78 | 1+4 | 4 |
| | infinigen_mental_rotation | 120 | 1+4 | 4 |
| | car_mental_rotation | 20 | 1+4 | 4 |

**Task Group Distribution.** The dataset contains a total of 2739 samples spanning seven major task groups with varying emphasis: Object-Relation Grounding tasks represent the largest category with 636 samples (23.2%), followed closely by Perspective Taking with 613 samples (22.4%). Identity Matching contributes 405 samples (14.8%), while Canonical View Selection and Dynamic Rotation each account for approximately 13–14% of the dataset (358 and 353 samples respectively). The smaller categories include Dynamic Translation with 156 samples (5.7%) and Mental Rotation with 218 samples (8.0%).

**Dataset Source Distribution.** Four distinct data sources contribute to the benchmark: Infinigen provides the majority with 1,525 samples (55.7%), followed by ABO Objects with 617 samples

(22.5%), Faces with 339 samples (12.4%), and Cars with 258 samples (9.4%). Notably, Infinigen exclusively covers Object-Relation Grounding, Perspective Taking, and Dynamic Translation tasks, while the other domains span Identity Matching, Canonical View Selection, and Dynamic Rotation tasks.

**Task Configuration Details.** The image structure varies systematically across task types, decomposed into reference images and candidate option images. Single reference image tasks (1+0 to 1+4 format) constitute the majority, including spatial relation tasks with text-only options (1+0), canonical view selection with 2–3 image options (1+2, 1+3), and mental rotation with 4 image options (1+4). Two-reference image tasks (2+0 format, 509 samples, 19.6%) appear exclusively in rotation classification and dynamic relationship tasks with text-only options. Identity matching tasks uniquely employ a no-reference format (0+3, 0+4), where all 3–4 images serve as candidate options for comparison.

The relationship between option images and answer choices follows a consistent pattern: when the image option count is 0, the task employs text-only multiple choice answers; otherwise, the number of image options directly corresponds to the number of answer choices.

**Answer Choice Distribution.** The benchmark employs a balanced choice structure: binary choices (A/B) represent 39.9% of tasks (1,094 samples), primarily in rotation classification and spatial transformation tasks. Ternary choices (A/B/C) account for 53.4% (1,463 samples), covering canonical view selection and most identity matching tasks. Four-way choices (A/B/C/D) only appears in quartet identity matching and mental rotation tasks. The answer distribution across options shows a reasonable balance: option A appears in 41.2% of cases (1,130 samples), option B in 41.2% (1,131 samples), option C in 14.3% (391 samples), and option D in 3.1% (87 samples).

## A.4 DETAILED TASK DESCRIPTION WITH EXAMPLES

### A.4.1 Identity Matching
The identity matching tasks evaluate a model's ability to recognize whether multiple images depict the same object, person, or vehicle under viewpoint variation. This capability serves as a foundational prerequisite for more complex spatial reasoning, since robust object identity recognition must occur before reasoning about spatial transformations. Identity matching tasks are presented across three domains—cars, faces, and generic objects—with further subdivisions based on presentation format (triplet vs. quartet, image-first vs. text-first vs. interleaved). Quartet setting compared to triplet setting tests whether one more image of the same object increases difficulty by presenting more tokens or decreases difficulty by presenting more views of the same object.

- **Car identity matching**(Fig. 8): The model must decide which image shows a different car, given triplets or quartets of cars photographed from different angles. Subtypes differ by whether the distractor is presented among three images, or within a quartet with either images first, text first, or an interleaved format.

- **Face identity matching**(Fig. 9): Analogous to the car tasks, but using human faces under pose variation. The distractor is a different individual, while the other images depict the same person from different viewpoints. This directly probes human face recognition under multi-view conditions.

- **Object identity matching** (Fig. 10 and Fig. 11): For the triplet form, the model receives three images, two of which depict the same object under viewpoint change, while one shows a different object. Subtypes vary by whether images are shown first, interleaved with text, or after text. Quartet form is a variation where the model must select the odd one out from four candidate images, again with differences in presentation format. This setting tests whether one more image of the same object increases difficulty by presenting more tokens or decreases difficulty by presenting more views of the same object.

### A.4.2 Dynamic Rotation
The dynamic rotation tasks evaluate whether models can track the orientation changes of a single object across sequential frames. Unlike static relation tasks, these examples isolate rotational transformations with a static camera and a constant background, thereby requiring models to reason about in-place turning rather than translation.

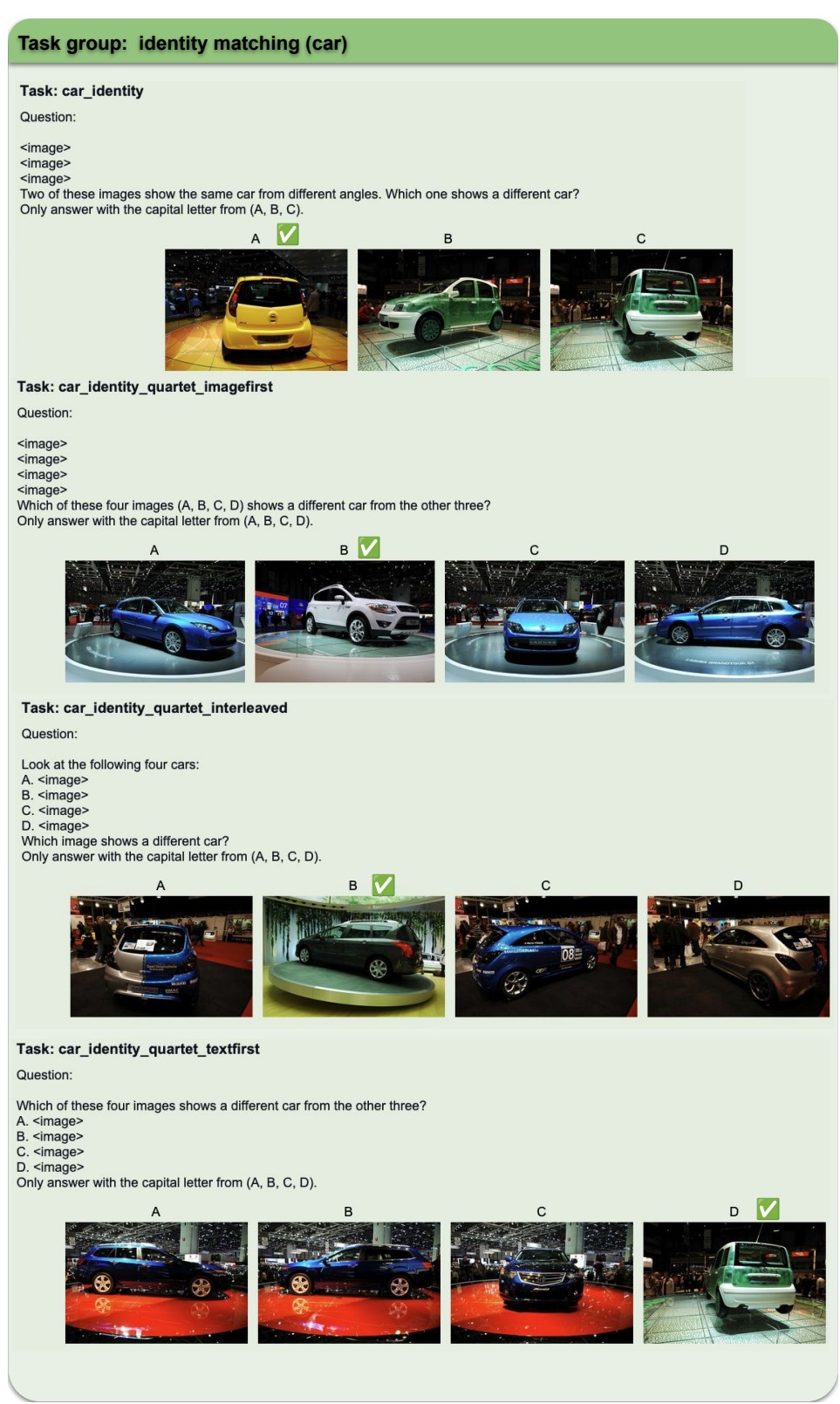

Figure 8: Examples of car identity matching tasks. Models must detect the odd car out across triplets and quartets, with different presentation styles (image-first, interleaved, text-first).

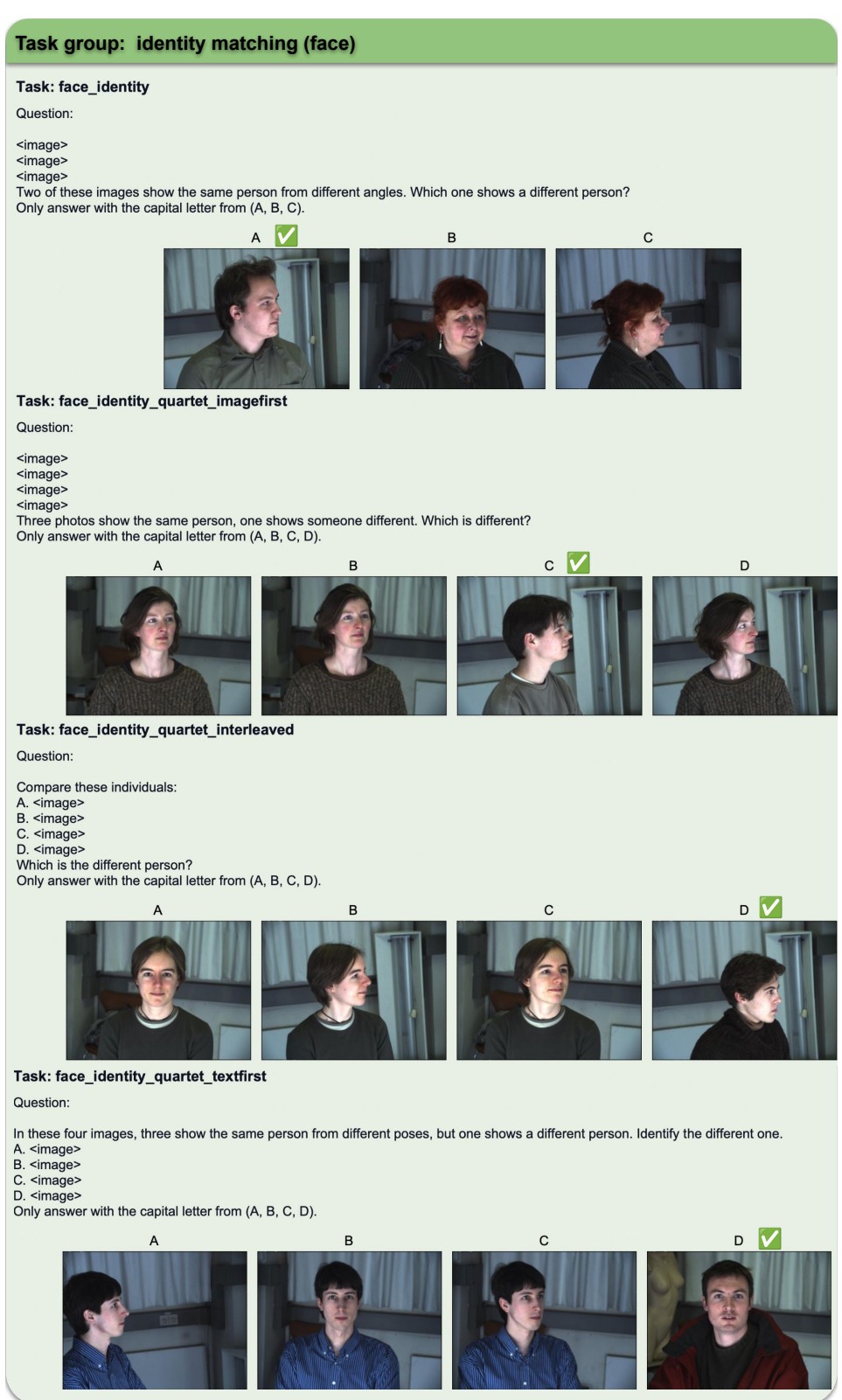

Figure 9: Examples of face identity matching tasks. The model must identify which image depicts a different individual, under both triplet and quartet setups, with varied presentation orders.

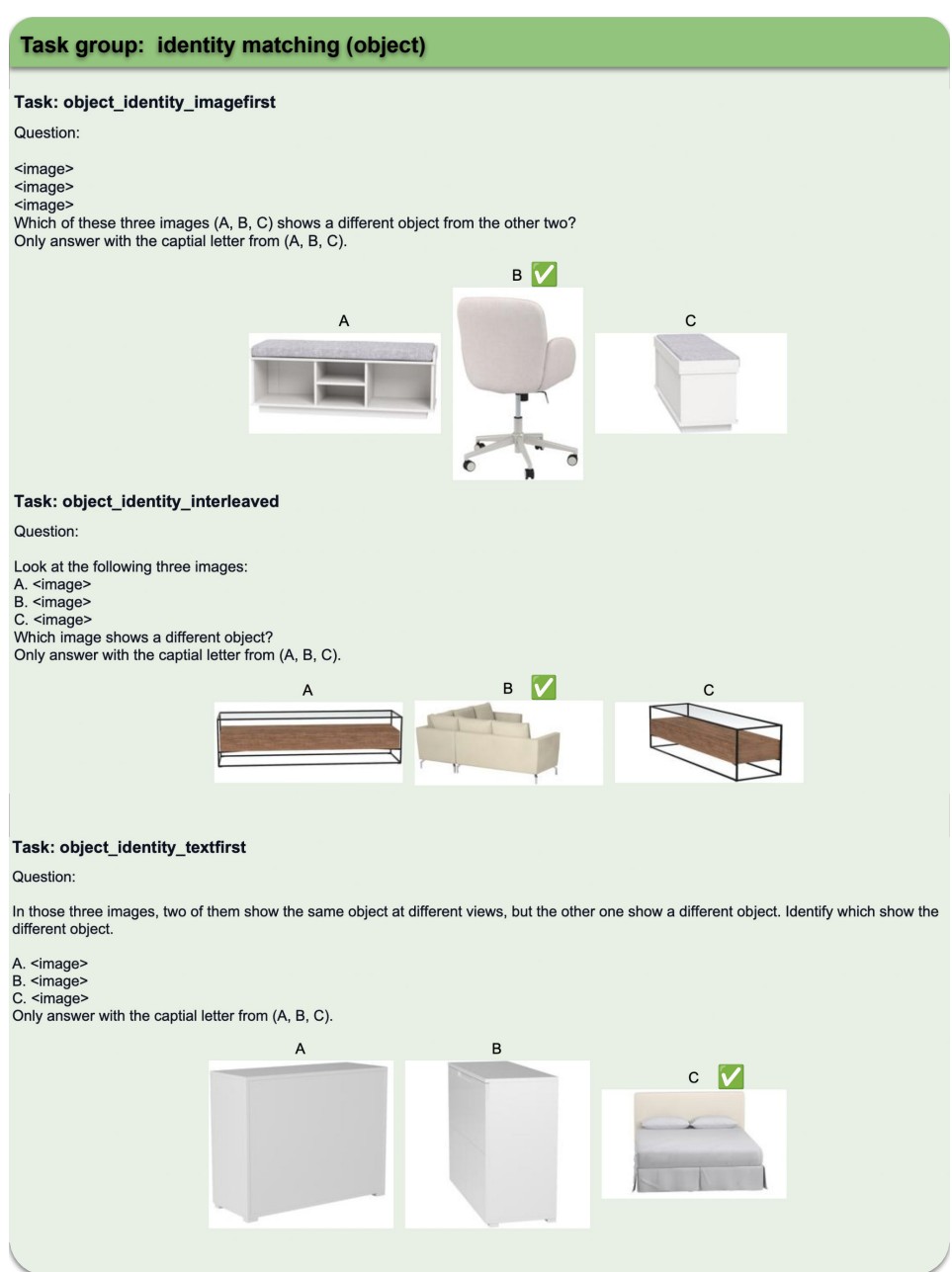

Figure 10: Examples of object identity matching with triplets. Each row contains three candidate images; two show the same object under view change, and one shows a different object.

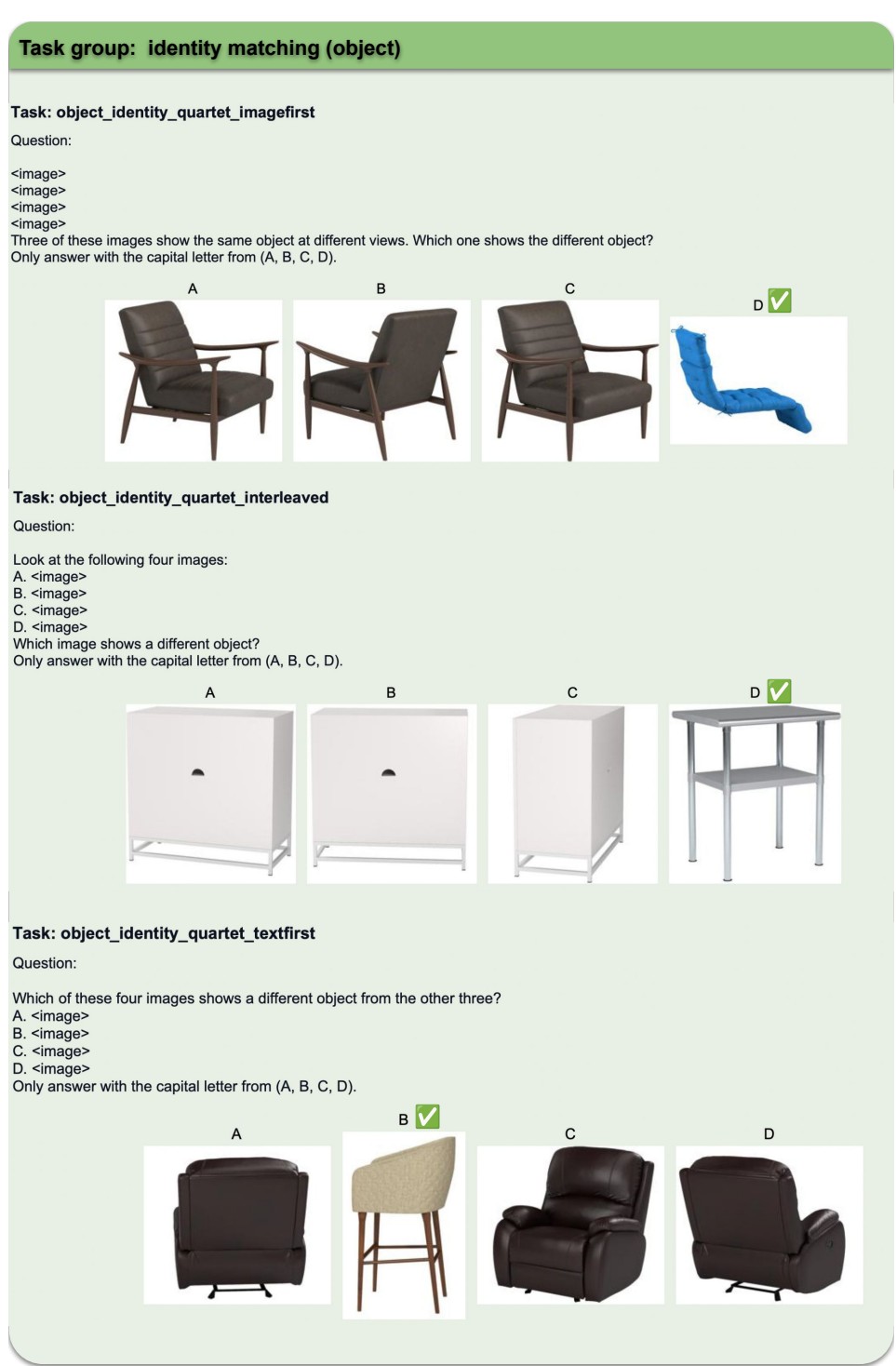

Figure 11: Examples of object identity matching with quartets. Models must identify the one image depicting a different object, with task variants controlling text–image ordering.

- **Car rotation classification**(Fig. 12): The model sees two sequential views of a car rotating in place. It must decide whether the rotation was clockwise or counterclockwise, with reference to a top-down view.

- **Face rotation classification** (own perspective vs. viewer perspective) (Fig. 13): These subtypes probe perspective-dependent interpretation. From the human in the image's own perspective, "left" and "right" correspond to their intrinsic body-centered frame. From the viewer's perspective, left/right must be relative to the camera's position or image frame.

- **Object rotation classification**(Fig. 14): Similar to cars, but applied to generic objects (e.g., furniture). Variants differ in presentation order (image-first, text-first, interleaved).

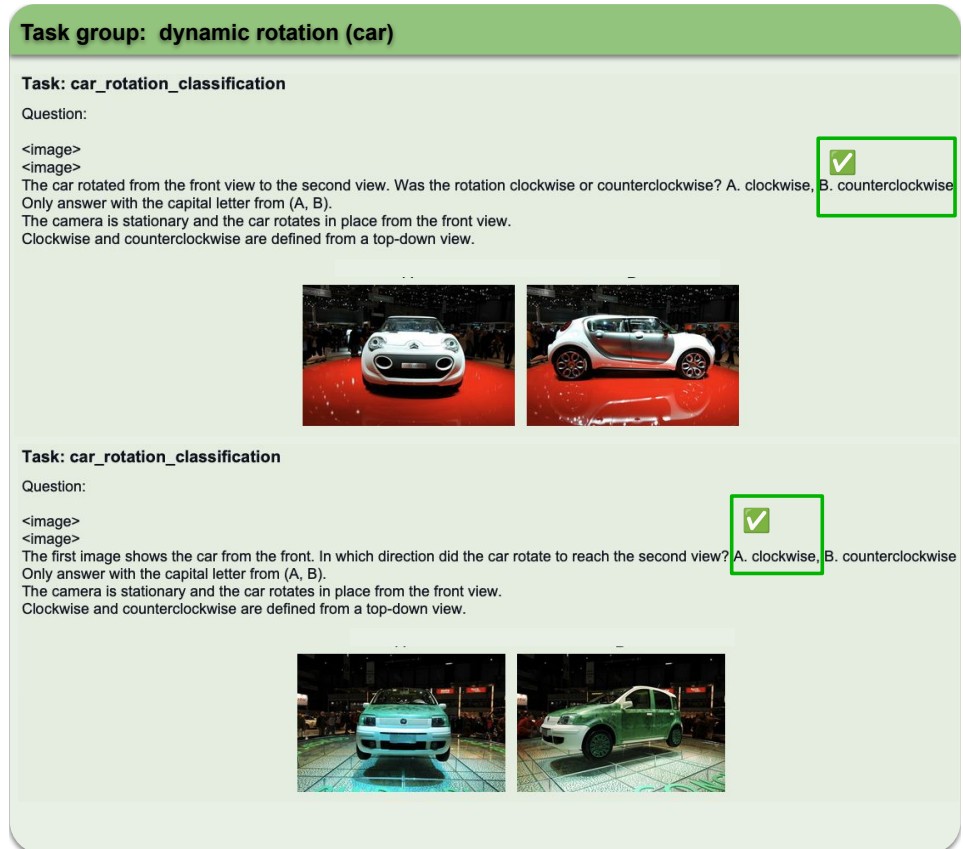

Figure 12: Examples of dynamic rotation (car) tasks. The car is shown rotating in place across two images, and the model must determine whether the transformation corresponds to a clockwise or counterclockwise rotation.

**A.4.3 Dynamic Translation** The dynamic translation tasks evaluate whether models can detect and interpret translational movements of objects across sequential frames. Unlike rotation classification, the focus here is on linear displacement within the viewer's frame of reference while the background and camera remain static. These tasks isolate directional movement (front/back or left/right) from rotational or other spatial transformations.

- **Front–back translation** (Fig. 15): The model observes two frames showing an object (e.g., box, canned food) shifted either forward or backward relative to the static camera. It must classify the displacement as "front" or "back."

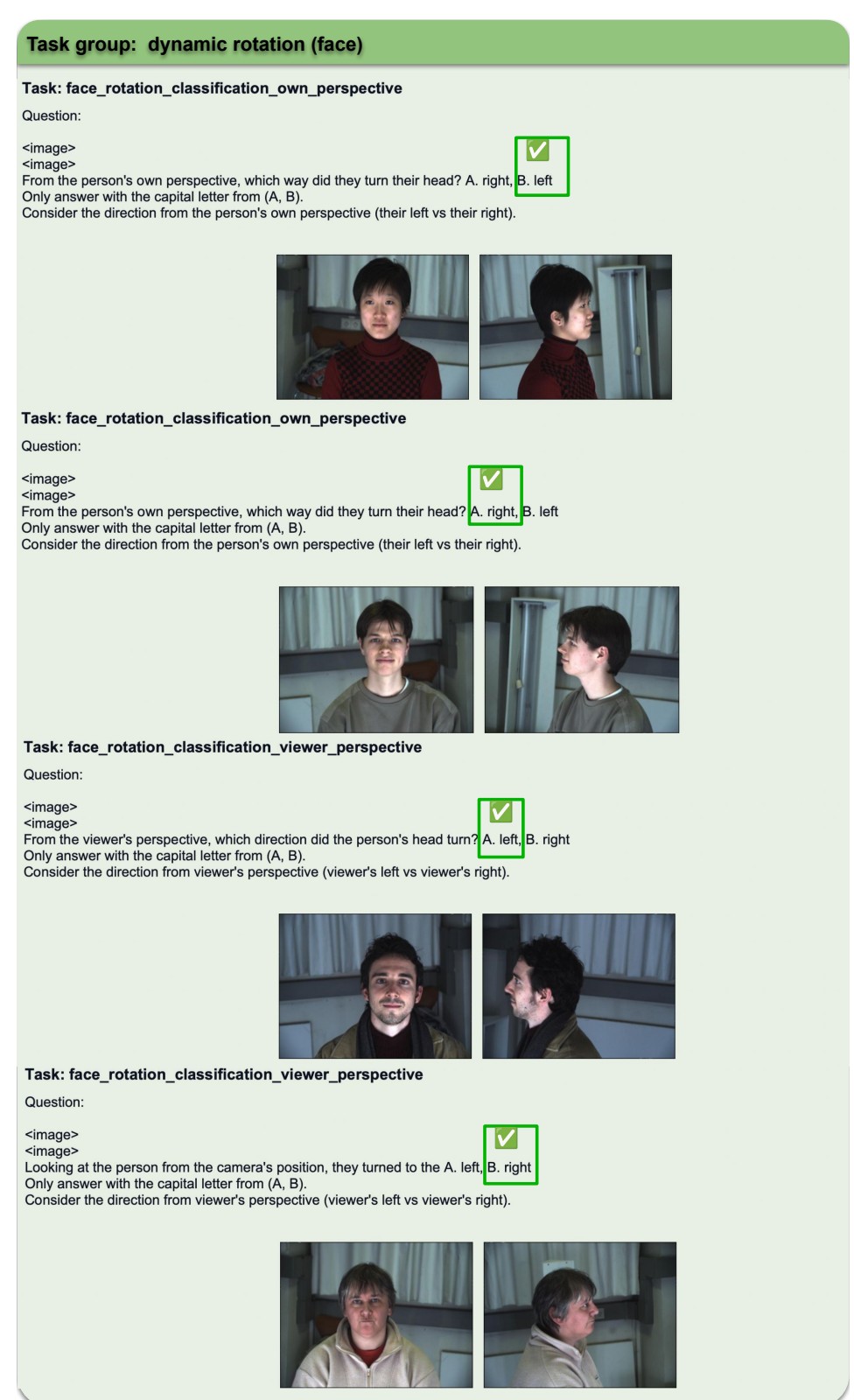

Figure 13: Examples of dynamic rotation (face) tasks. The model must classify the direction of a person's head turn, either from their own perspective (intrinsic left/right) or from the viewer's perspective (extrinsic left/right).

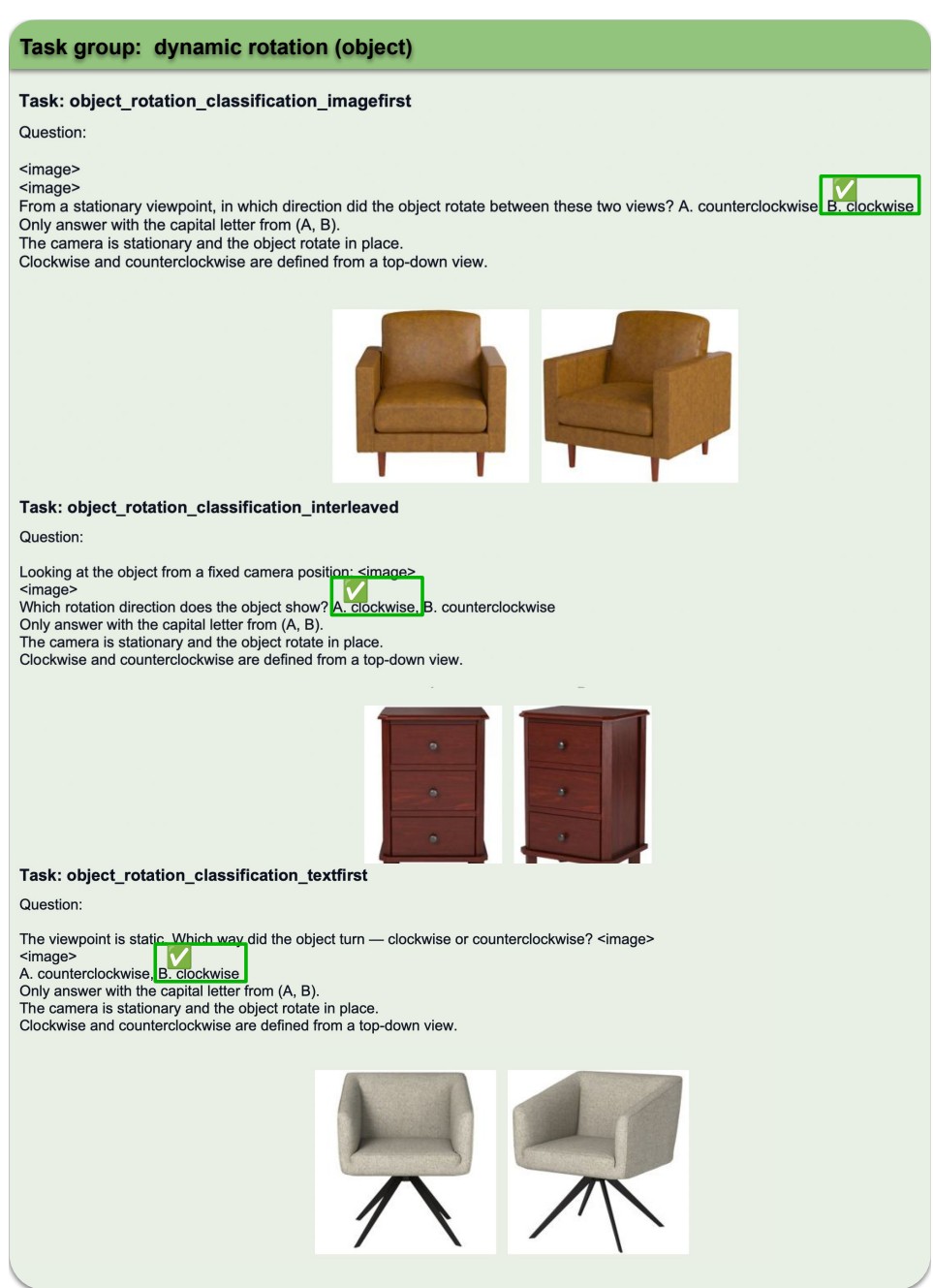

Figure 14: Examples of dynamic rotation (object) tasks. Object items are shown before and after rotation, and the model must classify the direction of turn. Subtypes vary by whether the question is posed text-first, image-first, or interleaved.

- **Left–right translation** (Fig. 15): The model observes an object (e.g., scissors, bottle) moving laterally within the scene. It must determine whether the movement occurred toward the left or the right, again from the static camera's viewpoint.

**A.4.4 Object-Relation Grounding** Object-relation grounding tasks assess a model's ability to infer object-relative spatial configurations from a single image. Each task involves two target objects within the same frame, and models must judge directional relations (e.g., left/right, in front of/behind) or distance-based relations (e.g., near/far). These tasks capture object-relative pose understanding in static scenes. Unlike dynamic or multi-view reasoning tasks, these examples isolate spatial grounding from both temporal reasoning and perspective transformations, serving as a controlled evaluation of whether models can interpret scene-centric spatial layouts from a static visual input. A key difficulty in these tasks is that the model must correctly **identify the correct objects of interest** among possibly multiple distractor objects in the scene. This makes the setup closer to an open-world detection problem: even if a model has a strong spatial reasoning ability, focusing on the wrong object will lead to incorrect answers.

For spatial relation tasks with inherent symmetries, we systematically generate equivalent reformulations through two complementary augmentation strategies to test reasoning consistency: ***Symmetrical Augmentation***: We create logically equivalent variants by swapping spatial relationships and flipping correct answers. For example, from a base query "Which object is on the left? A or B," we generate the symmetrical variant "Which object is on the right? A or B," with the corresponding answer flipped. This transformation preserves the underlying spatial configuration while testing whether models maintain consistent spatial reasoning across equivalent logical formulations. ***Syntactic Augmentation***: We reformulate question structures while preserving semantic content, such as transforming "Which object is on the left? A or B," into "Is A on the left or right of B? A. left B. right." These variations test whether models rely on specific linguistic patterns or demonstrate robust spatial understanding independent of question phrasing.

- **Left/Right Relations with Augmentations** (Fig. 16): The base task asks whether one object (e.g., Rubik's cube) is to the left or right of another (e.g., mustard bottle). Symmetrical augmentation flips the query to its logical equivalent ("Is the mustard bottle to the left or right of the Rubik's cube?"), while syntactic augmentation reformulates the phrasing into binary comparisons ("Which object is on the left?" vs. "Is A on the left or right of B?"). Together, these variations test whether models preserve consistent reasoning across symmetry and linguistic surface changes.
- **Near/Far Relations with Symmetry** (Fig. 17): The base task asks which of two objects (e.g., marker vs. foam brick) is closer to the viewer. Symmetrical augmentation flips the distance relation by instead asking which object is farther.
- **Front/Behind Relations with Augmentations** (Fig. 18): The base task asks which object is in front of the other (e.g., mug vs. Rubik's cube) from a front-view image. Symmetrical augmentation reverses the relation ("Which object is in the back?"), and syntactic augmentation reformulates the query into pairwise comparisons ("Is the mug in front of or behind the Rubik's cube?"). These augmentations jointly probe whether models generalize depth-order reasoning across logically equivalent but differently phrased prompts.

**A.4.5 Canonical View Selection** The canonical view selection tasks test whether models can correctly identify specified viewpoints of objects, cars, or faces. Unlike dynamic tasks, the images are presented as static alternatives, and the challenge lies in transforming the front-view reference into another canonical perspective (left, right, or back). These tasks isolate perspective transformation without involving temporal dynamics or multi-object relationships.

- **Car canonical view selection** (Fig. 19): Given a front-view reference image, the model must identify which candidate view corresponds to the car viewing from left, right, or back side. This evaluates object-centered perspective reasoning in controlled automotive scenes.
- **Face canonical view selection** (own vs. viewer perspective) (Fig. 20): These tasks introduce ambiguity in reference frames. From the person's *own perspective*, left and right correspond to their intrinsic orientation, whereas from the *viewer's perspective*, left/right are defined relative to the image frame.

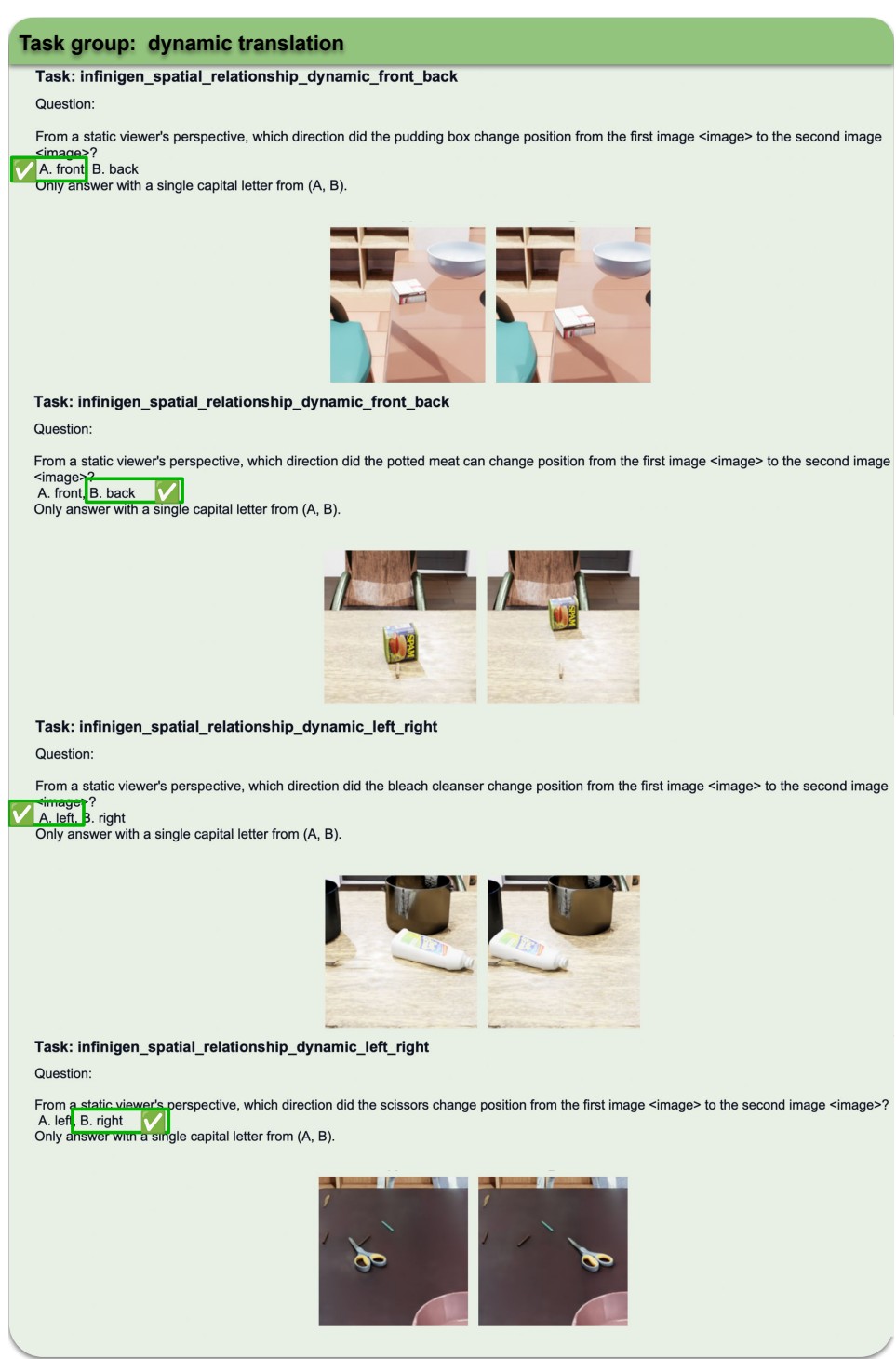

Figure 15: Examples of dynamic translation tasks. Objects undergo front–back (top) or left–right (bottom) displacements while the camera remains fixed. The model must classify the displacement direction.

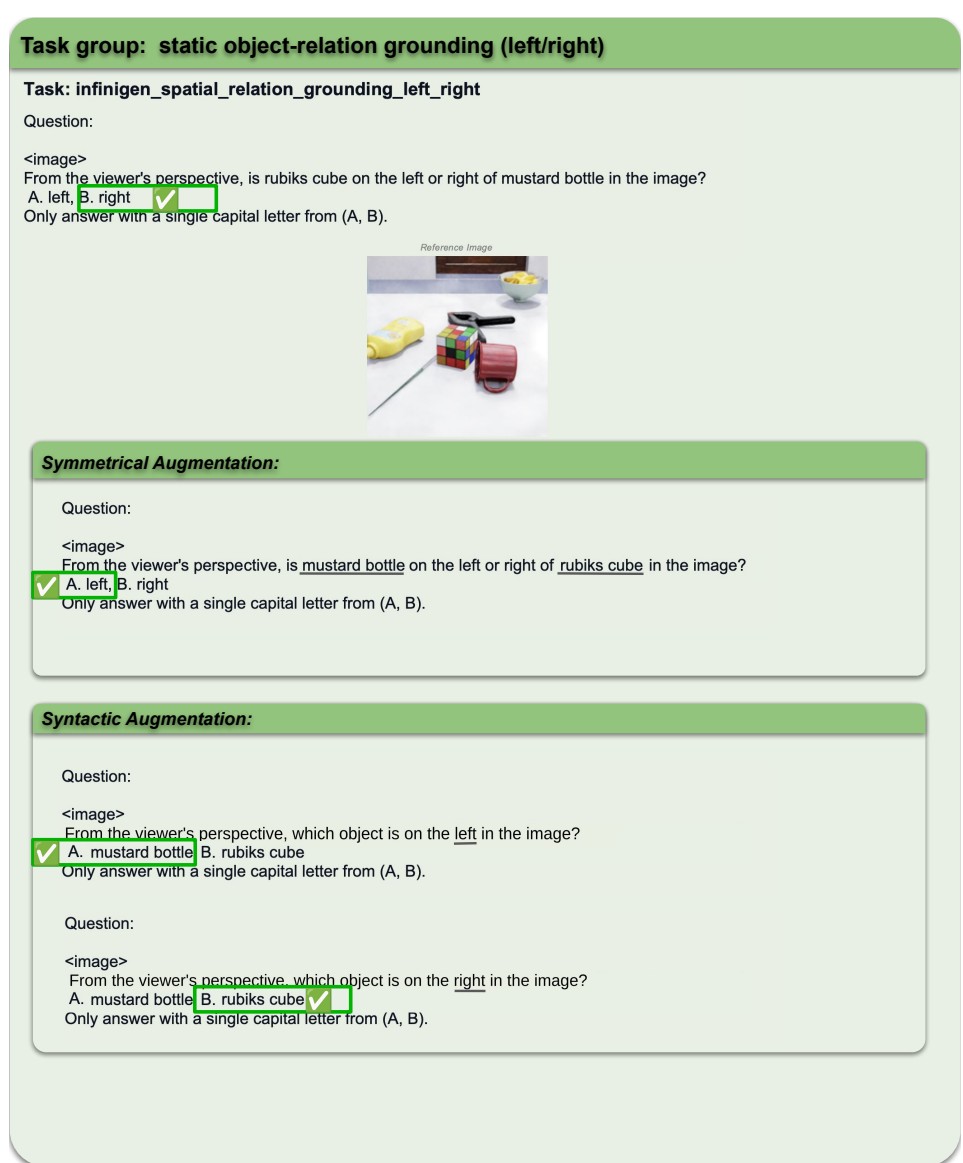

Figure 16: Examples of object-relation grounding **left/right** relation tasks. The base question asks whether a reference object (e.g., Rubik's cube) is positioned to the left or right of another object (e.g., mustard bottle) from the viewer's perspective. Symmetrical augmentation reverses the relation ("Is the mustard bottle to the left or right of the Rubik's cube?"), while syntactic augmentation reformulates the question style ("Which object is on the left?" vs. "Is object A on the left or right of object B?").

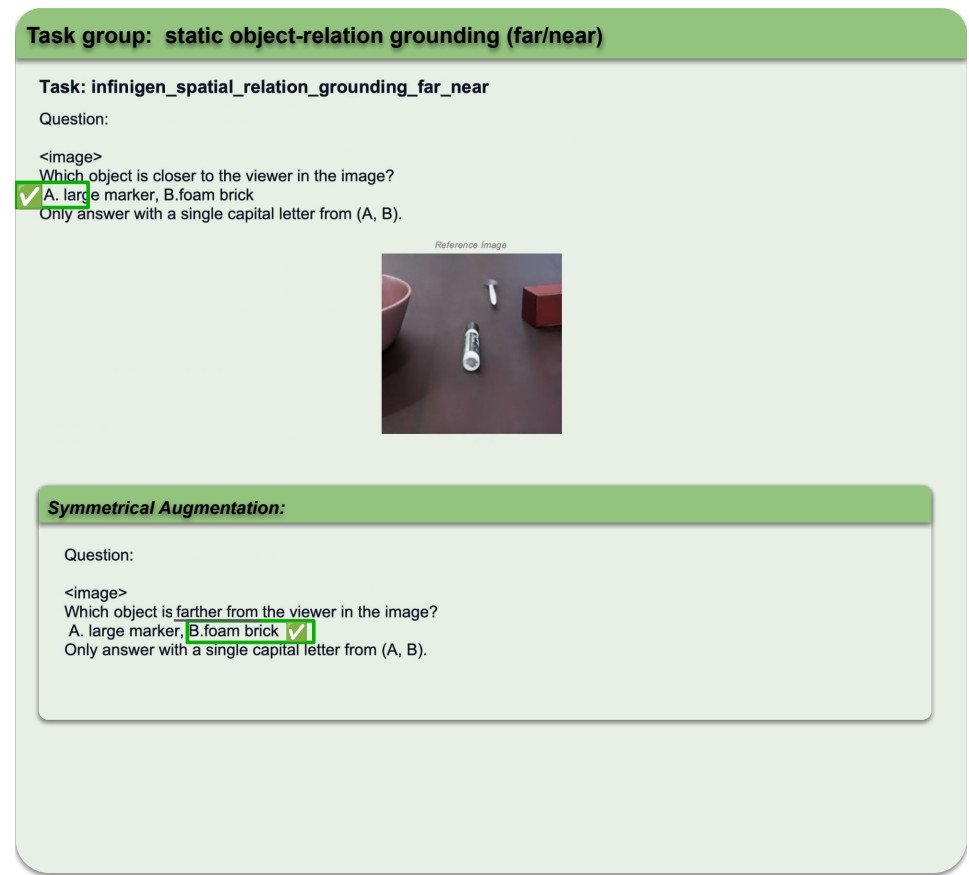

Figure 17: Examples of object-relation grounding **near/far** relation tasks. The model must determine which of two objects (e.g., marker vs. foam brick) is closer to the viewer within a single image. Symmetrical augmentation inverts the query ("Which object is farther?") while keeping the ground-truth relation consistent. This setup features distance-based reasoning from monocular perspective cues in static frames.

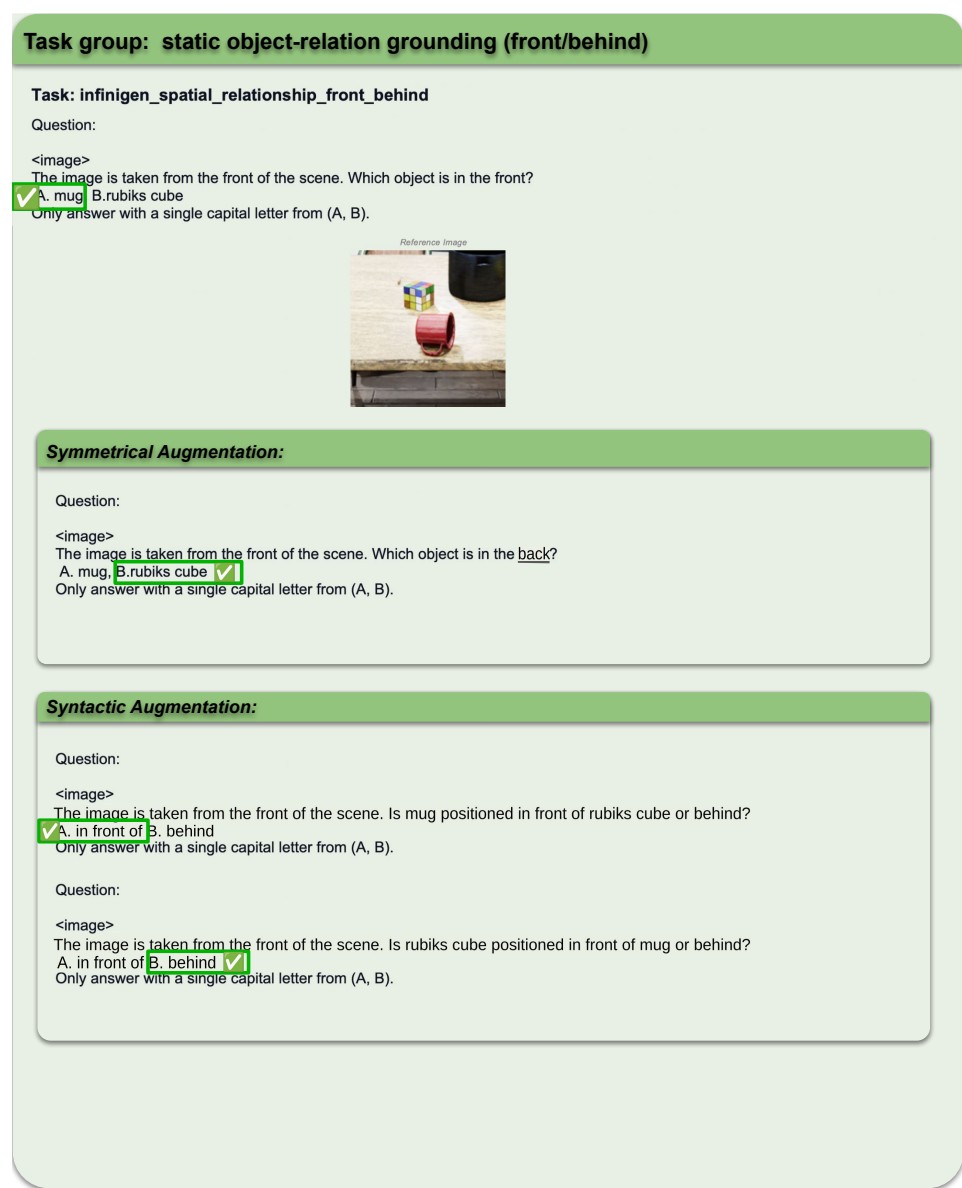

Figure 18: Examples of object-relation grounding **front/behind** relation tasks. Given a front-facing view, the model must decide which object (e.g., mug vs. Rubik's cube) is positioned in front or behind. Symmetrical augmentation flips the depth relation ("Which object is in the back?"), while syntactic augmentation reformulates the question ("Is the mug in front of or behind the cube?").

- **Object canonical view selection** (Fig. 21): Similar to cars, but applied to generic objects such as furniture. Models must map the front view to left, right, or back views, testing their ability to reason about viewpoint consistency across diverse shapes.

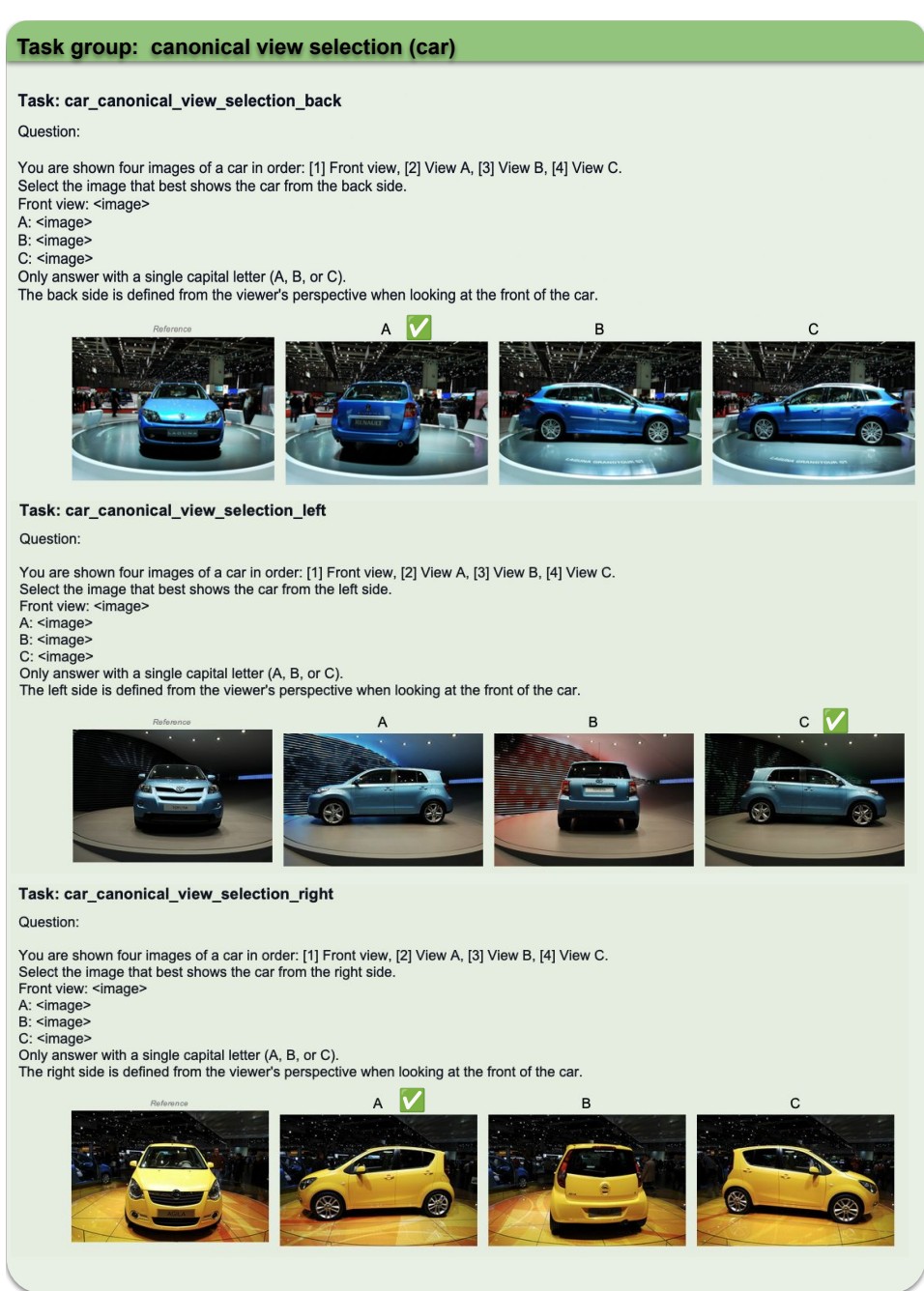

Figure 19: Examples of canonical view selection with cars. The model must select the correct left, right, or back view given a front-view reference.

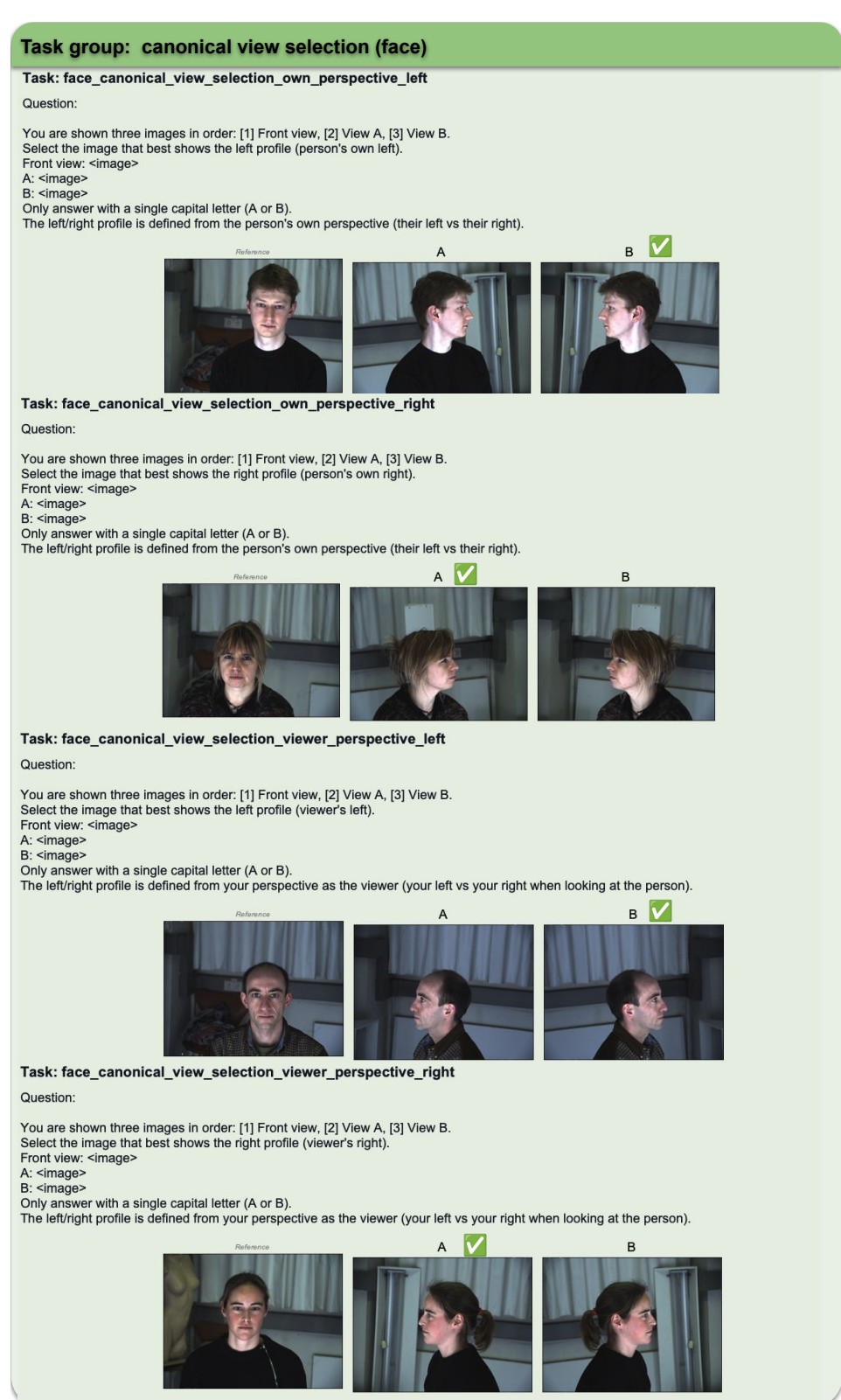

Figure 20: Examples of canonical view selection with faces. Tasks differ depending on whether left/right is defined from the subject's own perspective or from the external viewer's perspective.

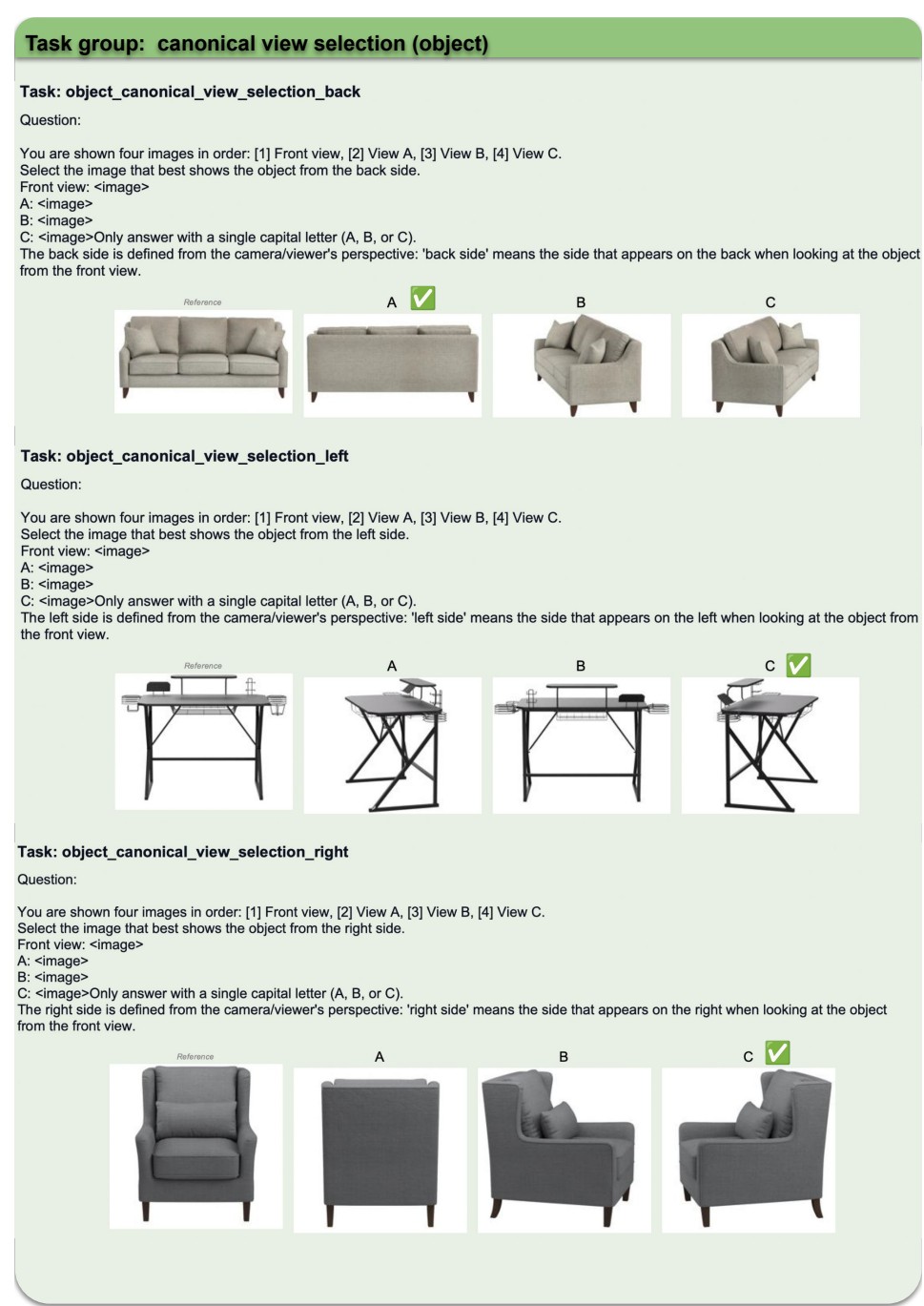

Figure 21: Examples of canonical view selection with objects. The model must determine left, right, or back views of generic objects such as furniture, based on a given front view.

**A.4.6 Perspective Taking (View Selection)** These tasks evaluate whether models can perform perspective taking when selecting the correct viewpoint of a scene, even when parts of objects are occluded. The challenge lies in integrating the reference view with multiple candidate perspectives, reasoning about hidden surfaces, and maintaining consistent spatial relationships. Variants differ in the extent of occlusion.

- **Full occlusion** (Fig. 22): The model sees a reference front view and must choose among candidate views taken from back, left, or right perspectives, where large occluders hide significant portions of the scene. Success requires inferring unseen object sides.
- **No occlusion** (Fig. 23): Similar setup, but with no major occlusion.
- **Partial occlusion** (Fig. 24): Candidate views contain moderate occlusion (e.g., objects partially blocking others). The model must still identify the correct viewpoint, balancing visible cues with inferred hidden structures.

**A.4.7 Perspective Taking (Relative Position Transformation)** This task group evaluates whether models can correctly predict how spatial relationships between objects transform under perspective shifts. Unlike view selection tasks, where the goal is to choose the correct viewpoint of a scene, these tasks explicitly probe relational transformations: given a reference view, the model must infer how relative positions (e.g., left/right, near/far) are altered when the viewpoint changes to the back, left, or right side. In other words, the challenge lies in mentally re-projecting the scene and predicting the new arrangement of objects from a different vantage point. To diagnose different failure modes in spatial reasoning, we introduce premise-based variations in spatial transformation tasks. In the *with-premise* condition, the relevant spatial relationship (e.g., "A is to the right of B in the front view") is provided directly in the prompt along with the corresponding image, allowing the model to reason over an explicit linguistic premise. In the *without-premise* condition, no such information is given, and the model must infer spatial relations from the reference image. This controlled comparison does not assume a specific order of grounding and reasoning, but instead helps identify whether failures arise from difficulties in extracting spatial relations from visual input, or from applying geometric reasoning given a known premise.

- **With premise** (Figs. 25, 26, 27): The model is given a linguistic statement of the relative positions (e.g., "X is closer than Y" or "X is to the left of Y") alongside the image, and must predict how that relation transforms under a new viewpoint.
- **Without premise** (Figs. 28, 29, 30): The model only sees the reference image and must infer the spatial relations itself before applying the geometric transformation to a new viewpoint.

**A.4.8 Mental Rotation** Mental rotation tasks evaluate a model's ability to simulate object transformations by imagining how an object's orientation changes under specified rotations. Unlike perspective-taking tasks, which require adopting a different viewpoint, mental rotation requires reasoning about the intrinsic geometry of a single object as it spins in place. In these tasks, the model is presented with a reference front view of an object and a description of a rotation (e.g., "rotate 135 degrees clockwise"). It must then select the correct image among several candidates that matches the object's new orientation. This requires integrating visual recognition with geometric transformation, a key hallmark of human mental imagery. These tasks are particularly diagnostic because they isolate the ability to track orientation without introducing multi-object relations or cluttered scene grounding.

- **Object Mental Rotation** (Fig. 31): The model is asked to determine the new orientation of a single object after a specified angular rotation. For example, given a chair in its canonical front-facing view, the model must predict which candidate corresponds to a 135-degree clockwise rotation. Success requires both accurate angle-tracking and strong spatial imagination.
- **Car Mental Rotation**: The model is asked to determine the new orientation of the car after a specified angular rotation.
- **Infinigen Mental Rotation**: The model must determine the new orientation of a target object (e.g., a clamp) on a cluttered tabletop after it undergoes a specified angular rotation.

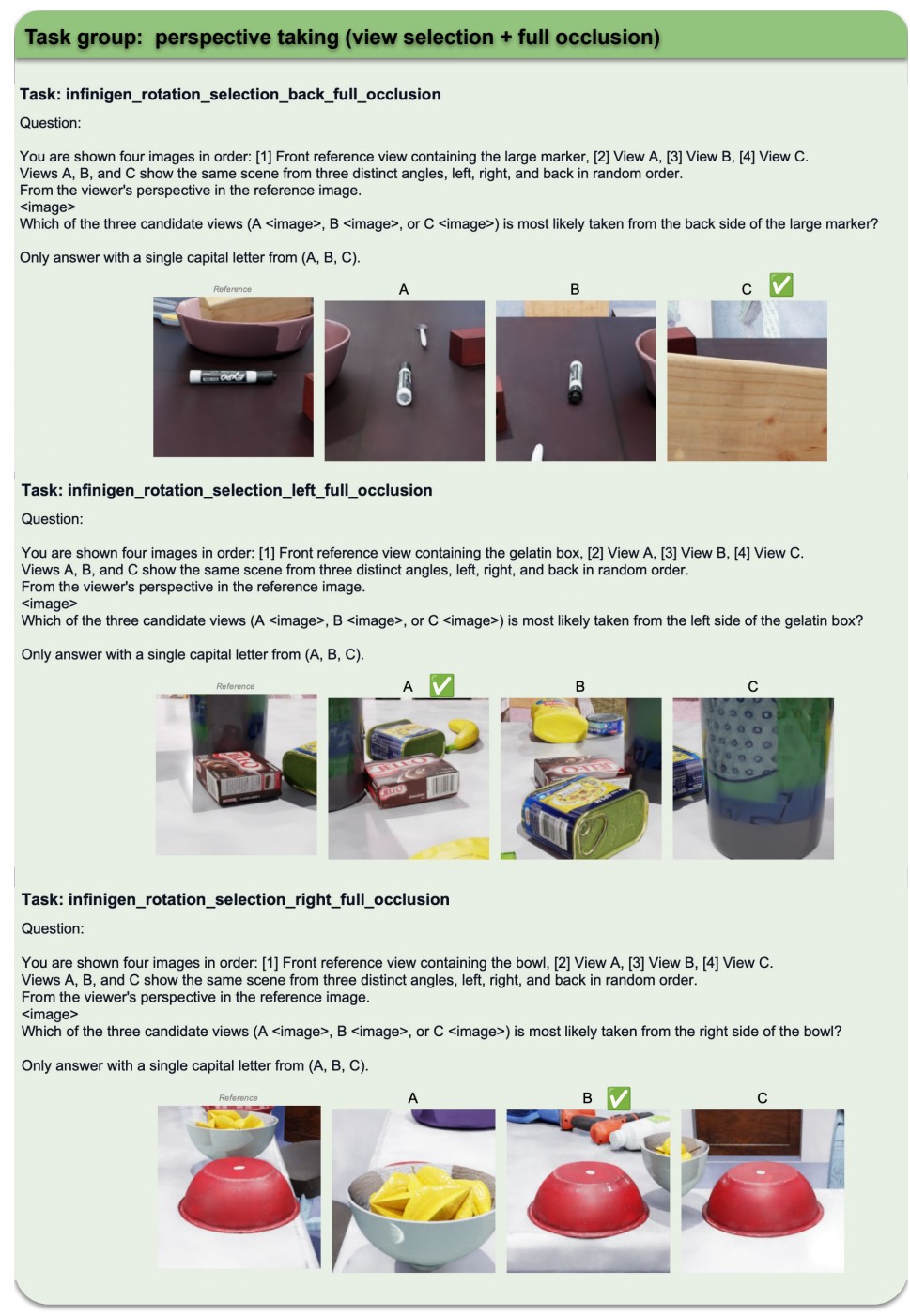

Figure 22: Examples of full occlusion perspective taking. Large occluders hide most of the target object, requiring inference about unseen sides.

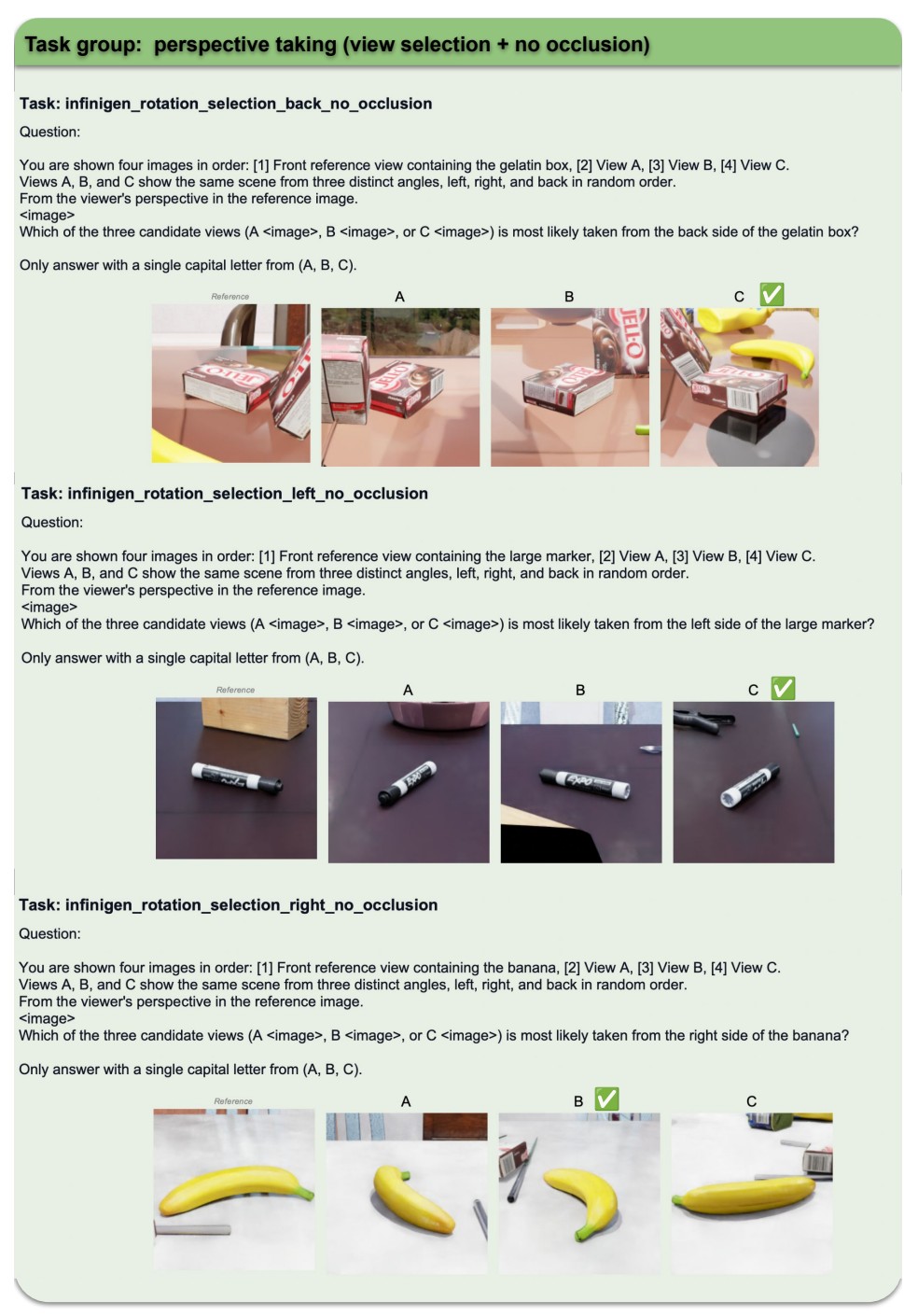

Figure 23: Examples of perspective taking without occlusion. Models rely solely on spatial consistency across viewpoints.

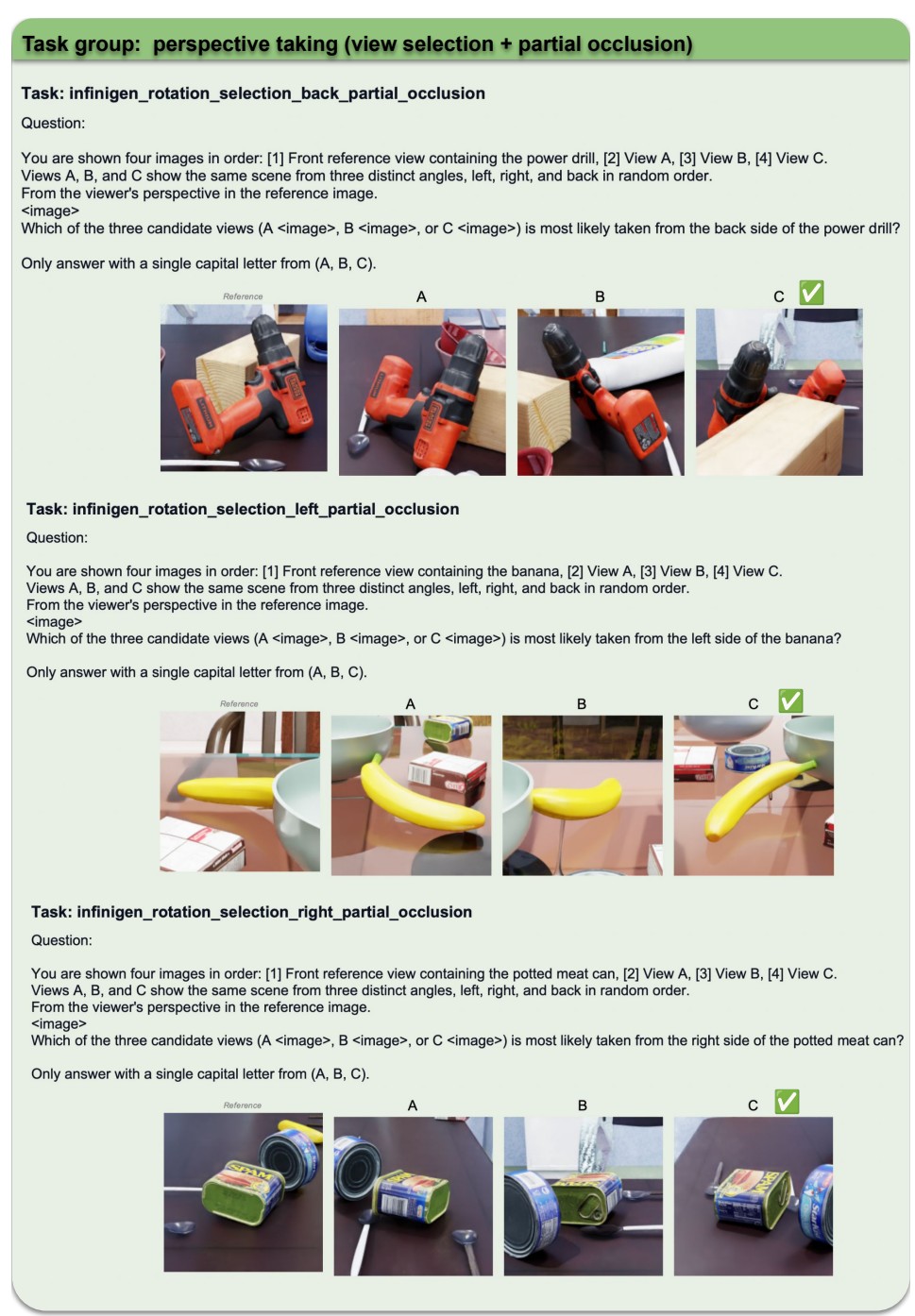

Figure 24: Examples of partial occlusion perspective taking. Candidate views contain moderate occluders, requiring reasoning across partially visible cues.

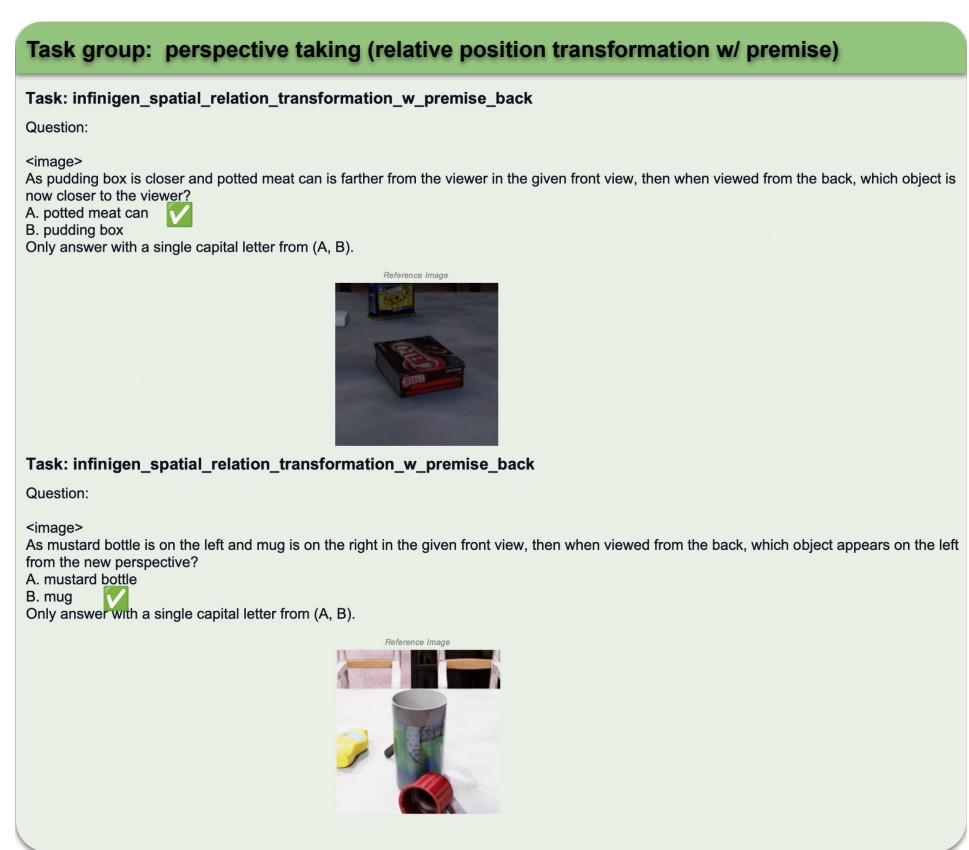

Figure 25: Examples of perspective taking with relative position transformation (with premise), back view.

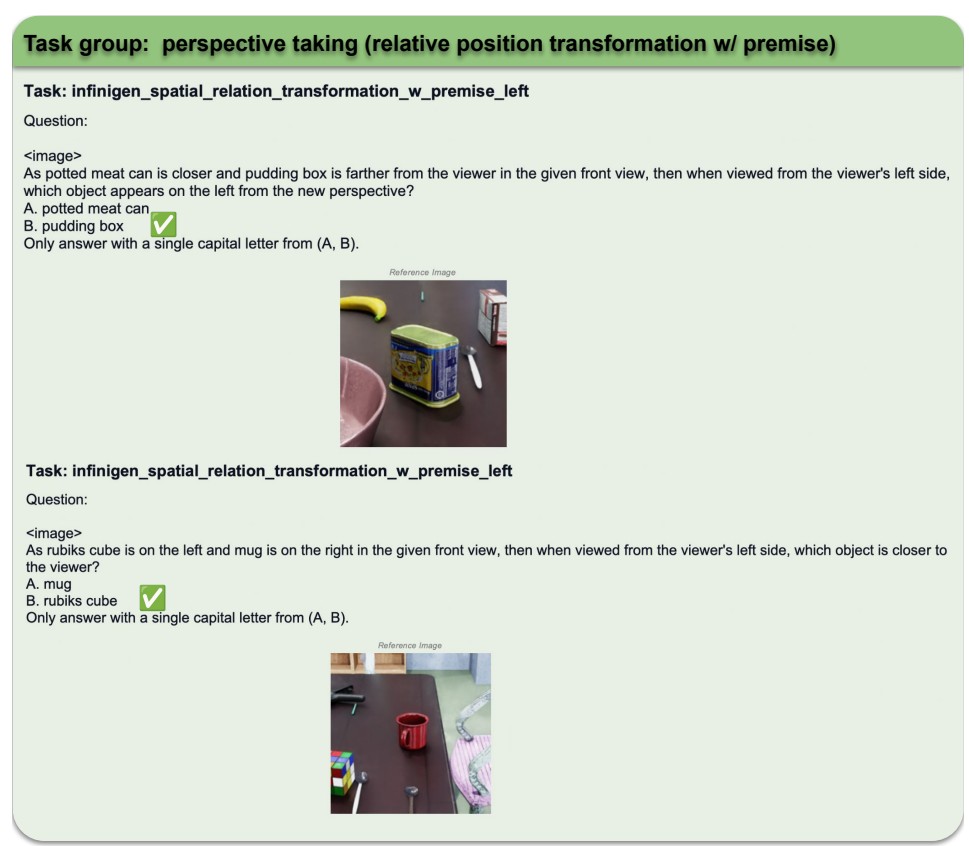

Figure 26: Examples of perspective taking with relative position transformation (with premise), left view.

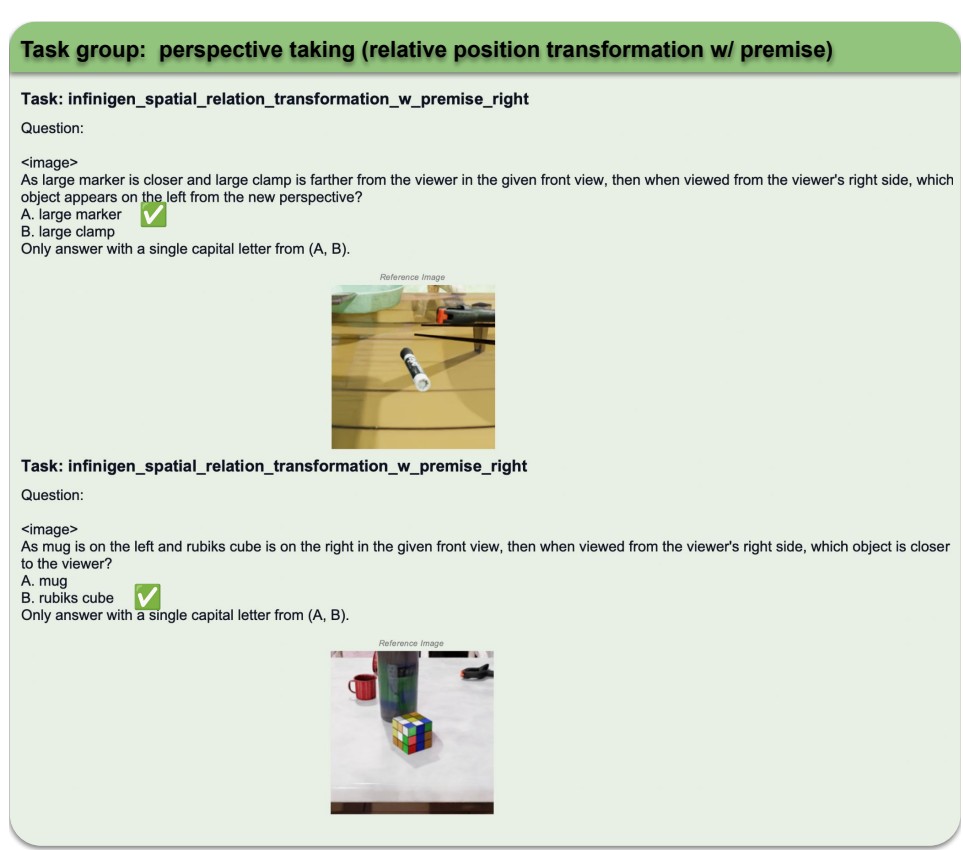

Figure 27: Examples of perspective taking with relative position transformation (with premise), right view.

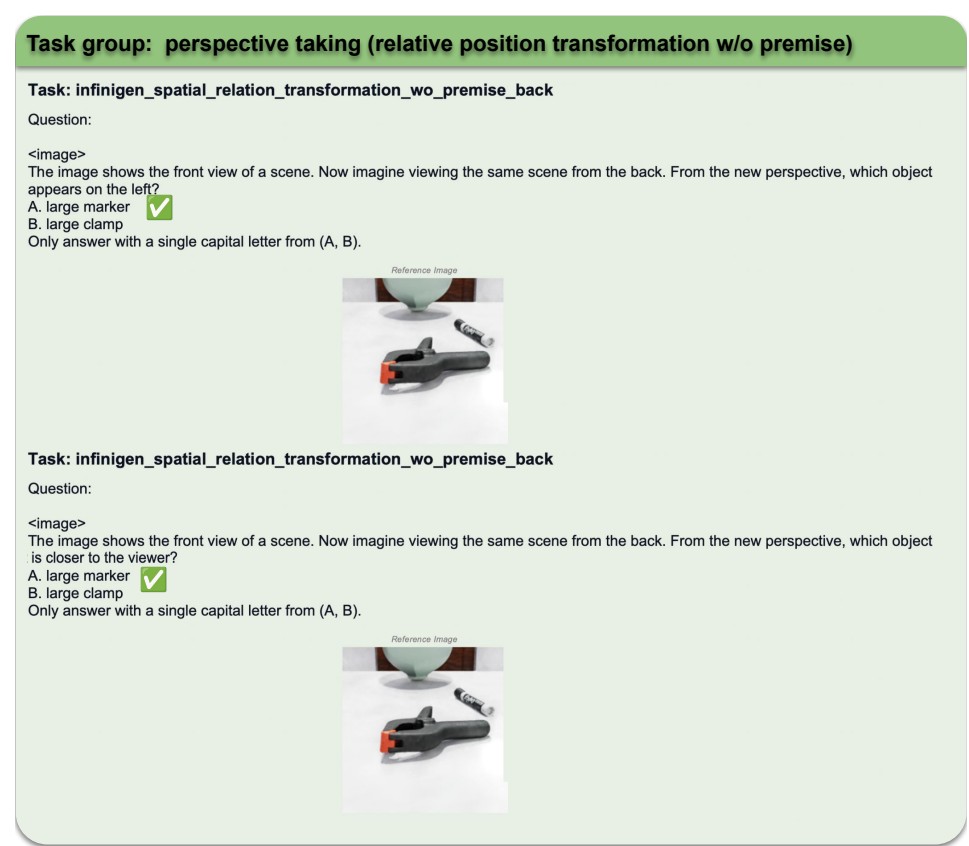

Figure 28: Examples of perspective taking with relative position transformation (without premise), back view.

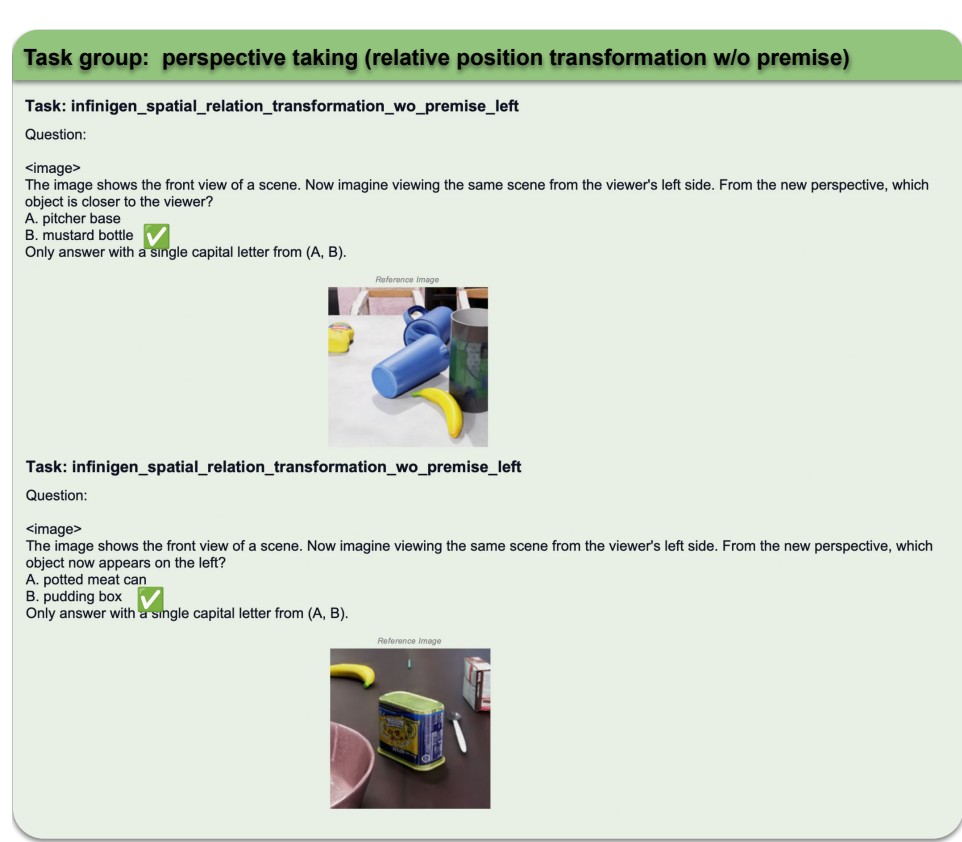

Figure 29: Examples of perspective taking with relative position transformation (without premise), left view.

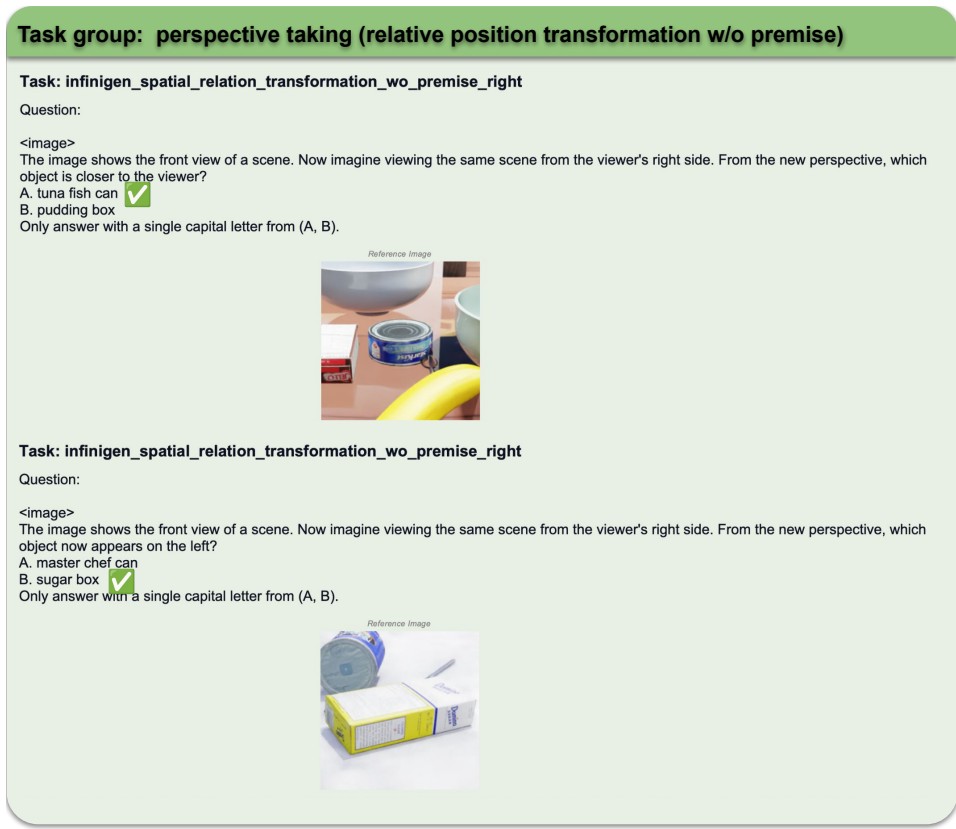

Figure 30: Examples of perspective taking with relative position transformation (without premise), right view.

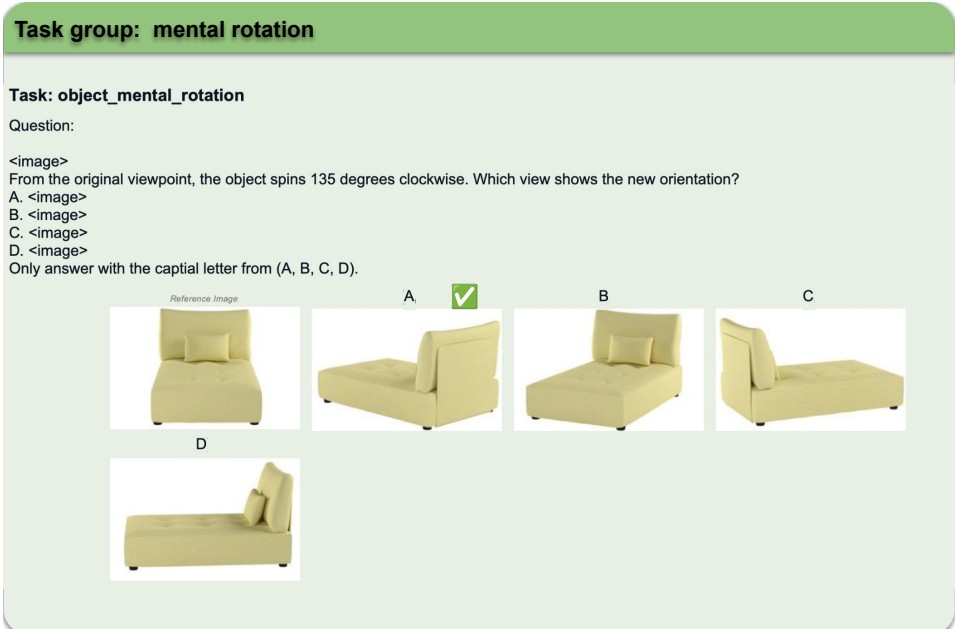

Figure 31: Examples of mental rotation tasks. The task presents a reference object (e.g., sofa with cushions) and specifies a degree of rotation (e.g., 135° clockwise). The model must identify which of the candidate views (A–D) corresponds to the rotated orientation.

# B DETAILED VLMS EVALUATION RESULTS

## B.1 RAW ACCURACY AND COHEN'S KAPPA

In addition to the main grouped heatmap reported in the paper, we provide complementary visualizations to support detailed analysis of model performance. Figure 32 reports raw accuracy for the grouped 26 task variants, enabling comparison with the chance-adjusted results in the main text. Figures 33 and 34 further expand to the ungrouped 51 subtype level, presenting both Cohen's $\kappa$ and raw accuracy. Together, these heatmaps give a complete view of performance across models, tasks, and evaluation metrics.

## B.2 DETAILED CONSISTENCY EVALUATIONS

**B.2.1 Augmentation Types** The benchmark employs two systematic augmentation strategies to probe reasoning consistency:

1. **Symmetric Augmentation:** Logically equivalent transformations that flip spatial relations while maintaining semantic meaning (e.g., *"Which object is on the left?"* → *"Which object is on the right?"* with corresponding answer adjustments).

2. **Syntactic Augmentation:** Surface-level reformulations that preserve semantic content while changing question structure (e.g., *"Which object is on the left?"* → *"Is object A on the left or right of object B?"*).

**B.2.2 Performance Metrics**

**Accuracy (%):** Overall correctness rate calculated as:

$$\text{Accuracy} = \left( \frac{\text{Correct Responses}}{\text{Total Test Cases}} \right) \times 100\%$$

**Consistency (%):** Average of pairwise consistency across task variants, where consistency is achieved when a pair of question variants yields identical outcomes (both correct or both incorrect).

**Perfect Rate (%):** Frequency of achieving complete consistency across all four question variants:

$$\text{Perfect Rate} = \left( \frac{\text{Cases with All-Agree Patterns}}{\text{Total Four-variant Cases}} \right) \times 100\%$$

This includes both **CCCC** (all correct) and **WWWW** (all wrong) patterns.

As shown in Table 5, the InternVL model family continues to demonstrate outstanding stability in spatial reasoning. Nearly all InternVL3 variants achieve consistency above 90% while maintaining strong accuracy and high perfect rates, making them the most uniformly reliable open-source models in the benchmark. Among proprietary models, gpt-5 remains the strongest overall model, achieving 78.5% accuracy and 97.0% consistency, setting a new standard for both competence and stability.

A strong positive correlation between accuracy and consistency is evident across the entire model distribution, but the relationship becomes nonlinear at higher performance levels. For example, gemini-2.5-pro attains the second-highest accuracy (75.2%) yet ranks noticeably lower in consistency (94.4%) compared with the InternVL family and gpt-5. These deviations illustrate that once models reach very high accuracy, gains in consistency no longer translate proportionally into improved reasoning, and correctness, not merely coherence, becomes the limiting factor. Mid-tier models show significant variability in the accuracy-consistency trade-off. Models like Molmo-7B achieve 73.5% consistency despite only 51.8% accuracy, indicating systematic but often incorrect reasoning patterns. Conversely, models like Gemma-3-27B maintain 59.8% accuracy but exhibit poor consistency at 53%, suggesting reliance on surface-level pattern matching rather than robust spatial understanding. The Perfect Rate metric reveals additional nuances in model behavior. High perfect rates indicate models that, when consistent, tend to be systematically correct or incorrect across variants. Lower perfect rates suggest fragmented reasoning where models may correctly

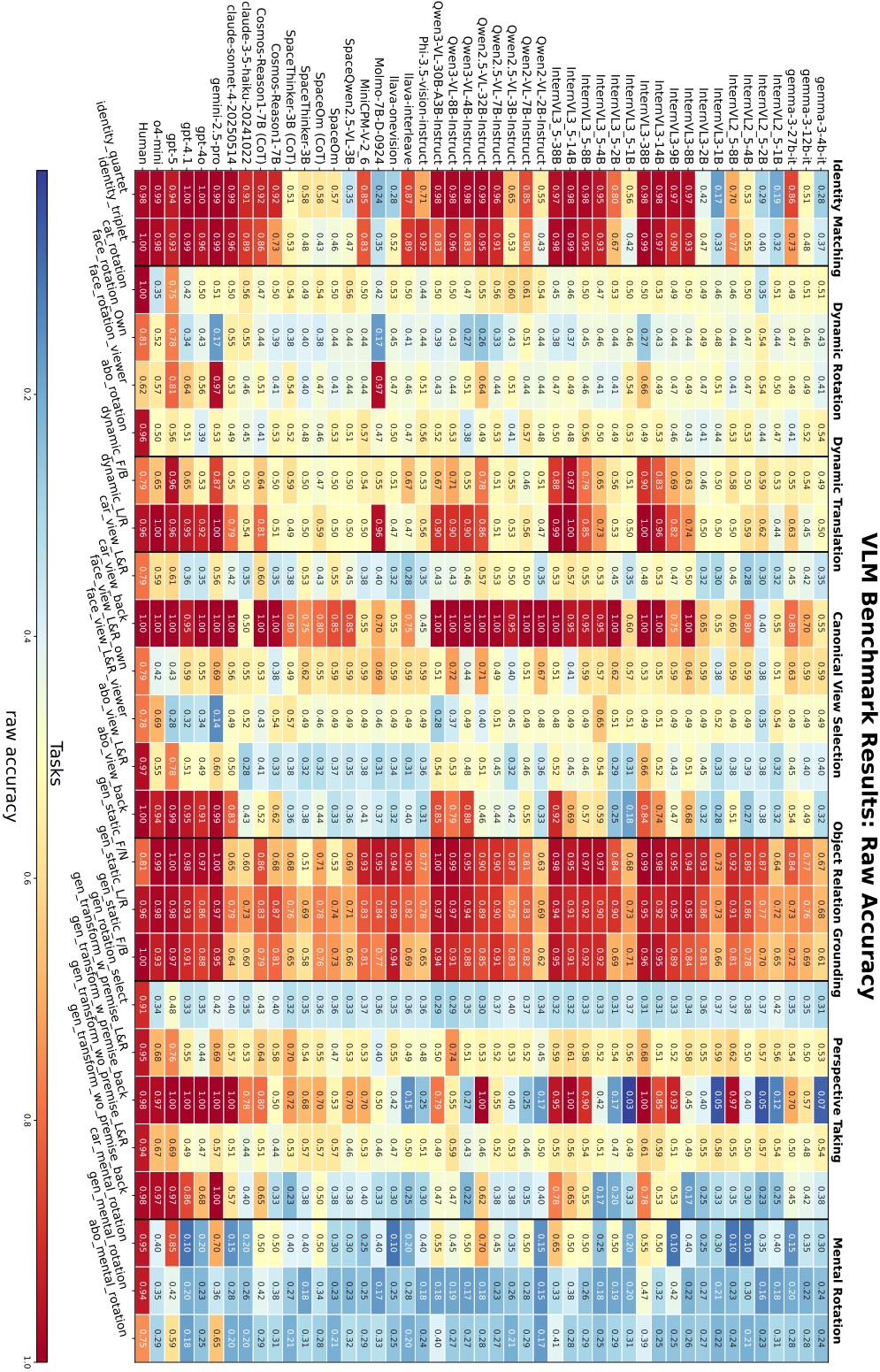

Figure 32: Performance heatmap of 43 VLMs across 26 grouped task variants, organized under 7 spatial reasoning categories. Values show raw accuracy rather than Cohen's $\kappa$. The 26 columns aggregate 51 fine-grained subtypes by domain or directional symmetry.

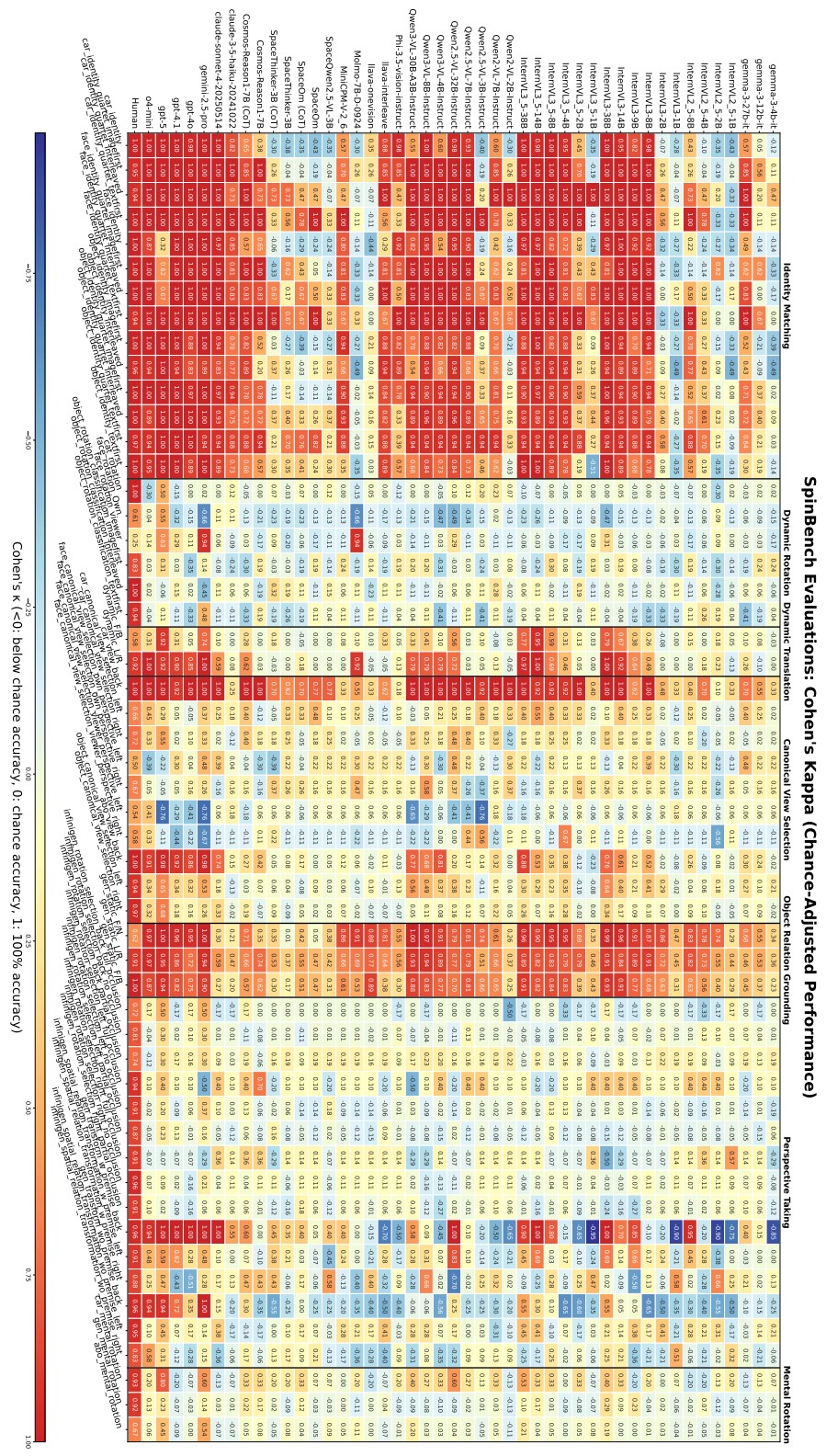

Figure 33: Ungrouped subtype-level heatmap of 43 VLMs showing Cohen's $\kappa$ performance across all 51 fine-grained spatial reasoning subtypes.

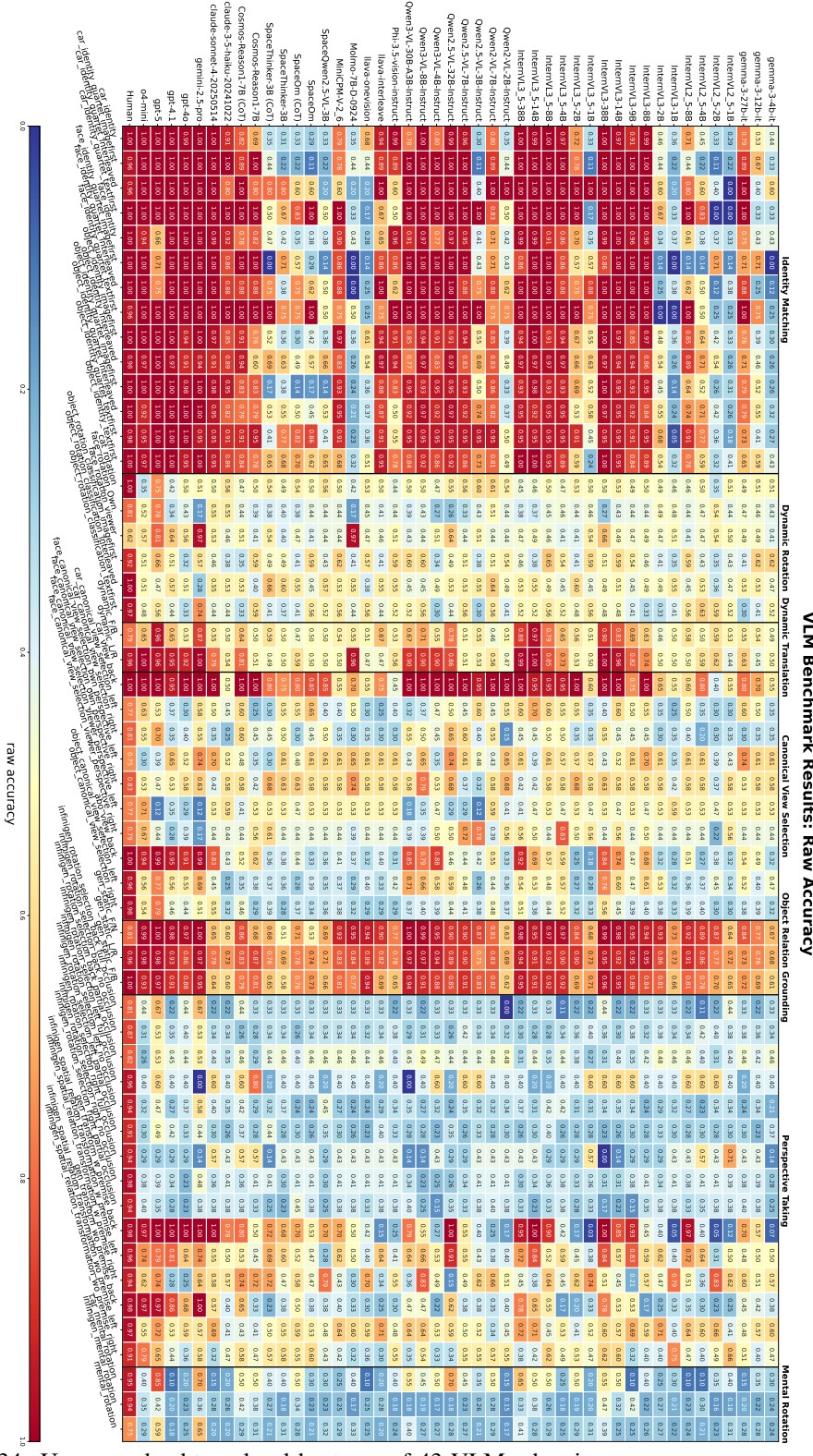

Figure 34: Ungrouped subtype-level heatmap of 43 VLMs showing raw accuracy across 51 fine-grained spatial reasoning subtypes. This complements Figure 33 by reporting unadjusted accuracy scores for the same set of subtypes.

answer some variants while failing others, indicating incomplete spatial representations. The substantial performance gap between top and bottom models underscores the significant challenges in achieving consistent and correct spatial reasoning.

Table 5: Comprehensive performance ranking of 43 vision-language models on spatial reasoning tasks. Accuracy represents overall correctness across all test cases. Consistency measures reasoning stability across question variants, and Perfect Rate indicates the frequency of achieving complete consistency across all four question variants (all correct or all incorrect).

| Model | Accuracy (%) | Consistency (%) | Perfect Rate (%) |
|---|---|---|---|
| gpt-5 | 78.5 | 97.0 | 75.8 |
| OpenGVLab_InternVL3_5_38B | 70.3 | 95.3 | 75.1 |
| OpenGVLab_InternVL3_38B | 72.6 | 95.7 | 71.1 |
| gemini-2.5-pro | 75.2 | 94.4 | 67.4 |
| o4-mini | 71.0 | 91.1 | 73.2 |
| OpenGVLab_InternVL3_5_14B | 68.3 | 94.6 | 68.7 |
| OpenGVLab_InternVL3_14B | 68.6 | 91.4 | 63.7 |
| Qwen_Qwen3_VL_8B_Instruct | 67.4 | 87.4 | 70.0 |
| Qwen_Qwen3_VL_30B_A3B_Instruct | 65.7 | 88.1 | 70.0 |
| Qwen_Qwen2.5_VL_32B_Instruct | 65.4 | 85.7 | 62.7 |
| gpt-4.1 | 67.2 | 85.9 | 59.5 |
| Qwen_Qwen2.5_VL_32B_Instruct | 67.3 | 85.7 | 62.7 |
| OpenGVLab_InternVL3_9B | 65.6 | 82.4 | 66.7 |
| OpenGVLab_InternVL3_5_8B | 68.6 | 83.9 | 59.7 |
| OpenGVLab_InternVL3_8B | 66.7 | 77.4 | 62.7 |
| OpenGVLab_InternVL3_5_4B | 65.8 | 77.1 | 61.7 |
| gpt-4o | 67.8 | 79.6 | 51.2 |
| Qwen_Qwen3_VL_4B_Instruct | 61.4 | 81.8 | 53.7 |
| Qwen_Qwen2.5_VL_7B_Instruct | 64.5 | 63.8 | 59.2 |
| OpenGVLab_InternVL2_5_8B | 61.2 | 65.9 | 57.7 |
| claude-sonnet-4-20250514 | 64.8 | 71.7 | 42.8 |
| llava-onevision-qwen2-7b-ov-hf | 55.6 | 67.4 | 58.2 |
| Cosmos-Reason1-cot-7B | 64.1 | 69.5 | 41.8 |
| openbmb_MiniCPM_V_2_6 | 62.1 | 64.2 | 45.3 |
| Qwen_Qwen2_VL_7B_Instruct | 61.8 | 60.2 | 44.8 |
| OpenGVLab_InternVL3_2B | 55.7 | 58.4 | 50.7 |
| allenai_Molmo_7B_D_0924 | 51.8 | 73.5 | 35.8 |
| OpenGVLab_InternVL2_5_4B | 55.9 | 63.4 | 39.3 |
| OpenGVLab_InternVL3_5_2B | 56.3 | 59.5 | 42.3 |
| llava_hf_llava_interleave_qwen_7b_hf | 58.9 | 65.2 | 31.3 |
| Qwen_Qwen2.5_VL_3B_Instruct | 55.3 | 53.4 | 38.3 |
| google_gemma_3_27b_it | 59.8 | 53 | 25.4 |
| Cosmos-Reason1-7B | 58.8 | 40.9 | 43.3 |
| google_gemma_3_12b_it | 54.3 | 58.4 | 25.9 |
| SpaceThinker-Qwen2.5VL-3B-cot | 51.4 | 53.4 | 33.3 |
| SpaceOm-cot | 53.2 | 51.6 | 31.8 |
| microsoft_Phi_3.5_vision_instruct | 57.7 | 40.5 | 23.9 |
| OpenGVLab_InternVL2_5_2B | 50.5 | 50.5 | 21.4 |
| claude-3-5-haiku | 57.3 | 36.9 | 21.4 |
| google_gemma_3_4b_it | 47.8 | 40.1 | 15.9 |
| OpenGVLab_InternVL3_1B | 47.2 | 35.1 | 22.4 |
| OpenGVLab_InternVL2_5_1B | 46.4 | 37.6 | 19.9 |
| SpaceQwen2.5-VL-3B-Instruct | 49.4 | 33.7 | 16.4 |
| Qwen_Qwen2_VL_2B_Instruct | 48.3 | 29 | 17.9 |

**Table 5 continued from previous page**

| Model | Accuracy (%) | Consistency (%) | Perfect Rate (%) |
|---|---|---|---|
| OpenGVLab_InternVL3_5_1B | 48.8 | 28.3 | 16.9 |
| internlm_internlm_xcomposer2d5_7b | 43.3 | 36.6 | 13.9 |
| SpaceOm | 49.2 | 17.9 | 18.4 |
| SpaceThinker-Qwen2.5VL-3B | 48.7 | 11.1 | 8.5 |

**B.2.3  Augmentation Strategy Analysis**   In Fig. 36, the augmentation comparison reveals similar consistency rates across different transformation types (66-68%), with symmetric augmentations performing marginally better. The small performance gaps (less than 3 percentage points) suggest that current vision-language models exhibit similar levels of sensitivity to symmetric and syntactic augmentation. The substantial error bars (approximately ±8-10%) indicate variance in augmentation sensitivity across models.

**B.2.4  Pattern Distribution Analysis**   The stacked bar chart Fig. 35 reveals several key insights into model consistency behavior. High-performing models (topmost bars) demonstrate substantially larger proportions of perfect consistency patterns (CCCC, all four variants correct), with top models achieving above 60% perfect consistency rates. Conversely, lower-performing models show more fragmented pattern distributions with higher prevalences of mixed consistency patterns and complete failure modes (WWWW). The visualization demonstrates a clear correlation between overall accuracy and consistency stability. Models that perform well on spatial reasoning tasks also maintain more coherent reasoning across question variants.

## B.3  Correlation Analysis

We calculated correlations between our diagnostic benchmark and four established spatial reasoning benchmarks at both overall and subtask levels (Table 6). Overall correlations between our diagnostic benchmark and holistic benchmarks were weak and non-significant: MindCube (r = -0.088, p = 0.836), ViewSpatial-Bench (r = 0.460, p = 0.299), OmniSpatial (r = 0.456, p = 0.137), and SpaCE-10 (r = 0.098, p = 0.803). These results validate that our approach captures distinct foundational capabilities rather than general spatial intelligence. Subtask correlations revealed targeted diagnostic relationships (Figure 37, 39, 38, 40). Significant correlations emerged between specific diagnostic and benchmark subtasks: dynamic rotation abilities strongly predict 3D reasoning performance in MindCube (r = 0.829, p = 0.021), identity matching correlates with person-based perspective taking in ViewSpatial-Bench (r = 0.915, p = 0.030), and static reasoning predicts object manipulation capabilities in OmniSpatial (r = 0.764, p = 0.006). SpaCE-10 showed no significant correlations, suggesting it evaluates distinct spatial reasoning components.

These patterns demonstrate that our diagnostic benchmark provides complementary rather than redundant evaluation. While overall performance correlations are minimal, specific subtask relationships reveal how foundational spatial deficits contribute to failures in complex holistic tasks, enabling targeted identification of improvement areas.

Table 6: Overall Average Correlation Analysis: Diagnostic Benchmark vs. Holistic Spatial Benchmarks

| Benchmark | number of models | Pearson r | p-value |
|---|---|---|---|
| MindCube Yin et al. (2025) | 8 | -0.088 | 0.836 |
| ViewSpatial-Bench Li et al. (2025a) | 7 | 0.460 | 0.299 |
| OmniSpatial Jia et al. (2025a) | 12 | 0.456 | 0.137 |
| SpaCE-10 Gong et al. (2025) | 9 | 0.098 | 0.803 |

## C  Human Evaluations

We conducted human evaluation with twelve subjects to establish performance baselines and validate task difficulty.  One subject completed the full benchmark (2,599 questions), while eleven

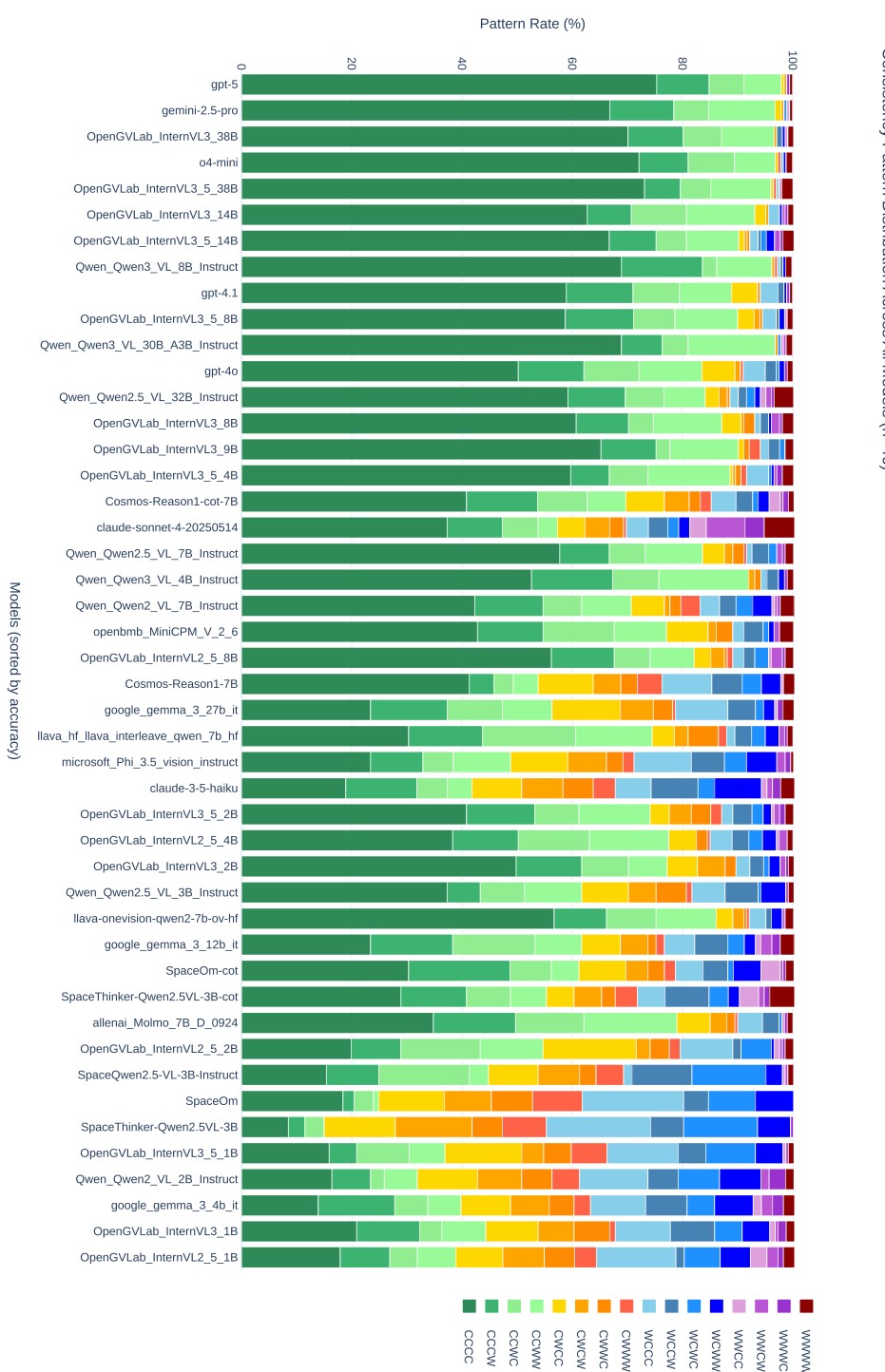

Figure 35: Comprehensive consistency pattern distribution across all 46 vision-language models, sorted by overall accuracy (top to bottom). Each stacked bar represents the percentage distribution of all 16 possible consistency patterns (CCCC through WWWW) for 4-variant question sets. Green shades indicate patterns with more correct responses (C), while red shades represent patterns with more wrong responses (W). Models with higher accuracy (left) show greater prevalence of all-correct patterns (CCCC, dark green).

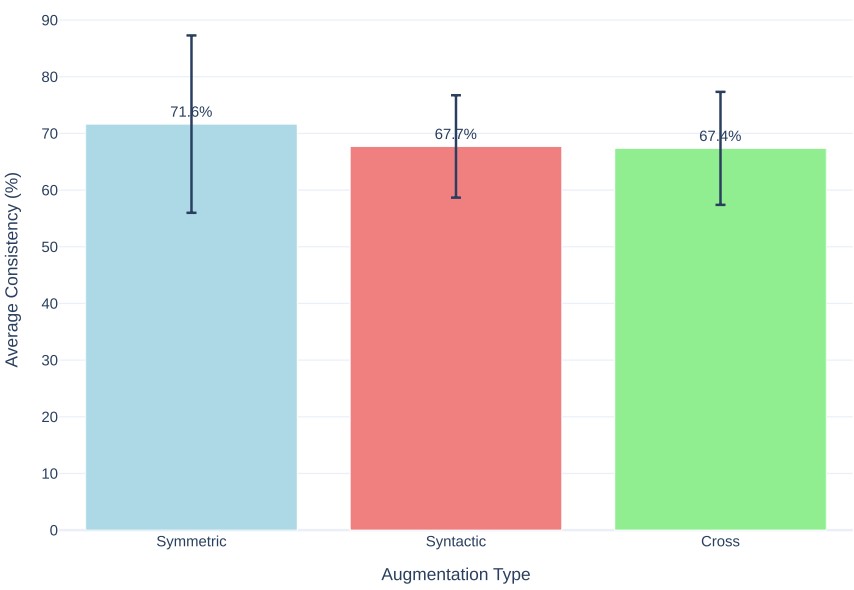

Figure 36: Average consistency rates by augmentation strategy across 46 vision-language models with 4-variant question sets. Error bars represent standard deviation across models. Symmetric augmentations (question reformulations maintaining logical equivalence) achieve slightly higher consistency than syntactic (surface-level rephrasing) and cross-augmentation (mixed transformations) approaches.

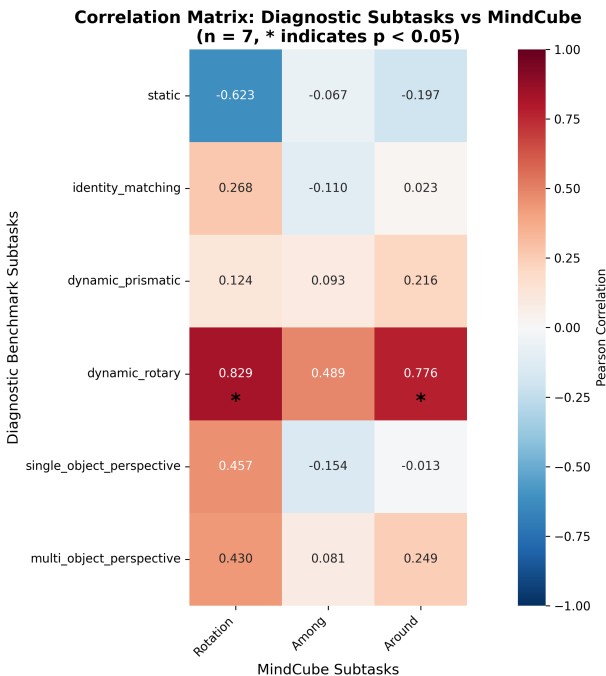

Figure 37: Subtask-level correlation matrix between SpinBench components and Mindcube. Rows represent our six diagnostic subtasks, columns represent subtasks from MindCube (n=8). Color intensity indicates correlation strength: red denotes positive correlations, blue denotes negative correlations.

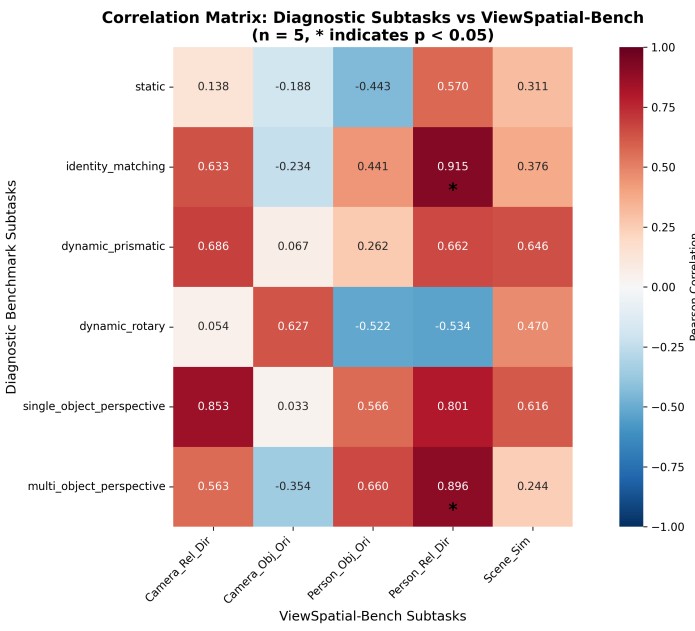

Figure 38: Subtask-level correlation matrix between SpinBench components and ViewSpatial-Bench. Rows represent our six diagnostic subtasks, columns represent subtasks from ViewSpatial-Bench (n=7). Color intensity indicates correlation strength: red denotes positive correlations, blue denotes negative correlations.

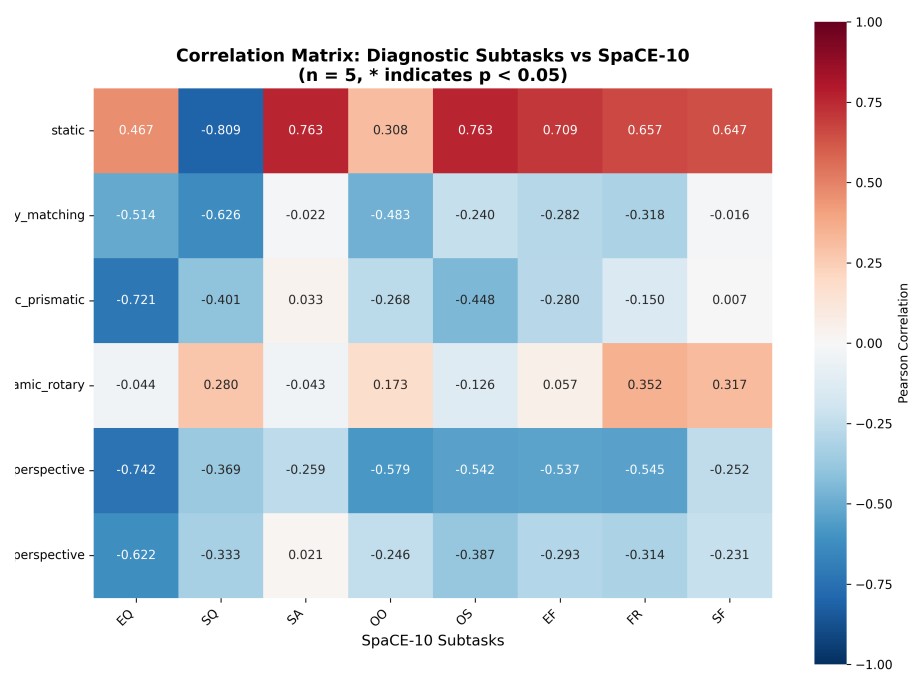

Figure 39: Subtask-level correlation matrix between SpinBench components and SpaCE-10. Rows represent our six diagnostic subtasks, columns represent subtasks from SpaCE-10 (n=9). Color intensity indicates correlation strength: red denotes positive correlations, blue denotes negative correlations.

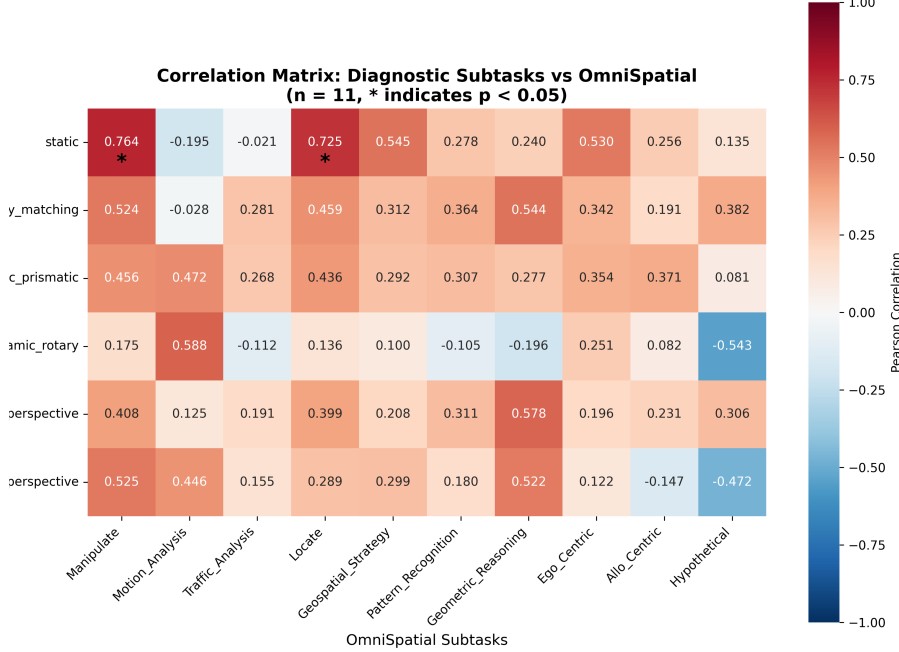

Figure 40: Subtask-level correlation matrix between SpinBench components and OmniSpatial. Rows represent our six diagnostic subtasks, columns represent subtasks from OmniSpatial (n=12). Color intensity indicates correlation strength: red denotes positive correlations, blue denotes negative correlations.

others completed balanced samples of 200 questions each, with equal representation across all task subtypes.

## C.1 HUMAN EVALUATION TOOL DESIGN

We developed a specialized application for human evaluation. The tool handles diverse question formats automatically, from single reference images with text options to complex multi-image image options mental rotation tasks.

**C.1.1 Question Type Detection and Display**  The system automatically parses question structure using pattern matching to distinguish between reference images and selectable options. For spatial reasoning tasks with text choices (e.g., "A. mug, B. mustard bottle"), it displays the reference image alongside clearly labeled text options. For mental rotation tasks presenting multiple candidate views, it identifies the initial reference state and labels the four candidate images as selectable options. This smart labeling prevents confusion about which elements are answerable choices versus contextual information.

**C.1.2 Progress Management and Resumption**  The tool implements progress tracking with automatic saving after each response. Subjects can resume interrupted sessions seamlessly. Questions are grouped and sorted by task type to minimize cognitive switching costs, with pop-up notifications when transitioning between task categories.

**C.1.3 Dataset Curation Integration**  Beyond collecting responses, the tool also supports real-time dataset quality control. Subjects can flag ambiguous or problematic questions for removal using a dedicated key. This dual-purpose design allows human evaluation to simultaneously serve as both a performance benchmark and a dataset refinement process.

**C.1.4 Response Collection**  The interface uses numbered keyboard input (1-4 corresponding to A-D) for efficient response collection, with visual feedback for correctness and validation to prevent invalid inputs. All responses include precise timestamps for response time analysis, with automatic filtering of extended intervals ($> 180s$) that indicate interruptions rather than genuine decision time. The tool generated detailed logs in JSONL format containing individual responses, task-specific performance breakdowns, and timing statistics, enabling comprehensive analysis of human performance patterns across different visual reasoning categories.

## C.2 HUMAN PERFORMANCE

Table 7: **Human Performance Statistics by Task Subtype.** Accuracy and response time statistics averaged across 12 human subjects, organized by task group categories.

| Task Subtype | Accuracy Mean ± SD | Response Time Mean ± SD (s) |
|---|---|---|
| **Canonical View Selection** | | |
| car canonical view selection back | 1.000 ± 0.000 | 12.5 ± 10.7 |
| car canonical view selection left | 0.771 ± 0.391 | 11.4 ± 6.7 |
| car canonical view selection right | 0.813 ± 0.386 | 6.7 ± 2.8 |
| face canonical view selection own perspective left | 0.750 ± 0.369 | 17.2 ± 10.2 |
| face canonical view selection own perspective right | 0.833 ± 0.389 | 6.2 ± 5.0 |
| face canonical view selection viewer perspective left | 0.771 ± 0.391 | 9.3 ± 6.5 |
| face canonical view selection viewer perspective right | 0.792 ± 0.382 | 5.3 ± 3.9 |
| object canonical view selection back | 0.999 ± 0.004 | 8.5 ± 3.4 |
| object canonical view selection left | 0.957 ± 0.097 | 9.3 ± 3.7 |
| object canonical view selection right | 0.979 ± 0.072 | 5.9 ± 4.0 |
| **Identity Matching** | | |

**Table 7 continued from previous page**

| Task Subtype | Accuracy Mean ± SD | Response Time Mean ± SD (s) |
|---|---|---|
| car identity | 0.999 ± 0.004 | 6.7 ± 4.1 |
| car identity quartet imagefirst | 0.963 ± 0.088 | 5.9 ± 4.3 |
| car identity quartet interleaved | 0.958 ± 0.097 | 10.0 ± 5.2 |
| car identity quartet textfirst | 1.000 ± 0.000 | 4.5 ± 3.3 |
| face identity | 1.000 ± 0.000 | 4.2 ± 1.9 |
| face identity quartet imagefirst | 1.000 ± 0.000 | 4.1 ± 1.8 |
| face identity quartet interleaved | 1.000 ± 0.000 | 3.3 ± 1.0 |
| face identity quartet textfirst | 0.958 ± 0.097 | 3.9 ± 1.6 |
| object identity imagefirst | 1.000 ± 0.000 | 3.5 ± 1.5 |
| object identity interleaved | 0.979 ± 0.072 | 4.5 ± 2.5 |
| object identity quartet imagefirst | 0.998 ± 0.007 | 2.6 ± 0.8 |
| object identity quartet interleaved | 1.000 ± 0.000 | 3.3 ± 1.2 |
| object identity quartet textfirst | 0.979 ± 0.072 | 2.1 ± 0.7 |
| object identity textfirst | 1.000 ± 0.000 | 2.5 ± 0.8 |
| **Dynamic Rotation** | | |
| car rotation classification | 0.999 ± 0.004 | 19.0 ± 11.1 |
| face rotation classification own perspective | 0.806 ± 0.220 | 12.2 ± 6.8 |
| face rotation classification viewer perspective | 0.624 ± 0.390 | 14.3 ± 12.4 |
| object rotation classification imagefirst | 0.917 ± 0.207 | 12.4 ± 7.3 |
| object rotation classification interleaved | 1.000 ± 0.000 | 8.5 ± 5.2 |
| object rotation classification textfirst | 0.972 ± 0.096 | 8.0 ± 5.1 |
| **Dynamic Translation** | | |
| infinigen spatial relationship dynamic front back | 0.792 ± 0.351 | 17.4 ± 12.9 |
| infinigen spatial relationship dynamic left right | 0.958 ± 0.097 | 8.5 ± 4.0 |
| **Object Relation Grounding** | | |
| infinigen spatial relation grounding far near | 0.810 ± 0.240 | 10.5 ± 3.8 |
| infinigen spatial relation grounding left right | 0.956 ± 0.097 | 13.6 ± 5.3 |
| infinigen spatial relationship front behind | 0.998 ± 0.007 | 13.9 ± 7.1 |
| **Perspective Taking** | | |
| infinigen rotation selection back full occlusion | 0.813 ± 0.155 | 33.0 ± 15.0 |
| infinigen rotation selection back no occlusion | 0.873 ± 0.167 | 24.4 ± 14.5 |
| infinigen rotation selection back partial occlusion | 0.825 ± 0.225 | 25.3 ± 12.6 |
| infinigen rotation selection left full occlusion | 0.958 ± 0.144 | 20.4 ± 13.2 |
| infinigen rotation selection left no occlusion | 0.938 ± 0.113 | 15.4 ± 6.5 |
| infinigen rotation selection left partial occlusion | 0.915 ± 0.122 | 15.5 ± 9.4 |
| infinigen rotation selection right full occlusion | 0.938 ± 0.113 | 15.9 ± 6.5 |
| infinigen rotation selection right no occlusion | 0.976 ± 0.072 | 15.8 ± 7.7 |
| infinigen rotation selection right partial occlusion | 0.938 ± 0.113 | 13.8 ± 5.8 |
| infinigen spatial relation transformation w premise back | 0.979 ± 0.072 | 25.0 ± 11.3 |
| infinigen spatial relation transformation w premise left | 0.957 ± 0.097 | 18.6 ± 6.4 |
| infinigen spatial relation transformation w premise right | 0.938 ± 0.113 | 17.4 ± 6.4 |
| infinigen spatial relation transformation wo premise back | 0.979 ± 0.072 | 16.6 ± 5.3 |
| infinigen spatial relation transformation wo premise left | 0.991 ± 0.021 | 14.8 ± 5.0 |
| infinigen spatial relation transformation wo premise right | 0.913 ± 0.161 | 13.6 ± 5.8 |
| **Mental Rotation** | | |
| object mental rotation | 0.749 ± 0.321 | 17.2 ± 8.8 |
| **Overall** | **0.921 ± 0.091** | **11.6 ± 6.9** |

**Overall Performance.** Human subjects achieved high overall accuracy (0.921 ± 0.091) across the benchmark, as detailed in Tables 7 and 8, demonstrating that while tasks vary significantly in cognitive difficulty, they remain within human capability. Some task groups showed excellent accuracy, with Identity Matching achieving the highest performance (0.988 ± 0.052). The primary

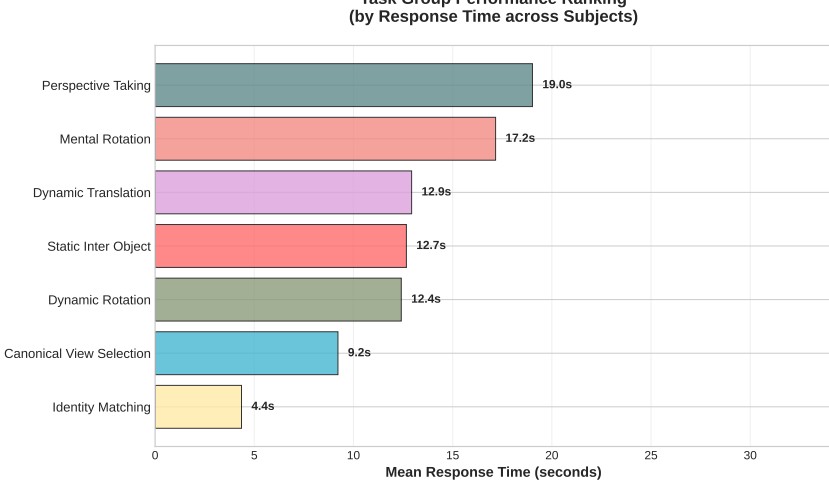

Figure 41: **Task Group Performance Ranking by Human Response Time.** Perspective Taking emerges as the most cognitively demanding task group (19.0s), followed by Mental Rotation (17.2s) and Dynamic Translation (12.9s). Identity Matching tasks show the fastest response times (4.4s), indicating a 4-fold difficulty range across major cognitive categories.

exceptions were Mental Rotation (0.749 ± 0.321), which showed the highest variability and included some of the most challenging scenarios in the benchmark. Response times varied dramatically across tasks, ranging from 2.1 seconds for the fastest subtypes to 33.0 seconds for the most challenging, indicating substantial variation in cognitive difficulty.

Table 8: **Human Performance Summary by Task Group.** Accuracy and response time statistics across 7 major task categories. Accuracy and response time means are averaged across 12 human subjects within each task group. Response time range shows the span of mean response times across different subtypes within each group (fastest to slowest subtype mean).

| Task Group | Accuracy Mean ± SD | Response Time Mean ± SD (s) | Response Time Range (s) | Tasks (n) |
|---|---|---|---|---|
| Identity Matching | 0.988 ± 0.052 | 4.4 ± 2.6 | 2.1 - 10.0 | 14 |
| Canonical View Selection | 0.866 ± 0.301 | 9.2 ± 6.3 | 5.3 - 17.2 | 10 |
| Object Relation Grounding | 0.921 ± 0.149 | 12.7 ± 5.6 | 10.5 - 14.0 | 3 |
| Dynamic Translation | 0.875 ± 0.257 | 12.9 ± 9.6 | 8.5 - 17.4 | 2 |
| Dynamic Rotation | 0.886 ± 0.205 | 12.4 ± 8.5 | 8.5 - 19.0 | 6 |
| Mental Rotation | 0.749 ± 0.321 | 17.2 ± 8.8 | 17.2 - 17.2 | 1 |
| Perspective Taking | 0.928 ± 0.128 | 19.0 ± 9.4 | 13.6 - 33.0 | 15 |
| **Overall** | **0.921 ± 0.091** | **11.6 ± 6.9** | **2.1 - 33.0** | **51** |

**Task Group Difficulty Ranking.** Analysis of the full benchmark results reveals clear difficulty hierarchies across major task groups, as shown in Figure 41. By response time, the most challenging groups are: (1) Perspective Taking (19.0s), demanding viewpoint reasoning often under occlusion; (2) Mental Rotation (17.2s), requiring complex 3D spatial transformations; (3) Dynamic Translation (12.9s), involving spatial movement tracking; (4) Object-Relation Grounding (12.7s), requiring analysis of spatial relationships between multiple objects; (5) Dynamic Rotation (12.4s), involving rotational movement classification; (6) Canonical View Selection (9.2s), focusing on optimal viewing angles; and (7) Identity Matching (4.4s), the fastest category involving object recognition across viewpoints.

The most demanding individual subtypes, detailed in Figure 42, include perspective-taking tasks under full occlusion (infinigen rotation selection back full occlusion: 33.0s), complex spatial transformations (infinigen spatial relation transformation with premise back: 25.0s), and partial occlu-

sion scenarios (infinigen rotation selection back partial occlusion: 25.3s). Conversely, the fastest responses occur in identity matching tasks, particularly object identity quartet text-first (2.1s), suggesting these tap into rapid visual recognition processes that require minimal deliberative reasoning.

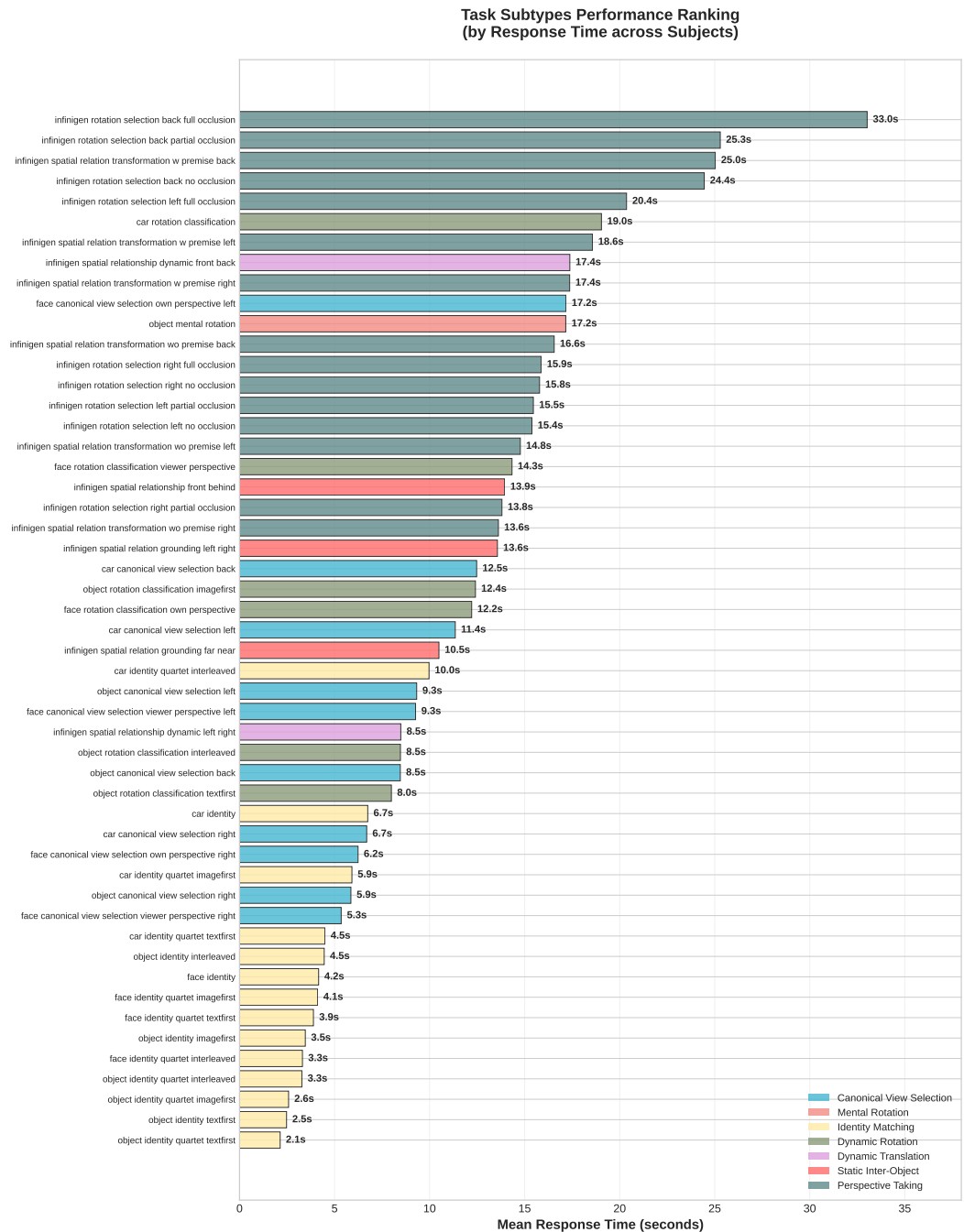

Figure 42: **Detailed Subtype Performance Rankings.** The 51 task subtypes ranked by mean human response time, revealing extreme variation from 2.1s to 33.0s. Perspective-taking tasks under occlusion (dark teal) dominate the most challenging subtypes, while identity matching tasks (yellow) cluster among the fastest responses. Color coding indicates task group membership.

**Task Design Implications.** The human performance data validates our benchmark's difficulty gradient and identifies genuinely challenging spatial reasoning scenarios. Tasks combining multiple cognitive demands—such as perspective taking under occlusion or spatial transformations requiring

premise integration—emerge as the most demanding, requiring both extended processing time while generally maintaining high accuracy. The 4.3-fold difference between the easiest (Identity Matching: 4.4s) and hardest (Perspective Taking: 19.0s) task groups demonstrates that our benchmark successfully spans a wide range of spatial reasoning difficulties, from rapid visual recognition to complex 3D transformations requiring nearly half a minute of deliberation.

## C.3 CORRELATION ANALYSIS

**Human-VLM Performance Correlation.** To validate that our benchmark captures genuine spatial reasoning difficulty rather than arbitrary task complexity, we analyzed the relationship between human cognitive load and VLM performance across task subtypes. We calculated the correlation between mean human response times (averaged across 12 subjects per task) and mean VLM accuracy (averaged across 37 models per task) for each of the 51 task subtypes in our benchmark.

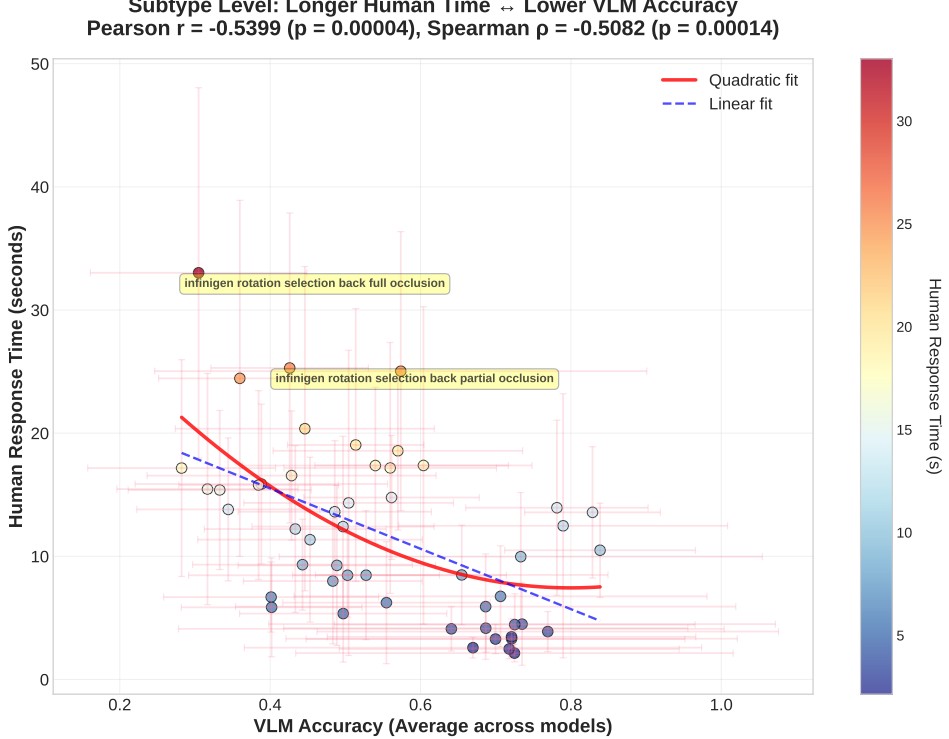

Figure 43: **Human-VLM Performance Correlation.** Scatter plot showing the relationship between VLM accuracy (x-axis) and human response time (y-axis) across 51 task subtypes. For each task subtype, we computed: (1) mean human response time by averaging individual response times across all 12 human subjects who completed that task, and (2) mean VLM accuracy by averaging performance across all 37 evaluated vision-language models on that same task. The correlation analysis treats each of the 51 task subtypes as an independent observation, examining whether tasks that require more human cognitive effort (longer response times) also prove more challenging for VLMs (lower accuracy). Color intensity indicates response time difficulty, with annotations highlighting the most challenging outliers.

Our analysis revealed a significant negative correlation between human response times and VLM accuracy (Pearson r = -0.5399, p < 0.0001; Spearman $\rho$ = -0.5082, p = 0.0001, n = 51 tasks), as illustrated in Figure 43. This moderate-to-strong correlation demonstrates that tasks requiring longer human processing time consistently challenge VLMs more severely, providing empirical evidence that our benchmark captures fundamental spatial reasoning difficulty shared across human and artificial intelligence systems.

**Cognitive Load.** The correlation analysis reveals that human cognitive load, as measured by response time, systematically predicts VLM performance degradation. Tasks in the upper-left region of Figure 43 with both long human response times and low VLM accuracy—represent the most cognitively demanding spatial reasoning scenarios in our benchmark. These include complex perspective-taking under occlusion (e.g., infinigen rotation selection back full occlusion: 33.0s response time), spatial transformations with premise integration (e.g., infinigen spatial relation transformation with premise back: 25.0s), and challenging mental rotation tasks (17.2s). Notably, while humans maintain high accuracy even on these slow tasks through extended deliberation, VLMs show systematic accuracy degradation on these same challenging scenarios. This divergence suggests that humans can leverage additional processing time to overcome spatial reasoning difficulties, while current VLMs face fundamental limitations .

**Benchmark Validity.** The systematic relationship between human cognitive difficulty and VLM performance provides strong evidence for our benchmark's construct validity. Rather than testing arbitrary visual challenges, our tasks appear to probe fundamental spatial reasoning capabilities that require significant cognitive resources for both human and artificial intelligence systems. The contrast between human speed-accuracy trade-offs (high accuracy with longer processing) and VLM limitations (lower accuracy regardless of computation time) highlights important gaps in current vision-language models' spatial reasoning abilities. This alignment suggests that improvements in VLM performance on our benchmark likely reflect genuine advances in spatial reasoning rather than dataset-specific optimizations.

**Benchmark Validity.** The systematic relationship between human cognitive difficulty and VLM performance provides strong evidence for our benchmark's construct validity. The negative correlation indicates that our tasks probe fundamental spatial reasoning capabilities that require significant cognitive resources across both biological and artificial intelligence systems. Rather than testing arbitrary visual challenges or dataset-specific artifacts, the alignment demonstrates that our benchmark captures core spatial reasoning demands. The contrast between human adaptive processing (achieving high accuracy through longer deliberation) and VLM limitations (showing lower accuracy) highlights important gaps in current vision-language models' spatial reasoning capabilities. This alignment suggests that improvements in VLM performance on our benchmark likely reflect genuine advances in spatial reasoning.

# D    DETAILS ON THE VLM EVALUATION SETUP

## D.1    EVALUATION CONFIGURATION

All models were evaluated with consistent parameters to ensure fair comparison:

- **Temperature:** 0.0 (deterministic sampling)

- **Top-p:** 1.0 (no nucleus sampling restriction)

**Image preprocessing:** Multi-image inputs were processed by interleaving text and image tokens according to each model's expected format.

**Answer extraction:** We employed robust pattern matching to extract answers (A, B, C, D) from model responses, checking for structured tags first (`<answer>A</answer>`) followed by standalone letters with word boundaries.

Referring to Chow et al. (Chow et al., 2025), during VLM evaluations, we appended an end prompt to each question-answer pair. The end prompt is as follows, depending on the actual option number for each task, as in Tab. 4:

```
Only answer with a single capital letter from (A, B).
Only answer with a single capital letter from (A, B, C).
Only answer with a single capital letter from (A, B, C, D).
```

## D.2 Model Implementations

### D.2.1 LMDeploy-Supported Models

For the majority of open-source models, we utilized LMDeploy (Contributors, 2023; Zhang et al., 2025a), a high-throughput inference engine optimized for large language models.

**Models using LMDeploy:**

- InternVL series: InternVL2.5 (1B–8B), InternVL3 (1B–38B), InternVL3.5 (1B–38B)

- Qwen-VL series: Qwen2-VL (2B–7B), Qwen2.5-VL (3B–32B)

- Gemma series:gemma-3-4b-it, gemma-3-27b-it, gemma-3-12b-it

- Additional models: Phi-3.5-vision-instruct, MiniCPM-V-2.6, Molmo-7B, llava-interleave-qwen-7b-hf

**Configuration:** We configured tensor parallelism (TP) settings based on model size: TP=1 for models less than 8B parameters, TP=2 for models less than 16B parameters, and TP=4 for larger models. Backend selection was automatically determined based on model compatibility, with TurboMind preferred for supported architectures and PyTorch as a fallback.

### D.2.2 Other Models

For models not supported by LMDeploy or requiring specialized handling, we employed the HuggingFace Transformers library with model-specific processors.

LLaVA-OneVision Model: We used the official LLaVA-OneVision implementation with `LlavaOnevisionForConditionalGeneration` and applied the chat template format for multi-image inputs.

Spatial Reasoning Models: For SpaceOm, SpaceThinker-Qwen2.5VL-3B, and SpaceQwen2.5-VL-3B-Instruct, we utilized `Qwen2_5_VLForConditionalGeneration` with specialized chat templates supporting structured reasoning formats.

Cosmos-Reason1-7B Model: we used the official LLaVA-OneVision implementation with vLLM (Kwon et al., 2023) with specialized vision processing utilities to handle multi-modal inputs efficiently.

## D.3 Prompt for Reasoning Models

In section 4.2, we evaluate the impact of CoT prompting across three specialized spatial reasoning models: Cosmos-Reason1 NVIDIA et al. (2025), SpaceOm Jia et al. (2025b), SpaceThinker Chen et al. (2024). We provide the prompts for each model: The prompt for Cosmos-Reason1 NVIDIA et al. (2025):

```
You are a helpful assistant.
Answer the question in the following format:
"<think>\nyour reasoning\n</think>
<answer>\nyour answer\n</answer>."
```

The prompt for SpaceOm Jia et al. (2025b) and SpaceThinker Chen et al. (2024):

```
You are VL-Thinking, a helpful assistant with
excellent reasoning ability.
You should first think about the reasoning process and then
provide the answer.
Use <think>...</think> and <answer>...</answer> tags.
```

# E  FINETUNING VLMS

To further validate whether improvements in foundational spatial reasoning skills transfer to the target perspective-taking task, we fine-tune Qwen3-VL-4B-Instruct and Qwen3-VL-8B-Instruct using GRPO for 6 epochs on 5,900 curated samples spanning four core spatial reasoning abilities: mental rotation, canonical-view selection, identity matching, and dynamic rotation.

## E.1  CROSS-TASK AND CROSS-DOMAIN VALIDATION.

Importantly, the validation set is drawn from a *different task type and domain* than training. Training samples contain object and face, while validation is performed on Infinigen-rendered 3D scenes from the perspective-taking task in SpinBench (not used in training). This setup evaluates both task-type generalization (from basic transformations to viewpoint reasoning) and domain generalization (from object/face images to simulated 3D scenes).

## E.2  TRAINING AND VALIDATION RESULTS

Figure 44 shows the GRPO learning curves.

We additionally report the initial validation accuracy reward (Step 0) and the highest validation accuracy reward achieved during training. These metrics reflect improvements on the target perspective-taking task:

| Model | Initial Val Accuracy Reward | Highest Val Accuracy Reward |
|---|---|---|
| Qwen3-VL-4B-Instruct | 0.344 | 0.389 |
| Qwen3-VL-8B-Instruct | 0.302 | 0.398 |

Table 9: Validation accuracy reward on the *perspective-taking* task before and after GRPO fine-tuning. 8B models improve noticeably from their initialization, demonstrating transfer from basic spatial skills to viewpoint transformation.

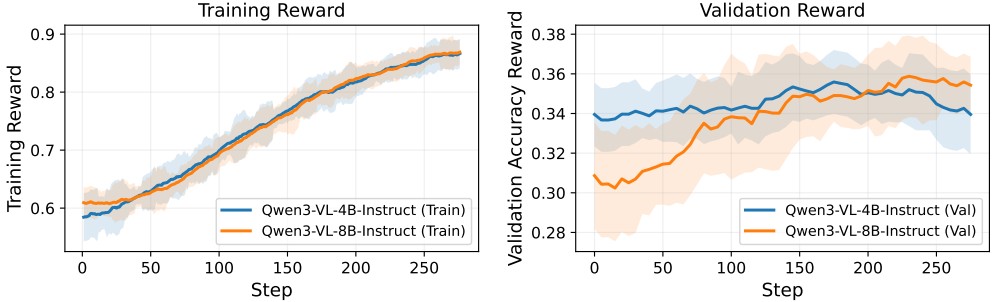

Figure 44: Training and validation reward curves for Qwen3-VL-4B-Instruct and Qwen3-VL-8B-Instruct under GRPO fine-tuning. The 8B model demonstrates greater improvement on the held-out perspective-taking validation task, consistent with its larger capacity. The 4B model shows a slight rise followed by a decline in validation reward, which may indicate partial overfitting to the training domain.

## E.3  GRPO TRAINING CONFIGURATION

We provide the full GRPO training configuration used for fine-tuning Qwen3-VL-4B-Instruct and Qwen3-VL-8B-Instruct. The same configuration is used for both 4B and 8B models unless otherwise noted. During rollout, we generate $n = 5$ samples per prompt with sampling temperature 1.0 and top-$p = 0.99$. Validation uses a more deterministic setting with temperature 0.5 and $n = 1$. Training uses global batch size 128, gradient norm clipping at 1.0, AdamW optimizer with learning rate $1 \times 10^{-6}$ and weight decay $1 \times 10^{-2}$.

## F    MORE RELATED WORKS

### F.1    SPATIAL REASONING BENCHMARKS

Beyond traditional vision-language datasets, BLINK (Fu et al., 2024) introduces tasks that humans can solve "within a blink," but which remain challenging for multimodal large language models (MLLMs). These tasks highlight persistent gaps between human perception and model capabilities—particularly in spatial reasoning. Recent benchmarks offer complementary perspectives on spatial reasoning: MindCube (Yin et al., 2025) and VSI-Bench (Yang et al., 2025b) focus on how MLLMs construct internal representations of space, a process analogous to cognitive mapping. These benchmarks primarily evaluate advanced, compositional tasks such as object identity tracking across frames, spatial relation grounding within a frame, and object motion understanding. However, they do not explicitly isolate or test foundational spatial skills like basic perspective taking or mental rotation. ViewSpatial-Bench (Li et al., 2025a) targets perspective-taking by evaluating object localization from different viewpoints. The core task is determining what is visible from a given perspective, a foundational problem in spatial understanding. SpaCE-10 (Gong et al., 2025) defines a taxonomy of atomic spatial skills for question answering, including object recognition, localization, spatial relations, size comparison, and counting. However, its reliance on scanned indoor scenes limits controlled testing of each skill in isolation. 3DSRBench (Ma et al., 2025a) centers on spatial reasoning in 3D environments, categorizing tasks into height, location, orientation, and multi-object reasoning. While comprehensive, its scope excludes key aspects of human spatial intelligence, such as perspective-taking and mental rotation. SPHERE (Zhang et al., 2024) proposes a hierarchical evaluation of vision-language models, progressing from single-skill to multi-skill tasks. Single-skill categories include position, counting, distance, and size. However, SPHERE primarily uses a single static image as input, limiting its capacity to evaluate dynamic or temporally grounded spatial understanding.

Several recent efforts draw inspiration from cognitive science: OmniSpatial (Jia et al., 2025a) offers tasks rooted in psychological theory, covering dynamic reasoning, complex spatial logic, spatial interactions, and perspective-taking. However, many of these tasks involve commonsense reasoning about motion and function, which are often entangled with spatial cognition, making it difficult to isolate spatial ability. SPACE (Ramakrishnan et al., 2024) categorizes spatial tasks into large-scale and small-scale cognition. Large-scale tasks assess environment-level spatial understanding, while small-scale tasks involve object-level reasoning. However, the object-level data is limited to 2D synthetic shapes, lacking real-world 3D variability and complexity.

In contrast, our benchmark is cognitively grounded and systematically progresses from small-scale to large-scale spatial reasoning tasks. We start from core perceptual challenges (e.g., object identity, canonical view recognition(single object), mental rotation(single object), dynamic translation/rotation(single object)) and scale up to relational and perspective-taking tasks in complex multi-object scenes. Our tasks are carefully designed to isolate spatial reasoning by controlling for distractors, motion, reference frame shifts, and multi-image input. The use of both real-world and photo-realistic synthetic data enables robust and interpretable evaluations. Our perspective-taking task serves as the most challenging task, requiring integrating of all subskills, making it a holistic test of spatial cognition. Existing benchmarks lack this layered structure and often conflate spatial understanding with unrelated reasoning skills.

### F.2    SPATIAL REASONING MODELS

One line of work enhances VLMs' spatial reasoning by leveraging explicit 3D abstractions of scenes. SpatialReasoner (Ma et al., 2025b) introduces a large vision-language model that incorporates 3D representations such as object locations and orientations to enable coherent and reliable reasoning. Similarly, Abstract Perspective Change (APC) (Lee et al., 2025) constructs perspective-aware scene abstractions using vision foundation models for object detection, segmentation, and orientation estimation, leading to significant improvements in perspective reasoning. SSR (Liu et al., 2025) transforms raw depth data into structured, interpretable textual rationales to be integrated in VLMs. Another direction relies on continued pre-training and reinforcement learning post-training. MetaSpatial (Pan & Liu, 2025) adopts a reinforcement learning framework to iteratively refine scene layouts with physics-aware constraints, generating coherent and realistic 3D arrangements without supervised annotations. SpatialVLM (Chen et al., 2024) introduces large-scale synthetic pre-training

data to equip models with quantitative 3D spatial reasoning, enabling accurate metric distance estimation and downstream improvements in VQA and robotics. Embodied-R (Zhao et al., 2025) combines large-scale VLMs and LMs in an RL framework that integrates embodied reasoning from video streams, using both fast and slow iterative processes to tackle diverse indoor and outdoor tasks. vsGRPO-7B (Liao et al., 2025) employs R1-Zero-like training with GRPO to boost visual-spatial reasoning, outperforming baselines and even surpassing GPT-4o on video-based benchmarks. SpaceR (Ouyang et al., 2025) proposes the SpaceR-151k dataset alongside a spatially-guided RLVR strategy (SG-RLVR), achieving state-of-the-art results and surpassing GPT-4o by 11.6% on VSI-Bench. Likewise, SVQA-R1 (Wang & Ling, 2025) extends R1-style reinforcement learning to spatial VQA through Spatial-GRPO, improving accuracy and interpretability without reliance on supervised fine-tuning. More recent efforts such as SpaceOm and Spacethinker (Chen et al., 2025a) attempt to enhance spatial reasoning through RL-driven linguistic fine-tuning, but their improvements exhibit limited generalization Yin et al. (2025), leaving fundamental questions about VLMs' spatial cognition unresolved. Ultimately, these works underscore that linguistic reasoning alone is insufficient (Zhang et al., 2025b); humans understand physical space through structured reasoning that does not always translate into words, highlighting the need for models that reason beyond language.

## G  THE USE OF LARGE LANGUAGE MODELS (LLMS)

Large language models were used only as general-purpose tools to assist with writing clarity and grammar refinement. All technical contributions, benchmark design, and evaluations were developed entirely by the authors themselves.

