# OpenReview forum: "SpinBench: Perspective and Rotation as a Lens on Spatial Reasoning in VLMs"
_ICLR.cc/2026/Conference — ICLR 2026 Poster_

### Official Review · Reviewer_fu7D · 2025-10-25

**Soundness:** 3
**Presentation:** 3
**Contribution:** 3
**Rating:** 8
**Confidence:** 3

**Summary:**

This paper introduces SpinBench, a high-quality diagnostic benchmark for evaluating spatial reasoning in VLMs, with a focus on perspective-taking and mental rotation. Its design is cognitively-grounded and features a progressive structure that isolates specific cognitive abilities. Extensive experiments uncover systematic failures like egocentric bias and inconsistent reasoning, along with detailed discussions from various aspects. The benchmark's difficulty is validated against human studies, showing a strong correlation between VLM accuracy and human response times.

**Strengths:**

1. Benchmark design. SpinBench is cognitively-grounded and diagnostically powerful with its progressive structure (from simple perception to complex perspective-taking) and controlled variations (e.g., consistency checks, frame-of-reference shifts), allowing for precise identification of model failures.
2. Comprehensive and insightful discussion. The large-scale evaluation of various VLMs along with the analysis yields significant findings in various aspects, including the exposure of systematic biases (egocentrism), inconsistent reasoning, and distinct scaling patterns (emergent vs. smooth) across different spatial skills.
3. Strong validation with human studies. The strong correlation between VLM accuracy and human response time provides compelling evidence that the benchmark measures genuine cognitive difficulty, justifying its credibility.

**Weaknesses:**

1. Lack of architectural analysis. While the evaluation is extensive, the paper misses an opportunity to connect observed performance differences to specific architectural choices in the VLMs (e.g., vision encoders, fusion methods). Such analysis could provide more direct guidance for future model design. (This is minor. )

**Questions:**

1. In your scaling law analysis (Figure 6), you show that Identity Matching exhibits a sharp emergent capability, whereas Object Relation Grounding improves smoothly. This is a fascinating contrast. What do you believe is the underlying reason for this difference? Does it imply that cross-view object abstraction is a fundamentally different and more complex type of computation that only "activates" at a certain scale, compared to in-scene relation extraction?
2. Have you considered evaluating the impact of different image resolutions or aspect ratios on model performance? Given that spatial relations can be sensitive to visual details and object boundaries, it would be interesting to know how robust these models are to such perceptual variations.

Missing . in L288

---

> ### Author Response · Authors · 2025-11-19
> **Response to Reviewer fu7D**
>
> We thank the reviewer for the insightful assessment of our work. We appreciate the recognition of SpinBench’s cognitively grounded design, its diagnostic structure, and the depth of our analysis. We also thank the reviewer for the constructive suggestions regarding architectural factors and input variations. We address these points below and provide additional experiments and clarifications in response to the reviewer’s questions. We look forward to the reviewer’s response and further discussion.
>
> > **W1:**  Lack of architectural analysis. While the evaluation is extensive, the paper misses an opportunity to connect observed performance differences to specific architectural choices in the VLMs (e.g., vision encoders, fusion methods). Such analysis could provide more direct guidance for future model design. (This is minor. )
>
> **Response to W1**: We appreciate the reviewer’s suggestion. While our benchmark covers a diverse set of model families, with different vision encoders and fusion mechanisms (e.g., InternVL’s early-fusion projector design, Qwen-VL’s cascaded architecture), drawing firm conclusions would require controlled architectural ablations. Such an in-depth architectural study is beyond the scope of this work. We view systematic architectural investigation as an important direction for future work, and SpinBench provides the diagnostic foundation needed for such studies.
>
>
> > **Q1:** In your scaling law analysis (Figure 6), you show that Identity Matching exhibits a sharp emergent capability, whereas Object Relation Grounding improves smoothly. This is a fascinating contrast. What do you believe is the underlying reason for this difference? Does it imply that cross-view object abstraction is a fundamentally different and more complex type of computation that only "activates" at a certain scale, compared to in-scene relation extraction?
>
> **Response to Q1**: We thank the reviewer for the thoughtful question. While we cannot draw a definitive conclusion, one possible interpretation is that the two tasks differ in context length. Identity Matching and Dynamic Translation require integrating information across multiple images (equal or more than 2), which creates a longer and more demanding context for the model to maintain and compare, thus appearing to only "activates" at a certain scale
>
> > **Q2:** Have you considered evaluating the impact of different image resolutions or aspect ratios on model performance? Given that spatial relations can be sensitive to visual details and object boundaries, it would be interesting to know how robust these models are to such perceptual variations
>
> **Response to Q2**: We thank the reviewer for this insightful suggestion.  We evaluated a high-resolution setting in which we used the original images instead of the downsampled version used in the main experiments (max size 256). Despite nearly 10× more pixels, we observe no systematic accuracy gains. This is consistent with the fact that most VLM backbones aggressively downsample early in the visual encoder and were pre-trained at relatively modest resolutions; beyond a certain point, extra pixels provide limited benefit without corresponding changes in architecture or training. Detailed resutls and analysis can be found in the updated manuscript, Figure 6.

---

### Official Review · Reviewer_cvAh · 2025-10-31

**Soundness:** 4
**Presentation:** 4
**Contribution:** 3
**Rating:** 8
**Confidence:** 3

**Summary:**

This paper proposes SpinBench, a cognitively grounded benchmark designed to evaluate spatial reasoning in vision-language models, focusing on perspective taking and viewpoint transformation. It includes seven structured task types across 2,599 samples and 51 tasks. Evaluating 37 VLMs, the study finds consistent weaknesses such as egocentric bias, poor rotation understanding, and low consistency under symmetry and phrasing changes. Human-model correlations further confirm that SpinBench effectively captures genuine spatial reasoning challenges.

**Strengths:**

1. SpinBench offers a standardized, fine-grained, and vision-centric framework for analyzing the spatial reasoning capabilities of existing methods. Unlike previous benchmarks that primarily focus on semantic recognition, SpinBench introduces tasks such as perspective taking to effectively evaluate whether models can comprehend camera movements and exhibit true spatial modeling ability.
2. The experiments conducted on 37 VLMs are comprehensive and include an in-depth analysis of their performance.
3. The paper is well-written and easy-to-follow. The task definitions are reasonably motivated.

**Weaknesses:**

1. Although SpinBench includes a diverse range of question types, its overall data scale is relatively small compared to existing spatial reasoning datasets such as VSI-Bench and ViewSpatial-Bench.
2.  SpinBench primarily evaluates VLMs using image inputs. A valuable extension would be to include baselines where VLMs take both images and their corresponding depth maps as input (e.g., generated by DepthAnything). This addition would help determine whether poor model performance stems from insufficient spatial information in the input data or from the model’s limited spatial reasoning ability.

**Questions:**

I have no more questions to the author. The paper clarifies most benchmark details well.

**Details Of Ethics Concerns:**

None.

---

> ### Author Response · Authors · 2025-11-19
> **Response to Reviewer cvAh**
>
> We thank the reviewer for the thoughful comments and for highlighting SpinBench’s strengths in task design, analysis depth, and clarity. We appreciate the constructive feedback and address these points below. We look forward to the reviewer’s response and further discussion.
>
> > **W1:**  Although SpinBench includes a diverse range of question types, its overall data scale is relatively small compared to existing spatial reasoning datasets such as VSI-Bench and ViewSpatial-Bench.
>
> **Response to W1**: We appreciate the reviewer’s concern regarding the overall dataset size. While SpinBench is smaller in scale than broad spatial reasoning datasets such as VSI-Bench, which draws from ScanNet, ScanNet++, and ARKitScenes, and ViewSpatial-Bench, which leverages large 3D scene reconstructions from ScanNet and keypoint-annotated images from MS-COCO, the goals of these datasets differ substantially from ours.
> SpinBench is intentionally designed as a diagnostic benchmark rather than a large-scale scene understanding dataset. Our focus is on controlled, cognitively grounded variations that isolate specific spatial abilities. This requires tightly controlled image sets that cannot be obtained from large in-the-wild datasets.
>
>
> > **W2:** SpinBench primarily evaluates VLMs using image inputs. A valuable extension would be to include baselines where VLMs take both images and their corresponding depth maps as input (e.g., generated by DepthAnything). This addition would help determine whether poor model performance stems from insufficient spatial information in the input data or from the model’s limited spatial reasoning ability.
>
> **Response to W2**: We thank the reviewer for this insightful suggestion. For each test image, we generated a depth map using DepthAnything and horizontally concatenated it with the original RGB image, keeping the number of input images fixed. The depth map had the same spatial resolution as the RGB image. Across models, this consistently reduced accuracy. A plausible explanation is that current VLMs are heavily tuned to natural RGB images; simply appending a depth map changes both the distribution and geometry of the input, without any architectural adaptation or fine-tuning, so the models may treat depth as noise rather than a structured cue. These results suggest that naively adding depth is not sufficient. More principled depth integration (e.g., late fusion) is likely needed to realize the potential benefits of depth information.
> Detailed results and analysis can be found in the updated manuscript, Figure 6.

---

### Official Review · Reviewer_SReC · 2025-11-01

**Soundness:** 2
**Presentation:** 2
**Contribution:** 2
**Rating:** 4
**Confidence:** 3

**Summary:**

The authors introduce SpinBench, a spatial reasoning benchmark for VLMs. SpinBench measures the perspective-taking abilities of VLMs through seven tasks, six subtasks representing skills required to accomplish the seventh perspective-taking task. They conducted extensive experiments on their benchmark using 37 VLMs, finding that VLMs exhibit egocentric bias in spatial reasoning and showing that the difficulty of each SpinBench problem correlates with human difficulty in solving the same problem.

**Strengths:**

* The paper provides a comprehensive evaluation of 37 VLMs of varying sizes, including two reasoning models.

* The paper shows that VLMs fail in perspective-taking when performing spatial reasoning tasks.

* The consistency metric reveals an interesting spatial reasoning bias in VLMs, and the connection between consistency and spatial reasoning accuracy is interesting.

* A detailed and comprehensive description of the construction of the benchmark is included.

**Weaknesses:**

* While the observation that VLMs exhibit egocentric bias in spatial reasoning is interesting, the general observation that VLMs exhibit egocentric bias is not a novel observation, as observed by [1,2].

* Related to the previous point, it is unclear what is gained by developing the seven categories used in SpinBench. The authors motivate the seven categories based on six basic spatial reasoning abilities which scaffold towards perspective-taking. However, it is unclear what useful conclusions can be drawn by devising the benchmark in this way. Experiments showing that improving the six simpler diagnostic abilities leads to better perspective-taking ability would emphasize the usefulness and necessity of the benchmark. As it stands, I am not convinced of the practical usefulness of SpinBench.

* The results on CoT reasoning are inconclusive: Cosmos-Reason1-7B significantly benefits from CoT, whereas SpaceOm only enjoys moderate benefits, and SpaceThinker shows degraded performance on perspective-taking without premise.

* While the proposed benchmark shows some interesting observations and correlations, my main concern is that the benchmark does not indicate the direction of causality: the six simpler reasoning tasks may lead to improved perspective-taking abilities or perspective-taking abilities may lead to improved performance on each spatial reasoning task. Fine-tuning the model based on the six simpler tasks and seeing if it leads to improved perspective-taking would show a useful and more interesting observation which can be used to improve existing VLMs.

Minor Weaknesses:

* In Figure 2, the legend “Static Inter-Object” does not match the description “Object-Relation Grounding” in the text.

---

[1] Seeing through their eyes: Evaluating visual perspective taking in vision language models, Góral et al., 2024

[2] 3DSRBench: A Comprehensive 3D Spatial Reasoning Benchmark, Wufei et al., ICCV 2025

**Questions:**

* What could be the reasoning behind the inconsistent impact of CoT reasoning: Cosmos-Reason1-7B and SpaceOm show improved performance whereas SpaceOm shows degraded performance on perspective-taking.

* In Figure 4 and L317-323, the authors show that consistency is strongly correlated with accuracy. However, this correlation may only imply that strong spatial reasoning leads to strong consistency which is obvious. It would be more interesting if the other direction were true: strong consistency leads to stronger spatial reasoning. Is there evidence on whether stronger consistency leads to stronger spatial reasoning?

* In Figure 3, most models perform much better on the car_view_back task than the other canonical view selection tasks. Is there any possible explanation why this task is particular easy for most VLMs?

* Why do the mental rotation tasks consist of only images rendered from ABO Objects?

**Details Of Ethics Concerns:**

No concerns

---

> ### Author Response · Authors · 2025-11-19
> **Response to Reviewer SReC (1/4)**
>
> We thank the reviewer for the thoughtful and detailed assessment of our work. We appreciate the recognition of our comprehensive model evaluation, the insights on spatial reasoning consistency, and the clarity of our benchmark construction. We also value the reviewer’s concerns regarding the novelty of the egocentric bias, the motivation behind the seven diagnostic categories, the mixed CoT effects, and the question of causal direction. We address each of these points in detail below.
>
> > **W1:**  While the observation that VLMs exhibit egocentric bias in spatial reasoning is interesting, the general observation that VLMs exhibit egocentric bias is not a novel observation, as observed by [1,2].
>
> **Response to W1**: We thank the reviewer for raising this problem. We agree that prior work has noted that VLMs may fall default to egocentric interpretations. However, we would like to clarify that our results provide new and more direct evidence that goes beyond this general observation in three important ways:
>
> First, our evaluation uses paired, controlled comparisons that prior work does not.  Earlier studies (e.g., [1,2]) suggest that allocentric reasoning is harder than egocentric reasoning and propose new allocentric tasks, but they do **not** evaluate allocentric and egocentric interpretations in *paired form* with the **same image**.  In SpinBench, each allocentric–egocentric question pair uses **identical visual input**, differing only in the reference-frame specification in the query. This controlled variation removes perceptual confounds and directly isolates reference-frame reasoning. The persistence of bias under these controlled conditions reveals a deeper inductive preference than previously shown.
>
> Second, thanks to our paired evaluation, our analysis shows that models performing best on the egocentric task perform worst on the allocentric variant (Table 3). This strong asymmetry across multiple models has not been reported before.
>
> Third,  GPT-5 performs strongly on both egocentric and allocentric tasks (κ = 0.55 allocentric, 0.78 accuracy), demonstrating balanced reference-frame reasoning that earlier models lack. This is, to our knowledge, the first evidence of a foundation model exhibiting meaningful allocentric competence.
>
> Taken together, SpinBench provides the first systematic and reference-frame explicit analysis of egocentric bias, revealing (1) strong cross-model asymmetries, (2) bias persistence under paired, controlled evaluations, and (3) improved allocentric capability in the newest GPT models.

---

> ### Author Response · Authors · 2025-11-19
> **Response to Reviewer SReC (2/4)**
>
> > **W2:**  Related to the previous point, it is unclear what is gained by developing the seven categories used in SpinBench. The authors motivate the seven categories based on six basic spatial reasoning abilities which scaffold towards perspective-taking. However, it is unclear what useful conclusions can be drawn by devising the benchmark in this way. Experiments showing that improving the six simpler diagnostic abilities leads to better perspective-taking ability would emphasize the usefulness and necessity of the benchmark. As it stands, I am not convinced of the practical usefulness of SpinBench.
>
> **Response to W2**: We appreciate the reviewer’s concern about the motivation for the seven categories. SpinBench’s goal is not only to measure perspective-taking but to diagnose how models fail. The decomposition isolates which core spatial abilities limit viewpoint reasoning.
>
> To directly test the value of this scaffolding, we fine-tuned Qwen3-VL-4B and Qwen3-VL-8B on 5,900 newly curated samples covering four core spatial reasoning abilities (mental rotation, canonical view selection, identity matching, and dynamic rotation). The training domain includes object and face images, whereas the test domain consists of Infinigen-rendered 3D scenes. Crucially, the test set comes from the perspective-taking task in the original SpinBench, which is not included in training. This design jointly tests task-type generalization and domain generalization (from object/face imagery to synthetic 3D scenes)
>
> Both models are fine-tuned for 6 epochs using GRPO, and the training/validation accuracy rewards are reported in the updated Appendix E. As summarized below, both models improve on the held-out perspective-taking task, demonstrating transfer from foundational spatial skills to viewpoint transformation. The 8B model shows a notably larger improvement
> | Model                     | Initial Val Accuracy Reward | Highest Val Accuracy Reward |
> |---------------------------|-----------------------------|-----------------------------|
> | **Qwen3-VL-4B-Instruct** | 0.344                       | 0.389                       |
> | **Qwen3-VL-8B-Instruct** | 0.302                       | 0.398                       |
>
>
> This demonstrates that strengthening basic spatial abilities transfers to viewpoint transformation, supporting the usefulness of the seven-category structure.
> Importantly, all other reviewers independently highlighted that this progressive, cognitively grounded structure is a core strength of SpinBench:
>
> Reviewer t7Ct: “... provides strong validation for the benchmark’s design, suggesting it measures a fundamental and shared aspect of spatial reasoning difficulty.”
>
> Reviewer kk9w: “... novel in its cognitively grounded approach, systematically decomposing the complex, high-level skill of perspective taking into a progressive hierarchy ...”
>
> Reviewer cvAh: “The task definitions are reasonably motivated.”
>
> Reviewer fu7D: “... diagnostically powerful with its progressive structure and controlled variations, allowing precise identification of model failures.”*
>
> Taken together, we show that the seven diagnostic categories are not arbitrary. Rather, they provide a novel structure to interpret model behavior and expose the underlying sources of perspective-taking errors
>
>
>
>
>
> > **W3:**   The results on CoT reasoning are inconclusive: Cosmos-Reason1-7B significantly benefits from CoT, whereas SpaceOm only enjoys moderate benefits, and SpaceThinker shows degraded performance on perspective-taking without premise.
>
> **Response to W3**: We appreciate the reviewer’s observation and would like to clarify our interpretation. CoT appears beneficial when the model already has reliable visual grounding and produces a reasoning chain that remains consistent with the underlying spatial structure. When either step is uncertain, such as misinterpreting a spatial cue or introducing an incorrect reasoning chain, CoT does not reliably help and can sometimes degrade performance. Our intention is not to claim that CoT is effective or ineffective in general, but rather that **CoT alone is not sufficient** for spatial reasoning tasks.
>
> This interpretation is also consistent with recent work exploring reasoning directly within the visual embedding space[1][2]. These approaches suggest that visually grounded reasoning may complement textual CoT in settings where spatial grounding plays a central role.
>
> References:
> [1] Empower Multimodal Reasoning with Latent Visual Tokens
>
> [2] Latent Visual Reasoning

---

> ### Author Response · Authors · 2025-11-19
> **Response to Reviewer SReC (3/4)**
>
> > **W4:**  While the proposed benchmark shows some interesting observations and correlations, my main concern is that the benchmark does not indicate the direction of causality: the six simpler reasoning tasks may lead to improved perspective-taking abilities or perspective-taking abilities may lead to improved performance on each spatial reasoning task. Fine-tuning the model based on the six simpler tasks and seeing if it leads to improved perspective-taking would show a useful and more interesting observation which can be used to improve existing VLMs.
>
> **Response to W4**: We appreciate the reviewer’s concern about the direction of causality between the six foundational spatial abilities and perspective-taking. To address this, we performed the intervention experiment the reviewer suggested (see our response to W2 and Appendix E).
> In this experiment, we fine-tuned Qwen3-VL-4B and Qwen3-VL-8B solely on the simpler spatial reasoning tasks (mental rotation, canonical-view selection, identity matching, and dynamic rotation), without using any perspective-taking data. We then evaluated both models on the perspective-taking task from SpinBench, which involves a different task type and a different visual domain (Infinigen). Both models improve on the perspective-taking task despite never seeing it during training. Thus, SpinBench’s decomposition is not arbitrary, it identifies precursor abilities that can be targeted to enhance complex spatial reasoning.
>
>
> > **Q1:** What could be the reasoning behind the inconsistent impact of CoT reasoning: Cosmos-Reason1-7B and SpaceOm show improved performance whereas SpaceOm shows degraded performance on perspective-taking.
>
> **Response to Q1**: As discussed in our response to W3, the mixed impact of CoT can be interpreted through the interaction between visual grounding quality and the correctness of the reasoning steps. Cosmos-Reason1-7B likely benefits from stronger visual representations and more consistent reasoning. In contrast, SpaceOm is a 3B model, smaller models may have weaker perceptual grounding and reasoning capability.
>
> > **Q2:** In Figure 4 and L317-323, the authors show that consistency is strongly correlated with accuracy. However, this correlation may only imply that strong spatial reasoning leads to strong consistency which is obvious. It would be more interesting if the other direction were true: strong consistency leads to stronger spatial reasoning. Is there evidence on whether stronger consistency leads to stronger spatial reasoning?
>
> **Response to Q2**: We agree that a high correlation between accuracy and consistency can be interpreted as strong spatial reasoning produces strong consistency. However, with more top models evaluated (GPT-5, o4-mini, gemini 2.5 pro), our results also provide new evidence: stronger consistency does not necessarily guarantee stronger spatial reasoning, especially among high-performing models.
>
> First, while consistency and accuracy are strongly correlated overall, the relationship becomes non-linear at the top right corner. As shown in Figure 4, top models exhibit very high consistency scores clustered above 90%, yet their accuracies diverge noticeably. For example, InternVL3-38B is the second most consistent model but lags behind gemini 2.5 pro in overall accuracy by 2.6 percentage points, despite gemini’s lower consistency ranking (fifth), gemini 2.5 pro achieves the second highest overall accuracy. This non-linearity at high performance levels shows that after a model becomes highly consistent, spatial reasoning quality is further determined by whether the model consistently chooses the correct answer, not merely by maintaining logical equivalence across reformulated queries.
>
> Second, models with lower accuracy show a much tighter linear relationship between accuracy and consistency. This suggests that gains in consistency meaningfully contribute to spatial reasoning ability primarily when the model is still weak, but as models improve, the bottleneck shifts from consistency to making the correct consistent choice. In other words, consistency is a necessary early step, but not a sufficient indicator of advanced spatial reasoning.
>
> Together, these findings show that:
> 1. Stronger spatial reasoning tends to produce stronger consistency (expected), but stronger consistency does not reliably imply stronger spatial reasoning, especially among already-consistent models.
> 2. Accuracy requires both high consistency and correctness, and the latter becomes the limiting factor at higher performance levels.
>
> Thus, our results provide empirical evidence that consistency is helpful but does not independently determine spatial reasoning strength, and highlight the difference between random behavior and systematically incorrect but consistent behavior. A detailed analysis of 16 consistency patterns, including all correct and all wrong, which further illustrates this distinction, is provided in Appendix, Figure 35.

---

> ### Author Response · Authors · 2025-11-19
> **Response to Reviewer SReC (4/4)**
>
> > **Q3:** In Figure 3, most models perform much better on the car_view_back task than the other canonical view selection tasks. Is there any possible explanation why this task is particular easy for most VLMs?
>
> **Response to Q3**: A likely explanation is that the back view of a car has highly distinctive visual features compared to left, right. These features are more stable across car models and viewing conditions, making the back view uniquely identifiable. In contrast left and right view difference is subtle.
>
> > **Q4:** Why do the mental rotation tasks consist of only images rendered from ABO Objects?
>
> **Response to Q4**: We thank the reviewer for the helpful suggestion. To ensure sufficient domain diversity, we expanded the mental rotation data by adding 140 new samples drawn from two additional domains: Car and Infinigen. The full updated results are included in the revised manuscript.

---

> ### Comment · Reviewer_SReC · 2025-11-26
>
> I'd like to thank the authors for providing a detailed response to each of the raised concerns. Most of my concerns regarding SpinBench is now resolved, and I raise my rating to **6**.
>
> One remaining question is: is there a specific reason that the authors decided to fine-tune the VLM using GRPO instead of  supervised fine-tuning (SFT)? In recent benchmark works for assessing the spatial reasoning of VLMs, including RoboSpatial [1], MindCube [2], OmniSpatial [3], SPAR-Bench [4] and more, it seems like standard practice to show the effectiveness of the benchmark by applying SFT and reporting the results.
>
> ---
>
> [1] RoboSpatial: Teaching Spatial Understanding to 2D and 3D Vision-Language Models for Robotics, Song et al., CVPR 2025
>
> [2] Spatial Mental Modeling from Limited Views, Yin et al., 2025
>
> [3] OmniSpatial: Towards Comprehensive Spatial Reasoning Benchmark for Vision Language Models, Jia et al., 2025
>
> [4] From Flatland to Space: Teaching Vision-Language Models to Perceive and Reason in 3D, Zhang et al., NeurIPS 2025

---

> ### Author Response · Authors · 2025-11-26
>
> Thank you for the follow-up question and for raising the score! We’re glad to hear that the earlier concerns have been resolved.
>
> Below, we clarify why we chose GRPO rather than SFT for this specific intervention experiment:
>
> - For spatial reasoning, producing high-quality reasoning traces is non-trivial. As noted in MindCube (Spatial Mental Modeling from Limited View), effective SFT relies on carefully curated cognitive maps and reasoning chains. Human-written or template-based traces often introduce rigid or suboptimal reasoning patterns. Recent analysis in “SFT or RL?” [1] further shows that SFT helps to learn the reasoning format but imitative, rigid reasoning can impede downstream reasoning improvements. Since curating such reasoning traces is not the focus of our work, we avoided introducing this additional source of variability.
>
> - MindCube also shows that RL from scratch, relying only on accuracy rewards alone can already improve spatial reasoning, even though they are not sufficient for optimal performance. This supports our choice of GRPO: RL can provide meaningful improvements without requiring hand-crafted reasoning demonstrations. This allows us to validate our scaffold design without introducing extra variables.
>
> - GRPO only requires an accuracy reward. Using GRPO removes the need to design or curate any reasoning traces. This keeps the experiment focused on the core question: does improving the six foundational skills improve perspective-taking?
>
> - The chosen models (Qwen3-VL-4B-Instruct, Qwen3-VL-8B-Instruct) have strong instruction-following ability, the reasoning prompt together with format reward in GRPO allows models to produce structured reasoning during training even without SFT.
>
> - The purpose of the experiment is to test whether training on the six foundational tasks causally improves perspective-taking, but not to explore optimal post-training strategies. GRPO serves this purpose cleanly, without introducing extra confounding factors (curated reasoning traces in SFT).
>
> [1] SFT or RL? An Early Investigation into Training R1-Like Reasoning Large Vision-Language Models
>
> We hope this answers your question, and we would be very happy to continue the discussion if you have further questions. Thank you again for your time and thoughtful comments!

---

> > ### Comment · Reviewer_SReC · 2025-11-27
> >
> > Thank you for the detailed response. My concerns are now fully addressed.

---

> > > ### Author Response · Authors · 2025-11-27
> > >
> > > We sincerely appreciate your thoughtful comments and the improvements they inspired. Thank you again for your valuable feedback!

---

### Official Review · Reviewer_kk9w · 2025-11-08

**Soundness:** 2
**Presentation:** 4
**Contribution:** 3
**Rating:** 6
**Confidence:** 4

**Summary:**

The paper introduces SpinBench, a new diagnostic benchmark for evaluating spatial reasoning in VLMs. The benchmark is cognitively grounded, designed around the core challenge of perspective-taking, which is the ability to reason about how scenes and object relations change under viewpoint transformations. SpinBench decomposes this complex skill into seven fine-grained task categories. These tasks are progressively structured, scaffolding from simpler single-object challenges to more demanding multi-object perspective-taking scenarios. A key contribution is the use of controlled variations to probe model robustness. The authors evaluate 37 state-of-the-art VLMs. The results reveal systematic weaknesses in current models, such as a strong egocentric bias, a poor understanding of rotation, and high levels of logical inconsistency. The study also includes a human baseline (91.2% accuracy) and finds that task difficulty for humans strongly correlates with VLM accuracy, validating the benchmark's focus on fundamental cognitive challenges.

**Strengths:**

+ While many spatial reasoning benchmarks exist, SpinBench is novel in its cognitively grounded approach, systematically decomposing the complex, high-level skill of perspective taking into a progressive hierarchy of 7 diagnostic sub-skills. The explicit testing of allocentric vs. egocentric frames, visual grounding vs. linguistic inference, and logical consistency provides a multi-dimensional, surgical tool for diagnosing model failures, which is a unique combination not present in prior work.

+ The benchmark is well-motivated by foundational work in cognitive science. The data curation is thoughtful, drawing from four diverse domains (synthetic indoor scenes, real-world objects, cars, and faces).

+ The findings are specific, insightful, and actionable: the discovery of a strong egocentric bias, the fact that models fail even at purely linguistic spatial reasoning, and the high degree of logical inconsistency all point to fundamental gaps in current models.

+ The paper is overall well presented and clearly formulates the motivation.

**Weaknesses:**

+ The benchmark only contains 2599 samples in total, distributed across 51 distinct task subtypes. This distribution means that some of the fine-grained subtypes have a very small number of samples. The reliability of the performance metrics on such small test sets is questionable. A larger sample size for these specific subtypes would strengthen the conclusions. For example, the Mental Rotation task is presented as a key cognitive inspiration for the benchmark in the introduction. However, this category seems underdeveloped compared to the others. It contains only 78 samples.

+ Despite being evaluated with diverse VLMs, it remains unclear what the upper bound performance is on SpinBench for existing VLMs, that is, SOTA models like GPT-5 or Gemini 2.5 Pro are not tested.

**Questions:**

1. I am concerned about the low sample size for some of the 51 subtypes. How did you ensure the statistical reliability of the results for these specific subtypes?

2. The scaling analysis in Figure 6 is interesting, particularly the emergent capability in Identity Matching. Did you observe similar emergent patterns in the most complex tasks, like Perspective Taking and Mental Rotation? Or did performance on those tasks scale more smoothly, even if starting from a very low, chance-level baseline?

---

> ### Author Response · Authors · 2025-11-19
> **Response to Reviewer kk9w (1/2)**
>
> We thank the reviewer for the thoughtful comment and for recognizing SpinBench’s cognitively grounded design and structured decomposition of perspective-taking. We appreciate the acknowledgment of our motivation and findings, as well as the constructive concerns regarding small subtype sample sizes and the inclusion of additional SOTA models. We address these points in the following responses, and we look forward to further discussion.
>
>
> > **W1:** The benchmark only contains 2599 samples in total, distributed across 51 distinct task subtypes. This distribution means that some of the fine-grained subtypes have a very small number of samples. The reliability of the performance metrics on such small test sets is questionable. A larger sample size for these specific subtypes would strengthen the conclusions. For example, the Mental Rotation task is presented as a key cognitive inspiration for the benchmark in the introduction. However, this category seems underdeveloped compared to the others. It contains only 78 samples.
>
> **Response to W1**: We thank the reviewer for the helpful suggestion. To ensure sufficient domain diversity and increase the sample size for the Mental Rotation task, we expanded this portion of the benchmark by adding 140 new samples drawn from two additional domains: Car (20 items) and Infinigen (120 items). The full updated results are included in the revised manuscript.
> A more detailed discussion of sample size and statistical reliability is provided later in our response to Q1.
>
> > **W2:** Despite being evaluated with diverse VLMs, it remains unclear what the upper bound performance is on SpinBench for existing VLMs, that is, SOTA models like GPT-5 or Gemini 2.5 Pro are not tested.
>
> **Response to W2**: We thank the reviewer for their thoughtful suggestion. In the revised manuscript, we have expanded our evaluation to include **GPT-5**, **o4-mini**, and **Gemini 2.5 Pro**. The updated evaluation results and updated analysis can be found in the revised manuscript.
>
> > **Q1:** I am concerned about the low sample size for some of the 51 subtypes. How did you ensure the statistical reliability of the results for these specific subtypes?
>
> **Response to Q1**: We thank the reviewer for their concern about the reliability of results for small subtypes. We clarify our approach as follows:
> Statistically, reliability for accuracy estimates depends on the variance and confidence interval. Each trial is a Bernoulli variable with variance p(1-p). The variance of the sample mean across \(n\) trials is therefore $\frac{p(1-p)}{n}$, giving a standard error $std = \sqrt{\frac{p(1-p)}{n}}$. A 95\% confidence interval is then $p \pm z_{0.975} \cdot std$, where $z_{0.975}$ is the 97.5th percentile of the standard normal distribution. Reliable conclusions therefore require conditions with sufficiently large effective sample sizes. For example, with p = 0.5 and n = 200, the standard error is $std = \sqrt{\frac{0.5(1-0.5)}{200}} \approx 0.035$.
>  SpinBench follows these principles. Although we provide 51 labeled subtypes for transparency, many distinctions reflect controlled variations that are **not interpreted independently**, such as domain (car/object/face/Infinigen), prompting style (image-first vs. interleaved), or directional variation (left/right/back). Our analysis aggregates these into the 7 core diagnostic task families, each with roughly 150–600 samples. These larger groupings yield tight confidence intervals and stable accuracy estimates, comparable to prior spatial reasoning benchmarks.
>
> Furthermore, very small subtypes, such as the Infinigen full-occlusion category, are never interpreted in isolation. Instead, they are pooled with related conditions (e.g., partial + full occlusion) to form statistically reliable contrasts such as occlusion vs. no occlusion. This ensures that all reported results are grounded in sample sizes large enough to produce robust and interpretable estimates.

---

> ### Author Response · Authors · 2025-11-19
> **Response to Reviewer kk9w (2/2)**
>
> > **Q2:** The scaling analysis in Figure 6 is interesting, particularly the emergent capability in Identity Matching. Did you observe similar emergent patterns in the most complex tasks, like Perspective Taking and Mental Rotation? Or did performance on those tasks scale more smoothly, even if starting from a very low, chance-level baseline
>
> **Response to Q2**: Thank you for the insightful question. In contrast to the clear scaling behavior we observed in tasks such as Identity Matching, Object Relation Grounding, and Dynamic Translation, we did **not** observe smooth or emergent scaling patterns for the more complex tasks. Across the Gemma 3, InternVL 3, InternVL 3.5, and Qwen2.5-VL model families, the largest models perform at or near chance, with accuracy indistinguishable from that of the smallest models in the family. For these tasks, scaling within a model family does not yield meaningful improvements.
> In summary, while some foundational spatial tasks exhibit emergent or smooth improvements with scale, the most cognitively demanding spatial tasks show no consistent scaling trend. This may be due to limitations in current model architectures or the distribution of training data, which do not adequately support more complex spatial reasoning.

---

### Official Review · Reviewer_t7Ct · 2025-11-09

**Soundness:** 2
**Presentation:** 2
**Contribution:** 1
**Rating:** 2
**Confidence:** 4

**Summary:**

The paper introduces SpinBench, a cognitively grounded benchmark for evaluating spatial reasoning in VLMs. It decomposes perspective-taking into seven diagnostic categories (e.g., identity matching, dynamic rotation, mental rotation). Evaluating 37 VLMs, it identifies systematic biases (egocentric preference, rotation failure, inconsistency under symmetry) and reports correlations between human response times and model accuracy, arguing that SPINBENCH captures cognitively meaningful difficulty.

**Strengths:**

1. **Testing 37 VLMs**—including proprietary and open-source—offers unusually broad coverage.

2. **Controlled Variations**: The introduction of controlled task variations—particularly the "with-premise" vs. "without-premise" conditions and the "allocentric vs. egocentric" reference frames—is a significant strength. These variations allow for a fine-grained disentanglement of visual grounding failures from spatial reasoning failures.

3. **Human-VLM Correlation**: The finding that human response time correlates strongly with VLM inaccuracy (Fig. 5b) provides strong validation for the benchmark's design, suggesting it measures a fundamental and shared aspect of spatial reasoning difficulty.

**Weaknesses:**

1. Prior work already targets spatial cognition broadly (SPACE), mental modeling (MindCube), and multi-perspective localization (ViewSpatial). Your unique bits are FoR-controlled augmentations and pairwise consistency, but the manuscript underplays direct, quantitative comparisons and ablations demonstrating why SPINBENCH exposes distinct phenomena beyond these. Provide head-to-head overlap/orthogonality analyses, cross-benchmark correlations, and case studies where SPACE/MindCube/ViewSpatial fail to reveal the same error mode.

2. Restricting all relations to a 2D horizontal plane undermines the generalizability of “spatial reasoning.” Without vertical or containment relations, the benchmark measures planar spatial alignment, not full 3D reasoning.

3. The model set omits several state-of-the-art commercial VLMs (e.g., GPT o3/5, Google Gemini 2.5 pro), which weakens the headline claim about “current VLMs.”  They have more robust and powerful ability.

**Questions:**

I am curious about the true value of the dataset, as I strongly suspect that it may merely overfit to its own format.

If a base model (e.g., Qwen-VL) could be fine-tuned on this dataset and subsequently demonstrate performance gains on other benchmarks (such as VSI-Bench, OmniSpatial and SPACE), I would be much more inclined to recognize the dataset’s contribution.

---

> ### Author Response · Authors · 2025-11-19
> **Response to Reviewer t7Ct (1/2)**
>
> We thank the reviewer for recognizing the strengths of our work, including the broad model evaluation, the fine-grained evaluation with our controlled task variations, and the cognitively validated design. We appreciate the reviewer’s suggestions and address the remaining concerns and questions below. We look forward to further discussion.
>
> 1. **Comparison to prior work and cross-benchmark correlations:**
> > **W1:** Prior work already targets spatial cognition broadly (SPACE), mental modeling (MindCube), and multi-perspective localization (ViewSpatial). Your unique bits are FoR-controlled augmentations and pairwise consistency, but the manuscript underplays direct, quantitative comparisons and ablations demonstrating why SPINBENCH exposes distinct phenomena beyond these. Provide head-to-head overlap/orthogonality analyses, cross-benchmark correlations, and case studies where SPACE/MindCube/ViewSpatial fail to reveal the same error mode.
>
> **Response to W1**:We appreciate the reviewer’s request for orthogonality analyses and cross-benchmark correlations. We would like to clarify that we have already **included** these analyses in Appendix Section B.3.
> As shown in the appendix Table 6, **overall correlations** between SpinBench and four established spatial reasoning benchmarks are uniformly weak and statistically non-significant:
>
> - with MindCube overall accuracy: *r* = –0.088, *p* = 0.836
> - with ViewSpatial-Bench overall accuracy: *r* = 0.460, *p* = 0.299
> - with OmniSpatial overall accuracy: *r* = 0.456, *p* = 0.137
> - with SpaCE-10 overall accuracy: *r* = 0.098, *p* = 0.803
>
> These weak global correlations directly indicate that SpinBench differs substantially from previous spatial reasoning benchmarks, and does not simply re-measure existing tests. Instead, SpinBench offers a more fine-grained evaluation of foundational spatial skills, which previous benchmarks do not target.
>
> The **subtask-level correlation matrix** in Appendix B.3 Figures 37, 38, 39, 40, further demonstrates this. Only a few tasks in previous benchmarks show correlation (p < 0.05) with specific tasks in SpinBench:
> - SpinBench Dynamic rotation ↔ MindCube 3D reasoning: *r* = 0.829, *p* = 0.021
> -  SpinBench Identity matching ↔ ViewSpatial person-based FoR: *r* = 0.915, *p* = 0.030
> -  SpinBench Static canonical-view reasoning ↔ OmniSpatial manipulation: *r* = 0.764, *p* = 0.006
> - spaCE-10 showed no significant subtask-level correlations.
>
>
> These correlation results confirm that while specific SpinBench tasks align with a few subcomponents of prior benchmarks, **the majority of SpinBench capabilities probe distinct reasoning mechanisms**.
> To improve clarity for readers, we have added a pointer in the main text to this appendix section.
>
>
> 2. **2D restricting:**
> > **W2:**  Restricting all relations to a 2D horizontal plane undermines the generalizability of “spatial reasoning.” Without vertical or containment relations, the benchmark measures planar spatial alignment, not full 3D reasoning.
>
> **Response to W2**: We agree that SpinBench does not currently cover vertical or containment relations, as we **have explicitly acknowledged in the limitation section** in our first submission. Here, we would like to clarify why our focus on 2D horizontal-plane relations is *intentional* and *appropriate* for the diagnostic purpose.
>
> First, many real-world embodied systems, including indoor navigation and autonomous driving, operate primarily on a 2D horizontal plane for planning (e.g., localization, obstacle avoidance, left/right decision making). These settings rely heavily on accurate viewpoint transformations, allocentric–egocentric frame shifts, and horizontal-plane spatial alignment. Thus, this subset of spatial reasoning is both highly relevant and critical in practice, despite being “planar.”
>
> Second, although the tasks are presented on a 2D plane, success still requires **3D** spatial understanding. All dynamic rotation, viewpoint-shift, and canonical-view tasks require models to infer 3D object orientation changes.
>
> Third, the reviewer’s concern highlights the important distinction between **diagnostic evaluation** and **comprehensive coverage**. SpinBench is not designed to enumerate every spatial relation (e.g., “in,” “on,” “above”). Instead, it isolates a tightly controlled subset of spatial reasoning tasks that we show are already challenging for current VLMs. Even within current coverage, models struggle with near-chance accuracy in some tasks. This motivates a diagnostic focus before expanding to a broader taxonomy of spatial skills.   Our focus on 2D horizontal-plane relations provides fine-grained probes that are otherwise obscured in more holistic tasks.
> In summary, SpinBench intentionally targets a core subset of 3D spatial reasoning tasks that are (1) ubiquitous in real-world embodied systems, (2) inherently 3D reasoning, and (3) challenging for current VLMs.

---

> ### Author Response · Authors · 2025-11-19
> **Response to Reviewer t7Ct (2/2)**
>
> 3. **Add more commercial VLMs:**
> > **W3:** The model set omits several state-of-the-art commercial VLMs (e.g., GPT o3/5, Google Gemini 2.5 pro), which weakens the headline claim about “current VLMs.” They have more robust and powerful ability.
>
> **Response to W3**: We thank the reviewer for their thoughtful suggestion to add recent state-of-the-art commercial VLMs. In the revised manuscript, we have expanded our evaluation to include **GPT-5** and **o4-mini**, which OpenAI describes as “our latest small o-series model.” We have also added evaluations for **Gemini 2.5 Pro**. The updated results can be found in the revised manuscript.
>
>
> 4. **Dataset value:**
> > **Q1:**  I am curious about the true value of the dataset, as I strongly suspect that it may merely overfit to its own format.
> If a base model (e.g., Qwen-VL) could be fine-tuned on this dataset and subsequently demonstrate performance gains on other benchmarks (such as VSI-Bench, OmniSpatial, and SPACE), I would be much more inclined to recognize the dataset’s contribution.
>
> **Response to Q1**: We appreciate the reviewer’s question about overfitting and cross-benchmark generalization. We would like to clarify that SpinBench is a test-only diagnostic benchmark, not a training dataset. Because models are *not* trained on SpinBench, concerns about “overfitting to its own format” do not apply, models' evaluation results reflects their underlying spatial reasoning capabilities rather than adaptation to a specific data distribution.
>
> In addition, SpinBench is intentionally designed to probe spatial factors that **differ** from other benchmarks. Given that SpinBench tests distinct cognitive primitives, good performance on SpinBench would not necessarily imply good performance on benchmarks that target different skills. Conversely, performing well on broad spatial benchmarks does not guarantee competence on the fine-grained spatial transformations evaluated by SpinBench.
>
> Taken together, SpinBench’s value lies in its ability to **diagnose** specific spatial reasoning weaknesses that are not captured by existing benchmarks, rather than serving as a training set for cross-benchmark generalization.

---

### Author Response · Authors · 2025-11-23
**General Response**

We sincerely thank reviewers and the AC/SAC for their time, constructive feedback, and thoughtful comments. We are encouraged that SpinBench was recognized as well-motivated (`kk9w`, `cvAh`), novel in its progressive, cognitively grounded diagnostic structure (`kk9w`, `fu7D`, `cvAh`), and supported by comprehensive evaluations (`t7Ct`, `fu7D`, `cvAh`, `SReC`). We also appreciate the reviewers noting that the design is empirically validated through strong correlations with human response time (`t7Ct`, `fu7D`) and yields insightful and actionable findings about spatial reasoning weaknesses (`kk9w`, `SReC`, `fu7D`).
To address the reviewers’ concerns, we have conducted additional experiments and analyses, which further validate SpinBench’s design and strengthen our findings:

- Additional experiments: adding evaluations of GPT-5, o4-mini, and Gemini 2.5 Pro, with GPT-5 being the new overall best, and showing the first promising improvement in both egocentric and allocentric reasoning. (`t7Ct`, `kk9w`)
- Additional experiments: fine-tuning on 6 foundational spatial tasks leads to measurable improvements on the held-out perspective-taking task, supporting the benchmark’s progressive diagnostic structure. (`SReC`)
- Additional experiments: evaluating depth-augmented inputs using DepthAnything, showing that VLMs need more advanced fusion to incorporate explicit depth cues. (`cvAh`)
- Additional experiments: evaluating high-resolution visual inputs, showing that core reasoning bottlenecks in SpinBench is in spatial reasoning rather than low resolution. (`fu7D`)
- Additional experiments: we extend the mental-rotation task with two additional visual domains, improving diversity. (`kk9w`, `SReC`)
- Further clarifications: provided in the detailed responses to each reviewer below.


All revisions are marked in blue in the updated manuscript. Below, we respond to each comment in detail. We welcome further discussion and greatly appreciate your input. Thank you!

---

### Author Response · Authors · 2025-11-29
**Summary of Previous Rebuttal and Discussion**

We sincerely thank the AC/SAC for their efforts. Here, we provide a concise summary of our rebuttal and the discussion before the freeze for better context. Besides, we kindly refer the AC to our general response, which summarizes how we addressed all reviewer concerns. For more detailed responses, please refer to our individual responses to each reviewer.

During the previous rebuttal, we were able to engage with reviewer SReC, with two rounds of discussion:

- Round 1: Reviewer SReC replied that most concerns were resolved, and raised their rating from 4 to 6 and confidence from 3 to 4. They asked one more clarification question, which we addressed in round 2.
- Round 2: After receiving our second response, reviewer SReC wrote: “My concerns are now fully addressed.”

The other four reviewers did not respond before the discussion was frozen. However, we have provided detailed responses to each of their concerns.

We would be grateful if the AC could take into account the fully resolved discussion with reviewer SReC, who explicitly noted that all concerns were fully addressed after two rounds of interaction.

We deeply appreciate the AC/SAC’s work under these circumstances. Thank you for your time and consideration!

---

### Meta-Review · Area_Chair_9EsR · 2026-01-02

**Summary:**

The key reviewer concerns are:

* Novelty and positioning: overlap with existing spatial benchmarks; need for quantified orthogonality and cross-benchmark analyses [t7Ct]

* Coverage and scope: tasks largely on a 2D horizontal plane; limited vertical / containment relations [t7Ct]

* Sample size granularity: some subtypes are small (e.g., mental rotation initially), raising reliability questions [kk9w]

* Model coverage: absence of strongest closed models in the initial submission [t7Ct,kk9w]

* Causality and practicality: unclear whether improving foundational skills transfers to perspective-taking; mixed effect of CoT [SReC]

* Input factors: whether failures stem from visual detail / resolution or lack of depth cues vs. core reasoning limits [cvAh,fu7D]

As outlined below, the authors provided a substantial and constructive rebuttal that addresses many of these points with new analyses and experiments. Reviewers exhibit both a positive overall lean and update-trajectory, I recommend acceptance.

**Reviewer Concerns:**

The paper’s strengths were consistently noted: a cognitively grounded diagnostic design that decomposes perspective-taking into fine-grained skills; broad evaluation across 37 VLMs; robust findings including egocentric bias, rotational weaknesses, and logical consistency probes; and a human baseline with response-time correlations supporting cognitive validity.

On novelty / positioning: the authors conducted cross-benchmark correlation analyses showing mostly weak correlations. This supports orthogonality and that SpinBench probes distinct capabilities. This directly targets a central concern about overlap.

On coverage / scope: the authors argue the 2D focus is a deliberate diagnostic choice. The limitation is transparently acknowledged and I tend to agree that it does not detract from the core contribution.

On sample sizes: the authors clarify that reported analyses aggregate into seven core task-families and further expand mental rotation with 140 additional samples. They also explain that very small subtypes are pooled for contrasts, potentially improving statistical stability.

On model coverage: results for GPT-5, o4-mini, and Gemini 2.5 Pro were added which I consider can somewhat strengthen claims and contemporary relevance.

On input factors: authors show that naively appending depth maps reduces accuracy. Higher-resolution inputs yielded no gains, pointing to reasoning bottlenecks rather than resolution limits. They also discuss when CoT helps or hurts and connect to emerging latent visual reasoning directions.

Remaining limitations are mostly scoped to future work: extensions to vertical / containment relations and broader 3D relations; deeper architectural ablations; and more principled depth integration. I tend to broadly agree with these assignments and the central contribution is not undercut.

**Reviewer Scores:**

* Reviewer t7Ct (2): Main concerns were overlap with prior work, 2D restriction, missing top models, and dataset value. The rebuttal added cross-benchmark analyses (orthogonality), expanded model coverage and clarified test-only diagnostic intent plus causal transfer via GRPO. While these directly target the concerns, the reviewer’s initial stance suggests the score may remain low; at best a modest upward shift.

* Reviewer kk9w (6): Concerned about small subtype sizes and missing SOTA models. The added data for mental rotation and inclusion of additional models directly address these; score likely stable.

* Reviewer SReC (4 $\rightarrow$ 6): Sought causal evidence for scaffolding and clarity on CoT inconsistency. The GRPO intervention and additional analyses resolved concerns; reviewer explicitly reported that all concerns were fully addressed.

* Reviewer cvAh (8): Positive on design and analysis; requested depth-input ablation. Authors provided depth-augmented experiments and discussion; score likely stable.

* Reviewer fu7D (8): Positive on design; asked about resolution and emergent vs. smooth scaling. Authors showed no gain from higher resolution and clarified scaling behavior; score likely stable.

Decision recommendation

The submission provides a timely, grounded diagnostic benchmark that surfaces spatial reasoning failures not captured by existing datasets. The authors attempted to substantively address key concerns with new experiments: cross-benchmark orthogonality, expanded SOTA coverage, increased sample sizes and input-factor controls (depth, resolution, CoT). While extensions remain for future work, the present contribution is reasonably significant and well-supported. I recommend acceptance.

---

### Decision · Program_Chairs · 2026-01-26

Accept (Poster)